# OpenPBTA: The Open Pediatric Brain Tumor Atlas

## Graphical abstract

## Highlights

- OpenPBTA collaborative analyses establish resource for 1,074 pediatric brain tumors

- NGS-based WHO-aligned integrated diagnoses generated for 644 of 1,074 tumors

- RNA-Seq analysis infers medulloblastoma subtypes, *TP53* status, and telomerase activity

- OpenPBTA will accelerate therapeutic translation of genomic insights

## Authors

Joshua A. Shapiro, Krutika S. Gaonkar, Stephanie J. Spielman, ..., Jaclyn N. Taroni, Children's Brain Tumor Network, Pacific Pediatric Neuro-Oncology Consortium

## Correspondence

rokita@chop.edu (J.L.R.), jaclyn.taroni@ccdatalab.org (J.N.T.)

## In brief

The OpenPBTA is a global, collaborative open-science initiative that brought together researchers and clinicians to genomically characterize 1,074 pediatric brain tumors and 22 patient-derived cell lines. Shapiro et al. create over 40 open-source, scalable modules to perform cancer genomics analyses and provide a richly annotated somatic dataset across 58 brain tumor histologies. The OpenPBTA framework can be used as a model for large-scale data integration to inform basic research, therapeutic target identification, and clinical translation.

 Shapiro et al., 2023, Cell Genomics *3*, 100340
July 12, 2023 © 2023 The Author(s).

CellPress

**Resource**

# OpenPBTA: The Open Pediatric Brain Tumor Atlas

Joshua A. Shapiro,[1] Krutika S. Gaonkar,[2,3,4] Stephanie J. Spielman,[1,5,36] Candace L. Savonen,[1,37] Chante J. Bethell,[1] Run Jin,[2,3] Komal S. Rathi,[2,4] Yuankun Zhu,[2,3] Laura E. Egolf,[6,7] Bailey K. Farrow,[2,3] Daniel P. Miller,[2,3] Yang Yang,[8] Tejaswi Koganti,[2,3] Nighat Noureen,[9] Mateusz P. Koptyra,[2,3] Nhat Duong,[4] Mariarita Santi,[10,11] Jung Kim,[12] Shannon Robins,[2,3] Phillip B. Storm,[2,3] Stephen C. Mack,[13] Jena V. Lilly,[2,3] Hongbo M. Xie,[4] Payal Jain,[2,3] Pichai Raman,[2,4] Brian R. Rood,[14,15] Rishi R. Lulla,[16,17] Javad Nazarian,[14,15,18] Adam A. Kraya,[2,3] Zalman Vaksman,[7] Allison P. Heath,[2,3] Cassie Kline,[7] Laura Scolaro,[7] Angela N. Viaene,[10,11] Xiaoyan Huang,[2,3] Gregory P. Way,[19] Steven M. Foltz,[1,20] Bo Zhang,[2,3] Anna R. Poetsch,[21,22] Sabine Mueller,[23] Brian M. Ennis,[2,3] Michael Prados,[24] Sharon J. Diskin,[7,25]

*(Author list continued on next page)*

[1]Childhood Cancer Data Lab, Alex's Lemonade Stand Foundation, Bala Cynwyd, PA 19004, USA
[2]Center for Data-Driven Discovery in Biomedicine, Children's Hospital of Philadelphia, Philadelphia, PA 19104, USA
[3]Division of Neurosurgery, Children's Hospital of Philadelphia, Philadelphia, PA 19104, USA
[4]Department of Bioinformatics and Health Informatics, Children's Hospital of Philadelphia, Philadelphia, PA 19104, USA
[5]Rowan University, Glassboro, NJ 08028, USA
[6]Cell and Molecular Biology Graduate Group, Perelman School of Medicine at the University of Pennsylvania, Philadelphia, PA 19104, USA
[7]Division of Oncology, Children's Hospital of Philadelphia, Philadelphia, PA 19104, USA
[8]Ben May Department for Cancer Research, University of Chicago, Chicago, IL 60637, USA
[9]Greehey Children's Cancer Research Institute, UT Health San Antonio, San Antonio, TX 78229, USA
[10]Department of Pathology and Laboratory Medicine, Children's Hospital of Philadelphia, Philadelphia, PA 19104, USA
[11]Department of Pathology and Laboratory Medicine, University of Pennsylvania Perelman School of Medicine, Philadelphia, PA 19104, USA
[12]Clinical Genetics Branch, Division of Cancer Epidemiology and Genetics, National Cancer Institute, Rockville, MD 20850, USA
[13]Department of Developmental Neurobiology, St. Jude Children's Research Hospital, Memphis, TN 38105, USA
[14]Children's National Research Institute, Washington, DC 20012, USA
[15]George Washington University School of Medicine and Health Sciences, Washington, DC 20052, USA
[16]Division of Hematology/Oncology, Hasbro Children's Hospital, Providence, RI 02903, USA
[17]Department of Pediatrics, The Warren Alpert School of Brown University, Providence, RI 02912, USA
[18]Department of Pediatrics, University of Zurich, Zurich, Switzerland
[19]Department of Biomedical Informatics, University of Colorado School of Medicine, Aurora, CO 80045, USA
[20]Department of Systems Pharmacology and Translational Therapeutics, University of Pennsylvania, Philadelphia, PA 19104, USA
[21]Biotechnology Center, Technical University Dresden, Dresden, Germany
[22]National Center for Tumor Diseases, Dresden, Germany
[23]Department of Neurology, Neurosurgery and Pediatrics, University of California, San Francisco, San Francisco, CA 94115, USA
[24]University of California, San Francisco, San Francisco, CA 94115, USA

*(Affiliations continued on next page)*

## SUMMARY

Pediatric brain and spinal cancers are collectively the leading disease-related cause of death in children; thus, we urgently need curative therapeutic strategies for these tumors. To accelerate such discoveries, the Children's Brain Tumor Network (CBTN) and Pacific Pediatric Neuro-Oncology Consortium (PNOC) created a systematic process for tumor biobanking, model generation, and sequencing with immediate access to harmonized data. We leverage these data to establish OpenPBTA, an open collaborative project with over 40 scalable analysis modules that genomically characterize 1,074 pediatric brain tumors. Transcriptomic classification reveals universal *TP53* dysregulation in mismatch repair-deficient hypermutant high-grade gliomas and *TP53* loss as a significant marker for poor overall survival in ependymomas and H3 K28-mutant diffuse midline gliomas. Already being actively applied to other pediatric cancers and PNOC molecular tumor board decision-making, OpenPBTA is an invaluable resource to the pediatric oncology community.

## INTRODUCTION

Pediatric brain and spinal cord tumors are collectively the second most common malignancy in children after leukemia, representing the leading disease-related cause of death in children.[1] Five-year survival rates vary widely across different histologic and molecular classifications of brain tumors. For example, most high-grade gliomas carry a universally fatal prognosis,

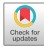

Siyuan Zheng,[9] Yiran Guo,[2] Shrivats Kannan,[2,3] Angela J. Waanders,[26,27] Ashley S. Margol,[28,29] Meen Chul Kim,[2,3] Derek Hanson,[30,31] Nicholas Van Kuren,[2,3] Jessica Wong,[2,3] Rebecca S. Kaufman,[4,7] Noel Coleman,[2,3] Christopher Blackden,[2,3] Kristina A. Cole,[7,25,32] Jennifer L. Mason,[2,3] Peter J. Madsen,[2,3] Carl J. Koschmann,[33,34] Douglas R. Stewart,[12] Eric Wafula,[4] Miguel A. Brown,[2,3] Adam C. Resnick,[2,3] Casey S. Greene,[1,19,35,38,39] Jo Lynne Rokita,[2,3,4,40,*] and Jaclyn N. Taroni[1,*] Children's Brain Tumor Network, Pacific Pediatric Neuro-Oncology Consortium

[25]Department of Pediatrics, University of Pennsylvania, Philadelphia, PA 19104, USA
[26]Division of Hematology, Oncology, Neuro-Oncology, and Stem Cell Transplant, Ann & Robert H Lurie Children's Hospital of Chicago, Chicago, IL 60611, USA
[27]Department of Pediatrics, Northwestern University Feinberg School of Medicine, Chicago, IL 60611, USA
[28]Division of Hematology and Oncology, Children's Hospital of Los Angeles, Los Angeles, CA 90027, USA
[29]Department of Pediatrics, Keck School of Medicine of University of Southern California, Los Angeles, CA 90033, USA
[30]Hackensack Meridian School of Medicine, Nutley, NJ 07110, USA
[31]Hackensack University Medical Center, Hackensack, NJ 07601, USA
[32]Abramson Family Cancer Research Institute, Perelman School of Medicine at the University of Pennsylvania, Philadelphia, PA 19104, USA
[33]Department of Pediatrics, University of Michigan Health, Ann Arbor, MI 48105, USA
[34]Pediatric Hematology Oncology, Mott Children's Hospital, Ann Arbor, MI 48109, USA
[35]Department of Systems Pharmacology and Translational Therapeutics, Perelman School of Medicine, University of Pennsylvania, Philadelphia, PA 19104, USA
[36]Present address: Childhood Cancer Data Lab, Alex's Lemonade Stand Foundation, Bala Cynwyd, PA 19004, USA
[37]Present address: Fred Hutchinson Cancer Center, Seattle, WA 98109, USA
[38]Present address: Center for Health AI, University of Colorado School of Medicine, Aurora, CO 80045, USA
[39]Present address: Department of Biomedical Informatics, University of Colorado School of Medicine, Aurora, CO 80045, USA
[40]Lead contact
*Correspondence: rokita@chop.edu (J.L.R.), jaclyn.taroni@ccdatalab.org (J.N.T.)

while children with pilocytic astrocytoma have an estimated 10-year survival rate of 92%.[2] Recent estimates suggest that children and adolescents aged 0–19 with brain tumors in the United States lose an average 47,631 years of life.[3]

The low survival rates for some pediatric tumors are multifactorial, explained partly by our lack of comprehensive understanding of ever-evolving brain tumor molecular subtypes, by difficulty drugging these tumors, and by the shortage of drugs specifically labeled for pediatric malignancies. Historically, fatal inoperable brain tumors, such as diffuse intrinsic pontine gliomas (DIPGs), were not routinely biopsied due to perceived biopsy risks and the paucity of therapeutic options. Thus, combined with rare incidences of pediatric tumors in the first place, limited availability of tissue for developing patient-derived cell lines and mouse models has hindered research.

To address these barriers, multiple national and international consortia have collaborated to uniformly collect clinically annotated surgical biosamples and associated germline materials through both observational and interventional clinical trials. The Pediatric Brain Tumor Atlas (PBTA) initiative, established in 2018 by the Children's Brain Tumor Network (CBTN, https://cbtn.org)[4] and the Pacific Pediatric Neuro-Oncology Consortium (PNOC, https://pnoc.us), built upon 12 years of enrollment, sample collection, and clinical followup across over 30 institutions. Just as cooperation accelerates specimens and data sharing, collaboration among computational researchers, bench scientists, clinicians, and pathologists is critical for rigorous genomic analysis.

Although there has been significant progress elucidating genomic bases of pediatric brain tumor formation and progression, translating therapeutic agents to phase II or III clinical trials and subsequent FDA approvals have not kept pace. Within the last 20 years, the FDA has approved only seven targeted agents for treating pediatric brain tumors.[5] This is partly due to pharmaceutical company priorities, posing challenges for researchers to obtain therapeutic agents for pediatric clinical trials. Critically, since August 2020, an amendment to the Pediatric Research Equity Act called the "Research to Accelerate Cures and Equity (RACE) for Children Act" mandates that all new adult oncology drugs also be tested in children when the molecular target exists in a childhood cancer. The RACE Act, coupled with genomics advances to identify putative molecular targets in pediatric cancers, will accelerate identification of previously overlooked but effective therapeutic options for pediatric diseases.

We anticipated that a model of open collaboration would enhance the PBTA's value and provide a framework for ongoing analysis of pediatric brain tumor datasets. Leveraging diverse scientific and analytical expertise, we established the OpenPBTA, which employs an open science model with features such as analytical code review[6,7] and continuous integration,[7,8] thereby ensuring reproducibility throughout the project's lifetime. Through the OpenPBTA, we present a comprehensive, collaborative, open genomic analysis of 1,074 tumors and 22 cell lines, comprised of 58 distinct brain tumor histologies from 943 patients. The data and containerized infrastructure of the OpenPBTA have already supported discovery and translational research studies,[9–12] are actively integrated into PNOC molecular tumor board decision-making, and have provided a foundational layer for the Childhood Cancer Data Initiative's (CCDI) recently established pediatric Molecular Targets Platform (https://moleculartargets.ccdi.cancer.gov/). We anticipate that the OpenPBTA will continue to be invaluable to the pediatric oncology community.

## Resource

CellPress

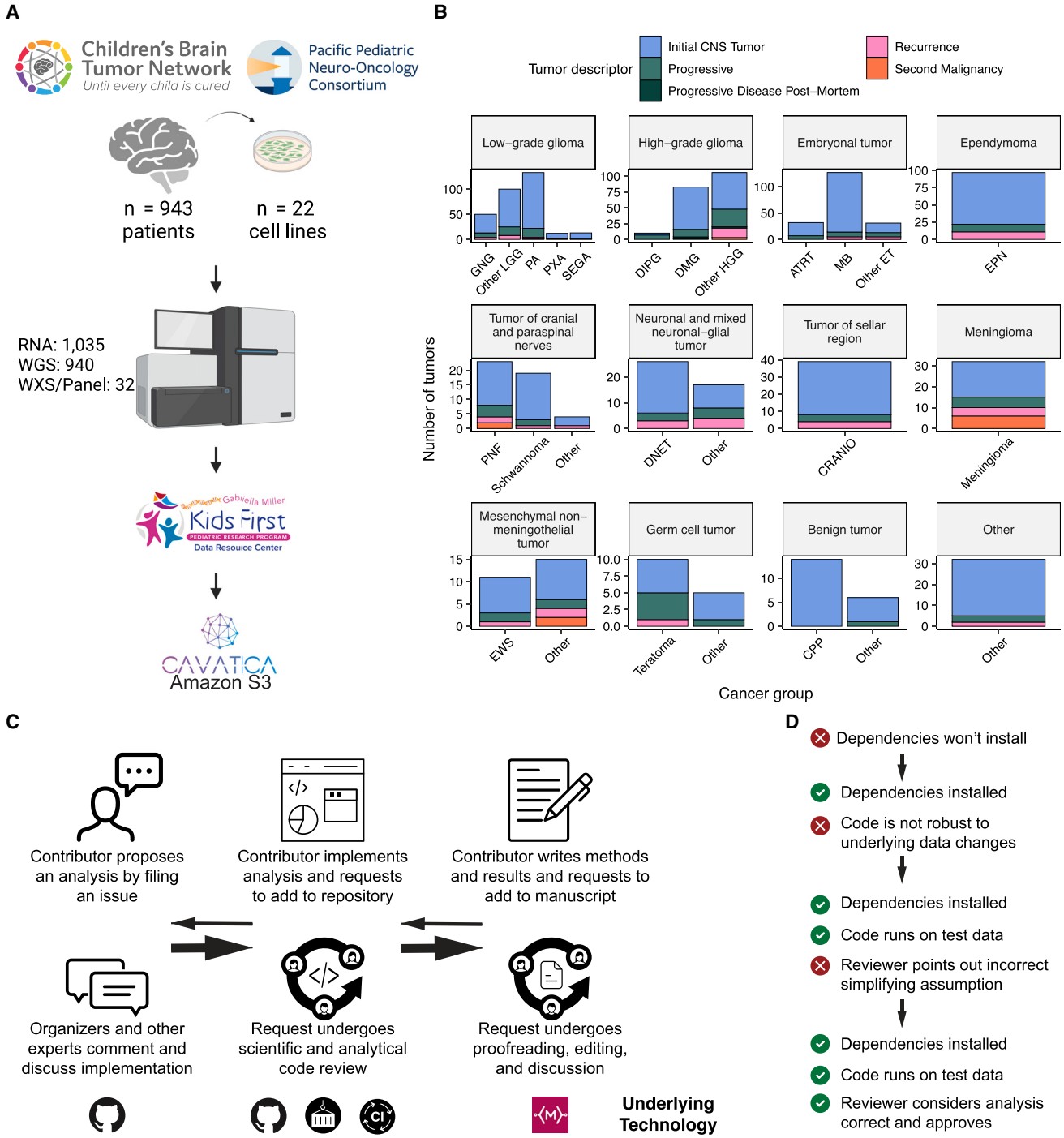

**Figure 1. Overview of the OpenPBTA project**

(A) CBTN and PNOC collected tumors from 943 patients. 22 tumor cell lines were created, and over 2,000 specimens were sequenced (n = 1035 RNA-seq, n = 940 WGS, and n = 32 WXS or targeted panel). The Kids First Data Resource Center Data harmonized the data using Amazon S3 through CAVATICA. Panel created with BioRender.com.

(B) Number of biospecimens across phases of therapy, with one broad histology per panel. Each bar denotes a cancer group (abbreviations: GNG, ganglioglioma; other LGG, other low-grade glioma; PA, pilocytic astrocytoma; PXA, pleomorphic xanthoastrocytoma; SEGA, subependymal giant cell astrocytoma; DIPG, diffuse intrinsic pontine glioma; DMG, diffuse midline glioma; other HGG, other high-grade glioma; ATRT, atypical teratoid rhabdoid tumor; MB, medulloblastoma; other ET, other embryonal tumor; EPN, ependymoma; PNF, plexiform neurofibroma; DNET, dysembryoplastic neuroepithelial tumor; CRANIO, craniopharyngioma; EWS, Ewing sarcoma; CPP, choroid plexus papilloma).

*(legend continued on next page)*

## RESULTS

### Crowd-sourced somatic analyses to create an open PBTA

We previously performed whole-genome sequencing (WGS), whole-exome sequencing (WXS), and RNA sequencing (RNA-seq) on matched tumor/normal tissues and selected cell lines[13] from 943 patients from the PBTA, consisting of 911 patients from the CBTN[4] and 32 patients from PNOC[10,14] (Figure 1A) across various histologies phrases of therapy (Figure 1B). We harnessed and extended the benchmarking efforts of the Gabriella Miller Kids First Data Resource Center to develop robust and reproducible data analysis workflows within the CAVATICA platform for comprehensive somatic analyses (Figure S1; STAR Methods) of the PBTA.

A key innovative feature of the OpenPBTA is its open contribution framework used for analytical code and manuscript writing. We created a public GitHub analysis repository (https://github.com/AlexsLemonade/OpenPBTA-analysis) to hold all analysis code downstream of Kids First workflows and a GitHub manuscript repository (https://github.com/AlexsLemonade/OpenPBTA-manuscript) with Manubot[15] integration to enable real-time manuscript creation. As all analyses and manuscript writing were conducted in public repositories, any researcher in the world could contribute to the OpenPBTA following the process outlined in Figure 1C. First, a potential contributor proposed an analysis by filing an issue in the GitHub analysis repository. Next, project organizers or other contributors with expertise provided feedback about the proposed analysis (Figure 1C). The contributor formally requested to include their analytical code and results—written in their own copy (fork) of repository—in the OpenPBTA analysis repository by filing a GitHub pull request (PR). All PRs underwent peer review to ensure scientific accuracy, maintainability, and readability of code and documentation (Figures 1C and 1D).

Beyond peer review, we implemented additional checks to ensure consistent results for all collaborators over time (Figure 1D). To provide a consistent software development environment, we created a monolithic image with all OpenPBTA dependencies using Docker[16] and the Rocker project.[17] We used the continuous integration (CI) service CircleCI to run analytical code in PRs on a test dataset before formal code review, allowing us to detect code bugs or sensitivity to data release changes.

We followed a similar process in our Manubot-powered[15] repository for proposed manuscript additions (Figure 1C); peer reviewers ensured clarity and scientific accuracy, and Manubot performed spellchecking.

### Molecular subtyping of OpenPBTA CNS tumors

Since 2000, neuro-oncology experts and the WHO have collaborated to iteratively redefine central nervous system (CNS) tumor classifications.[18,19] In 2016,[20] molecular subtypes driven by genetic alterations were integrated into these classifications. Since CBTN specimen collection began in 2011, most tumors lacked molecular subtype information when tissue was collected. Moreover, the PBTA does not yet feature methylation arrays, which are increasingly used to inform molecular subtyping and cancer diagnosis. Therefore, we created analysis modules to systematically consider key genomic features of tumors described by the WHO in 2016 or Ryall and colleagues.[21] Coupled with clinician and pathologist review, we generated high-confidence, research-grade integrated diagnoses for 60% (644/1,074) of tumors (Table S1) without methylation data, a major innovation of this project. We then aligned OpenPBTA specimen diagnoses with whom classifications (e.g., tumors formerly ascribed primitive neuro-ectodermal tumor [PNET] diagnoses) discovered rarer tumor entities (e.g., H3-mutant ependymoma, meningioma with YAP1::FAM118B fusion), as well as identified and corrected data entry errors (e.g., an embryonal tumor with multilayer rosettes (ETMRs) incorrectly entered as a medulloblastoma) and histologically mis-identified specimens (e.g., Ewing sarcoma sample labeled as a craniopharyngioma). Uniquely, we used transcriptomic classification to subtype 122 medulloblastomas into SHH, WNT, group 3, or group 4 with MedulloClassifier[22] and MM2S,[23] with 95% (41/43) and 91% (39/43) accuracy, respectively.

In total, we subtyped low-grade gliomas (LGGs) (n = 290), high-grade gliomas (HGGs) (n = 141), embryonal tumors (n = 126), ependymomas (n = 33), tumors of sellar region (n = 27), mesenchymal non-meningothelial tumors (n = 11), glialneuronal tumors (n = 10), and chordomas (n = 6), where n represents unique tumors (Table 1). For detailed methods, see STAR Methods and Figure S1.

### Somatic mutational landscape of pediatric brain tumors

We performed a comprehensive genomic analysis of somatic SNVs, copy number variants (CNVs), structural variants (SVs), and fusions across all 1,074 PBTA tumors (n = 1,019 RNA-seq, n = 918 WGS, n = 32 WXS/panel) and 22 cell lines (n = 16 RNA-Seq, n = 22 WGS) from 943 patients, 833 with paired normal specimens (n = 801 WGS, n = 32 WXS/panel). Tumor purity across PBTA samples was high (median 76%), though we observed some cancer groups with lower purity, including subependymal giant cell astrocytoma (SEGA), pleomorphic xanthoastrocytoma (PXA), and teratoma (Figure S3A). Unless otherwise noted, each analysis was performed for diagnostic tumors using one tumor per patient.

SNV consensus calling (Figures S1 and S2A–S2G) revealed, as expected, lower tumor mutation burden (TMB) (Figure S2H) in pediatric tumors compared with adult brain tumors from The Cancer Genome Atlas (TCGA) (Figure S2I), with hypermutant (>10 mutations [Mut]/Mb) and ultra-hypermutant (>100 Mut/Mb) tumors[24] only found within HGGs and embryonal tumors. Figures 2 and S3B depict oncoprints recapitulating known histology-specific driver genes in primary tumors across OpenPBTA

---

(C) Overview of the open analysis and manuscript contribution models. Contributors proposed analyses, implemented it in their fork, and filed a pull request (PR) with proposed changes. PRs underwent review for scientific rigor and accuracy. Container and continuous integration technologies ensured that all software dependencies were included and that code was not sensitive to underlying data changes. Finally, a contributor filed a PR documenting their methods and results to the Manubot-powered manuscript repository for review.

(D) A potential path for an analytical PR. Arrows indicate revisions.

histologies, and Table S2 summarizes all detected alterations across cancer groups.

### LGGs

As expected, most (62%, 140/226) LGGs harbored a somatic alteration in *BRAF*, with canonical *BRAF::KIAA1549* fusions as the major oncogenic driver[25] (Figure 2A). We observed additional mutations in *FGFR1* (2%), *PIK3CA* (2%), *KRAS* (2%), *TP53* (1%), and *ATRX* (1%) and fusions in *NTRK2* (2%), *RAF1* (2%), *MYB* (1%), *QKI* (1%), *ROS1* (1%), and *FGFR2* (1%), concordant with previous studies reporting near-universal upregulation of the RAS/MAPK pathway in LGGs.[21,25] Indeed, gene set variant analysis (GSVA) revealed significant upregulation (ANOVA Bonferroni-corrected p < 0.01) of the KRAS signaling pathway in LGGs (Figure 5B).

### Embryonal tumors

Most (n = 95) embryonal tumors were medulloblastomas from four characterized molecular subtypes (WNT, SHH, group 3, and group 4; see molecular subtyping of OpenPBTA CNS tumors), as identified by subtype-specific canonical mutations (Figure 2B). We detected canonical *SMARCB1/SMARCA4* deletions or inactivating mutations in atypical teratoid rhabdoid tumors (ATRTs; Table S2) and C19MC amplification in ETMRs (displayed within "other embryonal tumors" in Figure 2B).[26–29]

### HGGs

Across HGGs, *TP53* (57%, 36/63) and *H3F3A* (54%, 34/63) were both most mutated and co-occurring genes (Figures 2A and 2C), followed by frequent mutations in *ATRX* (29%, 18/63), which is commonly mutated in gliomas.[30] We observed recurrent amplifications and fusions in *EGFR*, *MET*, *PDGFRA*, and *KIT*, highlighting that these tumors leverage multiple oncogenic mechanisms to activate tyrosine kinases, as previously reported.[14,31,32] GSVA showed upregulation (ANOVA Bonferroni-corrected p < 0.01) of DNA repair, G2M checkpoint, and MYC pathways as well as downregulation of the TP53 pathway (Figure 5B). The two ultra-hypermutated tumors (>100 Mut/Mb) were from patients with mismatch repair deficiency syndrome.[13]

### Other CNS tumors

We observed that 25% (15/60) of ependymomas were *C11orf95::RELA* (now, *ZFTA::RELA*) fusion positive[33] and that 68% (2⅓ 1) of craniopharyngiomas contained *CTNNB1* mutations (Figure 2D). We observed somatic mutations or fusions in *NF2* in 41% (7/17) of meningiomas, 5% (3/60) of ependymomas, and 25% (3/12) of schwannomas, as well as rare fusions in *ERBB4*, *YAP1*, and/or *QKI* in 10% (6/60) of ependymomas. Dysembryoplastic neuroepithelial tumors (DNETs) harbored alterations in MAPK/PI3K pathway genes, as was previously reported,[34] including *FGFR1* (21%, 4/19), *PDGFRA* (10%, 2/19), and *BRAF* (5%, 1/19).

### Mutational co-occurrence, CNV, and signatures highlight key oncogenic drivers

We analyzed mutational co-occurrence across the OpenPBTA using a single tumor from each patient (n = 668) with WGS. The top 50 mutated genes (see STAR Methods for details) in primary tumors are shown in Figure 3 by tumor type (Figure 3A, bar plots), with co-occurrence scores illustrated in the heatmap (Figure 3B). As expected, *TP53* was the most frequently mutated gene across the OpenPBTA (8.7%, 58/668), significantly co-

occurring with *H3F3A* (odds ratio [OR] = 30.05, 95% confidence interval [CI]: 14.5–62.3, q = 2.34e−16), *ATRX* (OR = 23.3, 95% CI: 9.6–56.3, q = 8.72e−9), *NF1* (OR = 8.26, 95% CI: 3.5–19.4, q = 7.40e−5), and *EGFR* (OR = 17.5, 95% CI: 4.8–63.9, q = 2e−4), with all of these driven by HGGs and consistent with previous reports.[31,35,36]

In embryonal tumors, *CTNNB1* mutations significantly co-occurred with *TP53* mutations (OR = 43.6 95% CI: 7.1–265.8, q = 1.52e−3) as well as with *DDX3X* mutations (OR = 21.4, 95% CI: 4.7–97.9, q = 4.15e−3), events driven by medulloblastomas as previously reported.[37,38] *FGFR1* and *PIK3CA* mutations significantly co-occurred in LGGs (OR = 77.25, 95% CI: 10.0–596.8, q = 3.12e−3), consistent with previous findings.[38,39] Of HGG tumors with *TP53* or *PPM1D* mutations, 53/55 (96.3%) had mutations in only one of these genes (OR = 0.17, 95% CI: 0.04–0.89, q = 0.056), recapitulating previous observations that these mutations are usually mutually exclusive in HGGs.[36]

CNV and SV analyses revealed that HGG, diffuse midline glioma (DMG), and medulloblastoma tumors had the most unstable genomes, while craniopharyngiomas and schwannomas generally lacked somatic CNVs (Figure S3C). These CNV patterns largely aligned with our TMB estimates (Figure S2H). SV and CNV breakpoint densities were significantly correlated (linear regression p = 1.05e−38; Figure 3C), and as expected, the number of chromothripsis regions called increased with breakpoint density (Figures S3D and S3E). We identified chromothripsis events in 31% (n = 12/39) of DMGs and in 44% (n = 21/48) of other HGGs (Figure 3D) and found evidence of chromothripsis in over 15% of sarcomas, PXAs, metastatic secondary tumors, chordomas, glial-neuronal tumors, germinomas, meningiomas, ependymomas, medulloblastomas, ATRTs, and other embryonal tumors.

We assessed the contributions of eight adult CNS-specific mutational signatures from the RefSig database[40] across tumors (Figures 3E and S4A). Signature 1, which reflects normal spontaneous deamination of 5-methylcytosine, predominated in stage 0 and/or 1 tumors characterized by low TMBs (Figure S2H) such as pilocytic astrocytomas, gangliogliomas, other LGGs, and craniopharyngiomas (Figure S4A). Signature 1 exposures were generally higher in tumors sampled at diagnosis (pretreatment) compared with tumors from later phases of therapy (Figure S4B). This trend may have emerged from therapy-induced mutations that produced additional signatures (e.g., temozolomide treatment has been suggested to drive signature 11[41]), subclonal expansion, and/or acquisition of additional driver mutations during tumor progression, leading to detection of additional signatures. We observed the CNS-specific signature N6 in nearly all tumors. Signature 18 drivers (*TP53*, *APC*, *NOTCH1*; found at https://signal.mutationalsignatures.com/explore/referenceCancerSignature/31/drivers) are also canonical medulloblastoma drivers, and indeed, signature 18 had the highest signature weight in medulloblastomas. Finally, signatures 3, 8, 18, and MMR2 were prevalent in HGGs, including DMGs.

### Transcriptomic landscape of pediatric brain tumors

Most RNA-seq samples in the PBTA were prepared with ribosomal RNA depletion followed by stranded sequencing (n = 977), while remaining samples were prepared with poly-A

**Table 1. Molecular subtypes generated through the OpenPBTA project**

| Broad histology group | OpenPBTA molecular subtype | Patients | Tumors |
|---|---|---|---|
| Chordoma | CHDM, conventional | 2 | 2 |
| | CHDM, poorly differentiated | 2 | 4 |
| Embryonal tumor | CNS embryonal, NOS | 13 | 13 |
| | CNS HGNET-MN1 | 1 | 1 |
| | CNS NB-FOXR2 | 2 | 3 |
| | ETMR, C19MC altered | 5 | 5 |
| | ETMR, NOS | 1 | 1 |
| | MB, group 3 | 14 | 14 |
| | MB, group 4 | 48 | 49 |
| | MB, SHH | 24 | 30 |
| | MB, WNT | 10 | 10 |
| Ependymoma | EPN, H3 K28 | 1 | 1 |
| | EPN, ST RELA | 25 | 28 |
| | EPN, ST YAP1 | 3 | 4 |
| High-grade glioma | DMG, H3 K28 | 18 | 24 |
| | DMG, H3 K28, TP53 activated | 10 | 13 |
| | DMG, H3 K28, TP53 loss | 30 | 40 |
| | HGG, H3 G35 | 3 | 3 |
| | HGG, H3 G35, TP53 loss | 1 | 1 |
| | HGG, H3 wild type | 26 | 31 |
| | HGG, H3 wild type, TP53 activated | 5 | 5 |
| | HGG, H3 wild type, TP53 loss | 14 | 21 |
| | HGG, IDH, TP53 activated | 1 | 2 |
| | HGG, IDH, TP53 loss | 1 | 1 |
| Low-grade glioma | GNG, BRAF V600E | 13 | 13 |
| | GNG, BRAF V600E, CDKN2A/B | 1 | 1 |
| | GNG, FGFR | 1 | 1 |
| | GNG, H3 | 1 | 1 |
| | GNG, IDH | 1 | 2 |
| | GNG, KIAA1549-BRAF | 5 | 5 |
| | GNG, MYB/MYBL1 | 1 | 1 |
| | GNG, NF1-germline | 1 | 1 |
| | GNG, NF1-somatic, BRAF V600E | 1 | 1 |
| | GNG, other MAPK | 4 | 4 |
| | GNG, other MAPK, IDH | 1 | 1 |
| | GNG, RTK | 2 | 3 |
| | GNG, wild type | 14 | 14 |
| | LGG, BRAF V600E | 25 | 27 |
| | LGG, BRAF V600E, CDKN2A/B | 5 | 5 |
| | LGG, FGFR | 8 | 8 |
| | LGG, IDH | 3 | 3 |
| | LGG, KIAA1549-BRAF | 106 | 113 |
| | LGG, KIAA1549-BRAF, NF1-germline | 1 | 1 |
| | LGG, KIAA1549-BRAF, other MAPK | 1 | 1 |
| | LGG, MYB/MYBL1 | 2 | 2 |
| | LGG, NF1-germline | 6 | 6 |
| | LGG, NF1-germline, CDKN2A/B | 1 | 1 |
| | LGG, NF1-germline, FGFR | 1 | 2 |

*(Continued on next page)*

**Table 1.** *Continued*

| Broad histology group | OpenPBTA molecular subtype | Patients | Tumors |
|---|---|---|---|
| | LGG, NF1-somatic | 2 | 2 |
| | LGG, NF1-somatic, FGFR | 1 | 1 |
| | LGG, NF1-somatic, NF1-germline, CDKN2A/B | 1 | 1 |
| | LGG, other MAPK | 11 | 12 |
| | LGG, RTK | 8 | 10 |
| | LGG, RTK, CDKN2A/B | 1 | 1 |
| | LGG, wild type | 33 | 34 |
| | SEGA, RTK | 1 | 1 |
| | SEGA, wild type | 10 | 11 |
| Mesenchymal non-meningothelial tumor | EWS | 9 | 11 |
| Neuronal and mixed neuronal-glial tumor | CNC | 2 | 2 |
| | EVN | 1 | 1 |
| | GNT, BRAF V600E | 1 | 1 |
| | GNT, KIAA1549-BRAF | 1 | 2 |
| | GNT, other MAPK | 1 | 1 |
| | GNT, other MAPK, FGFR | 1 | 1 |
| | GNT, RTK | 1 | 2 |
| Tumor of sellar region | CRANIO, ADAM | 27 | 27 |
| Total | – | 577 | 644 |

Broad tumor histologies, molecular subtypes generated, and number of patients and tumors subtyped within the OpenPBTA.

selection (n = 58). Since batch correction was not feasible (see limitations of the study and Figure S7A), the following transcriptomic analyses considered only stranded samples.

***Prediction of TP53 oncogenicity and telomerase activity***

We applied a TCGA-trained classifier[42] to calculate a *TP53* score, a proxy for *TP53* gene or pathway dysregulation, and subsequently infer tumor *TP53* inactivation status. We identified "true positive" *TP53* alterations from high-confidence SNVs, CNVs, SVs, and fusions in *TP53*, annotating tumors as "activated" if they harbored one of the p.R273C or p.R248W gain-of-function mutations[43] or "lost" if (1) the patient had a Li-Fraumeni syndrome (LFS) predisposition diagnosis, (2) the tumor harbored a known hotspot mutation, or (3) the tumor contained two hits (e.g., both SNVs and CNVs), suggesting that both alleles were affected. If the *TP53* mutation did not reside within the DNA-binding domain or no alterations in *TP53* were detected, we annotated the tumor as "other," indicating an unknown *TP53* alteration status. The classifier achieved a high accuracy (area under the receiver operating characteristic curve [AUROC] = 0.86) for rRNA-depleted, stranded tumors, but it did not perform as well on the poly-A tumors in this cohort (AUROC = 0.62; Figure S5A).

We observed that "activated" and "lost" tumors had similar *TP53* scores (Figure 4B; Wilcoxon p = 0.92), contrasting our expectation that "lost" tumors would have higher *TP53* scores. This difference suggests that classifier scores >0.5 may actually represent an oncogenic, or altered, *TP53* phenotype rather than solely *TP53* inactivation, as interpreted previously.[42] However, "activated" tumors showed higher *TP53* expression compared with those with *TP53* "loss" mutations (Wilcoxon p = 0.006; Fig-

ure 4C). DMGs, medulloblastomas, HGGs, DNETs, ependymomas, and craniopharyngiomas, all known to harbor *TP53* mutations, had the highest median *TP53* scores (Figure 4D). By contrast, gangliogliomas, LGGs, meningiomas, and schwannomas had the lowest median scores.

We hypothesized that tumors (n = 10) from patients with LFS (n = 8) would have higher *TP53* scores, which we indeed observed for 8/10 tumors (Table S3). Although two tumors had low *TP53* scores (BS_DEHJF4C7 at 0.09 and BS_ZD5HN296 at 0.28), pathology reports confirmed that both patients were diagnosed with LFS and harbored a *TP53* pathogenic germline variant. These two LFS tumors also had low tumor purity (16% and 37%, respectively), suggesting that accurate classification may require a certain level of tumor content. We suggest that this classifier could be generally applied to infer *TP53* function in the absence of a predicted oncogenic *TP53* alteration or DNA sequencing.

We used gene expression data to predict telomerase activity using expression-based telomerase enzymatic activity detection (*EXTEND*)[44] as a surrogate measure of malignant potential,[44,45] where higher *EXTEND* scores indicate higher telomerase activity. Aggressive tumors such as DMGs, other HGGs, and medulloblastoma (MB) had high *EXTEND* scores (Figure 4D), and low-grade lesions such as schwannomas, gangliogliomas (GNGs), DNETs, and other LGGs had among the lowest scores (Table S3), supporting previous reports that aggressive tumor phenotypes have higher telomerase activity.[46–49] While *EXTEND* scores were not significantly higher in tumors with *TERT* promoter (TERTp) mutations (n = 6; Wilcoxon p value = 0.1196), scores were significantly correlated with *TERC* (R = 0.619,

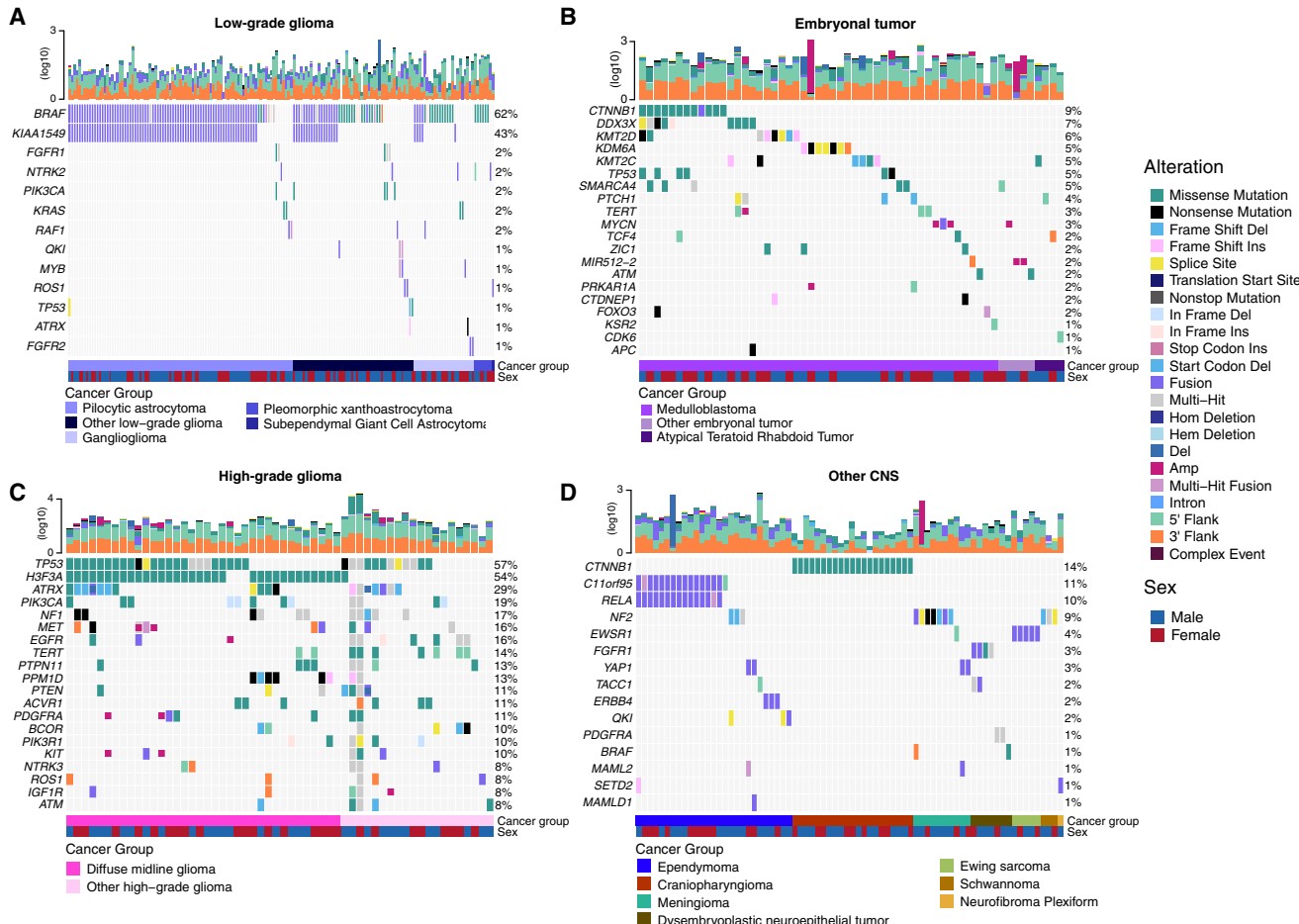

**Figure 2. Mutational landscape of PBTA tumors**
Frequencies of canonical somatic gene mutations, CNVs, fusions, and TMB (top bar plot) for the top mutated genes across primary tumors within the OpenPBTA dataset.
(A) LGGs (n = 226): pilocytic astrocytoma (n = 104), other LGG (n = 68), ganglioglioma (n = 35), pleomorphic xanthoastrocytoma (n = 9), and subependymal giant cell astrocytoma (n = 10).
(B) Embryonal tumors (n = 129): medulloblastoma (n = 95), atypical teratoid rhabdoid tumor (n = 24), and other embryonal tumor (n = 10).
(C) HGGs (n = 63): diffuse midline glioma (n = 36) and other HGG (n = 27).
(D) Other CNS tumors (n = 153): ependymoma (n = 60), craniopharyngioma (n = 31), meningioma (n = 17), dysembryoplastic neuroepithelial tumor (n = 19), Ewing sarcoma (n = 7), schwannoma (n = 12), and neurofibroma plexiform (n = 7). Rare CNS tumors are displayed in Figure S3B. Histology (cancer group) and sex annotations are displayed under each plot. Only tumors with mutations in the listed genes are shown. Multiple CNVs are denoted as a complex event. n denotes the number of unique tumors (one tumor per patient).

p < 0.01) and *TERT* (R = 0.491, p < 0.01) log2 FPKM expression values (Figures S5B and S5C). Since catalytically active telomerase requires full-length *TERT*, *TERC*, and certain accessory proteins,[50] we expect that *EXTEND* scores may not be exclusively correlated with *TERT* alterations and expression.

### Hypermutant tumors share mutational signatures and have dysregulated *TP53*

We investigated the mutational signature profiles of hypermutant (TMB >10 Mut/Mb; n = 3) and ultra-hypermutant (TMB >100 Mut/Mb; n = 4) tumors and/or derived cell lines from six patients in the OpenPBTA (Figure 4E). Five tumors were HGGs and one was a brain metastasis of an MYCN non-amplified neuroblastoma tumor. Signature 11, which is associated with exposure to temozo-

lomide plus *MGMT* promoter and/or mismatch repair deficiency,[51] was indeed present in tumors with previous exposure to the drug (Table 2). We detected the MMR2 signature in tumors of four patients (PT_0SPKM4S8, PT_3CHB9PK5, PT_JNEV57VK, and PT_VTM2STE3) diagnosed with either constitutional mismatch repair deficiency (CMMRD) or Lynch syndrome (Table 2), genetic predisposition syndromes caused by a variant in a mismatch repair gene such as *PMS2*, *MLH1*, *MSH2*, *MSH6*, or others.[52] Three of these patients harbored pathogenic germline variants in one of the aforementioned genes. While we did not detect a known pathogenic variant in the germline of PT_VTM2STE3, this patient's pathology report contained a self-reported *PMS2* variant, and we indeed found

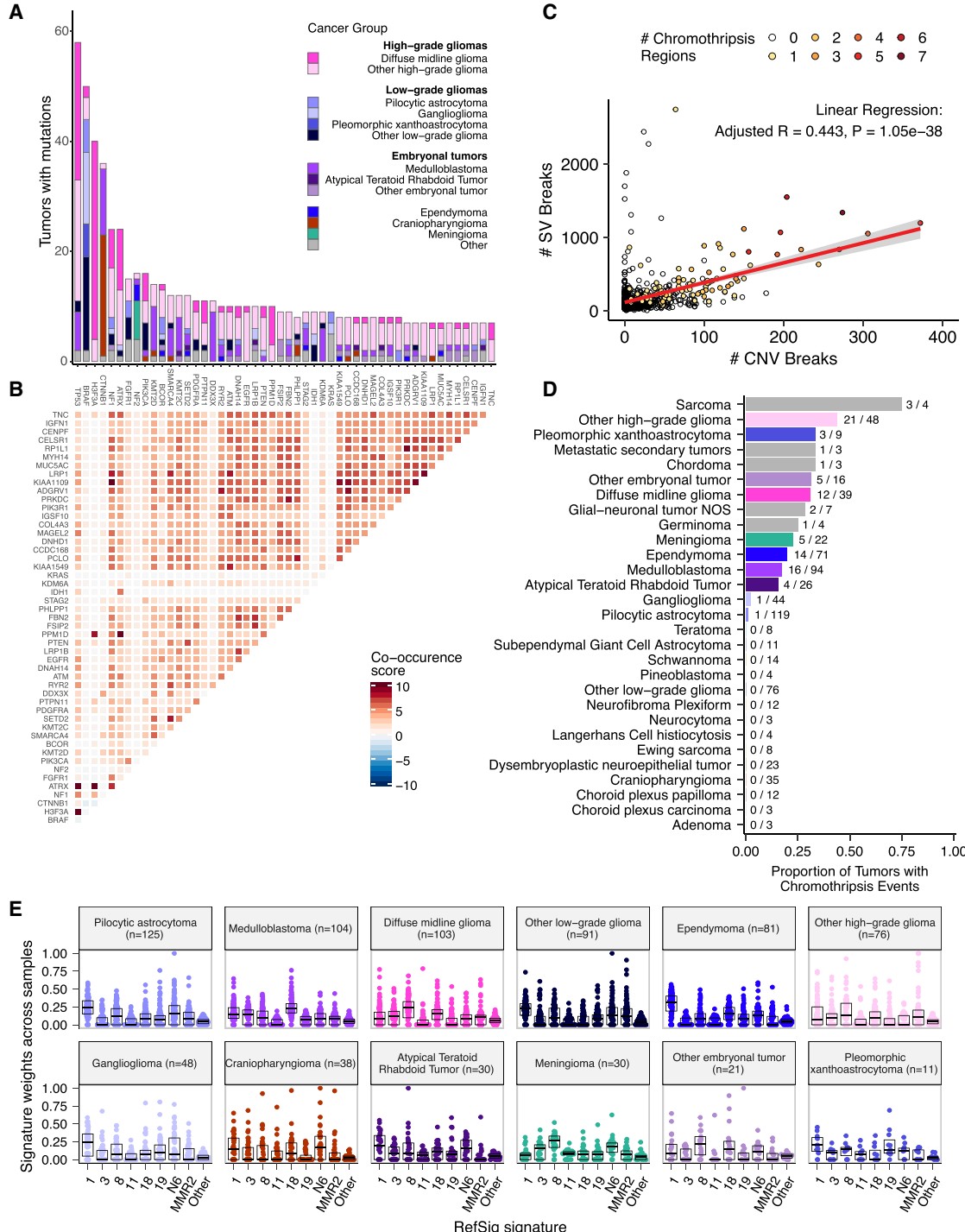

**Figure 3. Mutational co-occurrence and signatures highlight key oncogenic drivers**

(A) Non-synonymous mutations for 50 most commonly mutated genes across all histologies. "Other" denotes a histology with <10 tumors.

(B) Co-occurrence and mutual exclusivity of mutated genes. The co-occurrence score is defined as $I(-\log_{10}(P))$ where $P$ is Fisher's exact test and $I$ is 1 when mutations co-occur more often than expected or −1 when exclusivity is more common.

(C) Number of SV and CNV breaks are significantly correlated (adjusted R = 0.443, p = 1.05e−38).

(D) Chromothripsis frequency across cancer groups with n ≥ 3 tumors.

(E) Sina plots of RefSig signature weights for signatures 1, 11, 18, 19, 3, 8, N6, MMR2, and other across cancer groups. Boxplot represents 5% (lower whisker), 25% (lower box), 50% (median), 75% (upper box), and 95% (upper whisker) quantiles.

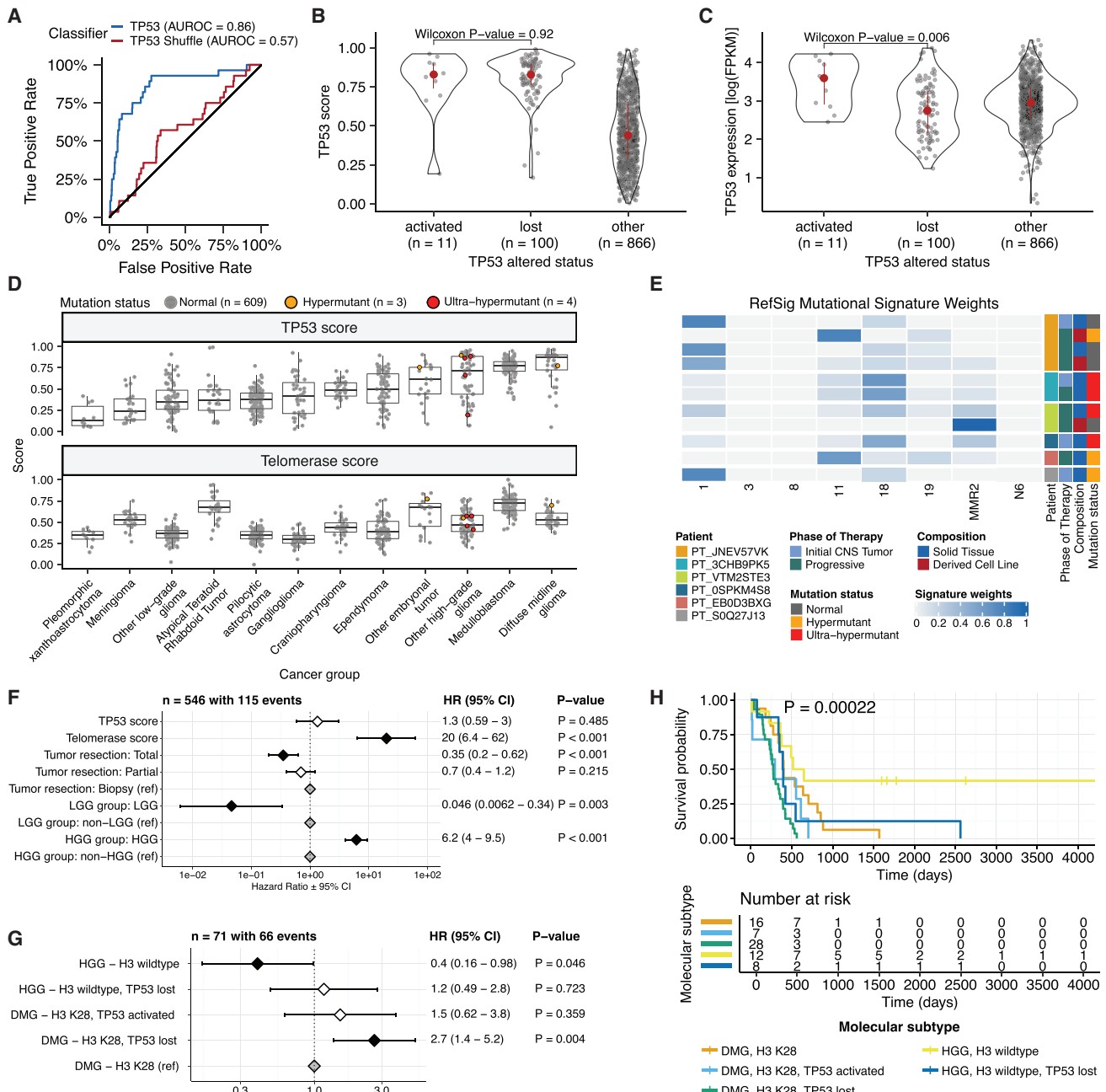

**Figure 4. *TP53* and telomerase activity**

(A) Receiver operating characteristic for *TP53* classifier run on stranded FPKM RNA-seq.

(B) Violin and strip plots of *TP53* scores plotted by *TP53* alteration type ($n_{activated}$ = 11, $n_{lost}$ = 100, $n_{other}$ = 866, Wilcoxon p = 0.92).

(C) Violin and strip plots of *TP53* RNA expression plotted by *TP53* activation status ($n_{activated}$ = 11, $n_{lost}$ = 100, $n_{other}$ = 866, Wilcoxon p = 0.006).

(D) Boxplots of *TP53* and telomerase (EXTEND) scores across cancer groups. TMB status is highlighted in orange (hypermutant) or red (ultra-hypermutant). Boxplot represents 5% (lower whisker), 25% (lower box), 50% (median), 75% (upper box), and 95% (upper whisker) quantiles.

(E) Heatmap of RefSig mutational signatures for patients with at least one hypermutant tumor or cell line.

(F) Forest plot depicting prognostic effects of *TP53* and telomerase scores on overall survival (OS), controlling for extent of tumor resection, LGG group, and HGG group.

(G) Forest plot depicting the effect of molecular subtype on HGG OS. Hazard ratios (HRs) with 95% confidence intervals and p values (multivariate Cox) are given in (F) and (G). Black diamonds denote significant p values, and gray diamonds denote reference groups.

(H) Kaplan-Meier curve of HGGs by molecular subtype.

**Table 2. Patients with hypermutant tumors**

| Kids First participant ID | Kids First biospecimen ID | CBTN ID | Phase of therapy | Composition | Therapy post-biopsy | Cancer predisposition | Pathogenic germline variant | TMB | OpenPBTA molecular subtype |
|---|---|---|---|---|---|---|---|---|---|
| PT_0SPKM4S8 | BS_VW4XN9Y7 | 7316-2640 | initial CNS tumor | solid tissue | radiation, temozolomide, CCNU | none documented | NM_000535.7(PMS2):c.137G>T (p.Ser46Ile) (LP) | 187.4 | HGG, H3 wild type, TP53 activated |
| PT_3CHB9PK5 | BS_20TBZG09 | 7316-515 | initial CNS tumor | solid tissue | radiation, temozolomide, irinotecan, bevacizumab | CMMRD | NM_000179.3(MSH6):c.3439-2A>G (LP) | 307 | HGG, H3 wild type, TP53 loss |
| PT_3CHB9PK5 | BS_8AY2GM4G | 7316-2085 | progressive | solid tissue | radiation, temozolomide, irinotecan, bevacizumab | CMMRD | NM_000179.3(MSH6):c.3439-2A>G (LP) | 321.6 | HGG, H3 wild type, TP53 loss |
| PT_EB0D3BXG | BS_F0GNWEJJ | 7316-3311 | progressive | solid tissue | radiation, nivolumab | none documented | none detected | 26.3 | metastatic NBL, MYCN non-amplified |
| PT_JNEV57VK | BS_85Q5P8GF | 7316-2594 | initial CNS tumor | solid tissue | radiation, temozolomide | Lynch syndrome | NM_000251.3(MSH2):c.1906G>C (p.Ala636Pro) (P) | 4.7 | DMG, H3 K28, TP53 loss |
| PT_JNEV57VK | BS_HM5GFJN8 | 7316-3058 | progressive | derived cell line | radiation, temozolomide, nivolumab | Lynch syndrome | NM_000251.3(MSH2):c.1906G>C (p.Ala636Pro) (P) | 35.9 | DMG, H3 K28, TP53 loss |
| PT_JNEV57VK | BS_QWM9BPDY | 7316-3058 | progressive | derived cell line | radiation, temozolomide, nivolumab | Lynch syndrome | NM_000251.3(MSH2):c.1906G>C (p.Ala636Pro) (P) | 7.4 | DMG, H3 K28, TP53 loss |
| PT_JNEV57VK | BS_P0QJ1QAH | 7316-3058 | progressive | solid tissue | radiation, temozolomide, nivolumab | Lynch syndrome | NM_000251.3(MSH2):c.1906G>C (p.Ala636Pro) (P) | 6.3 | DMG, H3 K28, TP53 activated |
| PT_S0Q27J13 | BS_P3PF53V8 | 7316-2307 | initial CNS tumor | solid tissue | radiation, temozolomide, irinotecan | none documented | none detected | 15.5 | HGG, H3 wild type, TP53 activated |
| PT_VTM2STE3 | BS_ERFMPQN3 | 7316-2189 | progressive | derived cell line | unknown | Lynch syndrome | none detected | 5.7 | HGG, H3 wild type, TP53 loss |
| PT_VTM2STE3 | BS_02YBZSBY | 7316-2189 | progressive | solid tissue | unknown | Lynch syndrome | none detected | 274.5 | HGG, H3 wild type, TP53 activated |

Patients with at least one hypermutant or ultra-hypermutant tumor or cell line. Pathogenic or likely pathogenic germline variants, coding region TMB, phase of therapy, therapeutic interventions, cancer predisposition (constitutional mismatch repair deficiency), and molecular subtypes are included. P, pathogenic; LP, likely pathogenic; CMMRD, constitutional mismatch repair deficiency.

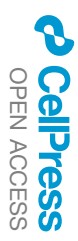

CellPress

19 intronic variants of unknown significance (VUSs) in their *PMS2*. This is not surprising since an estimated 49% of germline *PMS2* variants in patients with CMMRD and/or Lynch syndrome are VUSs.[52] Interestingly, while the cell line derived from patient PT_VTM2STE3's tumor at progression was not hypermutated (TMB = 5.7 Mut/Mb), it only contained the MMR2 signature, suggesting selective pressure to maintain a mismatch repair (MMR) phenotype *in vitro*. Only one of the two cell lines derived from patient *PT_JNEV57VK*'s progressive tumor was hypermutated (TMB = 35.9 Mut/Mb). The hypermutated cell line was strongly weighted toward signature 11, while the non-hypermutated cell line showed several lesser signature weights (1, 11, 18, 19, MMR2; Table S2). This mutational process plasticity highlights the importance of careful genomic characterization and model selection for preclinical studies.

Signature 18, which has been associated with high genomic instability and can induce a hypermutator phenotype,[40] was uniformly represented among hypermutant solid tumors. Additionally, all hypermutant HGG tumors or cell lines had dysfunctional *TP53* (Table 2), consistent with previous findings that tumors with high genomic instability signatures require *TP53* dysregulation.[40] With one exception, hypermutant and ultra-hypermutant tumors had high *TP53* scores (>0.5) and telomerase activity. Interestingly, none of the hypermutant tumors showed evidence of signature 3 (present in homologous recombination-deficient tumors), 8 (arises from double nucleotide substitutions/unknown etiology), or N6 (a universal CNS tumor signature). The mutual exclusivity of signatures 3 and MMR2 corroborates previous suggestions that tumors do not generally feature both deficient homologous repair and MMR.[42]

Next, we asked whether transcriptomic classification of *TP53* dysregulation and/or telomerase activity recapitulate these oncogenic biomarkers' known prognostic influence. We identified several expected trends, including a significant overall survival benefit following full tumor resection (hazard ratio [HR] = 0.35, 95% CI = 0.2–0.62, p < 0.001) or if the tumor was an LGG (HR = 0.046, 95% CI = 0.0062–0.34, p = 0.003), and a significant risk if the tumor was an HGG (HR = 6.2, 95% CI = 4.0–9.5, p < 0.001) (Figure 4F; STAR Methods). High telomerase scores were associated with poor prognosis across brain tumor histologies (HR = 20, 95% CI = 6.4–62, p < 0.001), demonstrating that *EXTEND* scores calculated from RNA-seq are an effective rapid surrogate measure for telomerase activity. Higher *TP53* scores were associated with significant survival risks (Table S4) within DMGs (HR = 6436, 95% CI = 2.67–1.55e7, p = 0.03) and ependymomas (HR = 2003, 95% CI = 9.9–4.05e5, p = 0.005). Given this result, we next assessed whether different HGG molecular subtypes carry different survival risks if stratified by *TP53* status. We found that DMG H3 K28 tumors with *TP53* loss had significantly worse prognosis (HR = 2.8, CI = 1.4–5.6, p = 0.003) than those with WT *TP53* (Figures 4G and 4H), recapitulating results from two recent restrospective analyses of DIPG tumors.[10,53]

### Histologic and oncogenic pathway clustering

Uniform manifold approximation and projection (UMAP) visualization of gene expression variation across brain tumors (Figure 5A) showed expected histological clustering of brain tumors. We further observed that, except for three outliers, *C11orf95::*

*RELA* (*ZFTA::RELA*) fusion-positive ependymomas fell within distinct clusters (Figure S6A). MB tumors clustered by molecular subtype, with WNT and SHH in distinct clusters and groups 3 and 4 showing some expected overlap (Figure S6B). Notably, two MB tumors annotated as SHH did not cluster with the other MB tumors, and one clustered with group 3/4 tumors, suggesting potential subtype misclassification or different underlying biology of these two tumors. *BRAF*-driven LGGs (Figure S6C) fell into three separate clusters, suggesting additional shared biology within each cluster. Histone H3 G35-mutant HGGs generally clustered together and away from K28-mutant tumors (Figure S6D). Interestingly, although H3 K28-mutant and H3 WT tumors have different biological drivers,[54] they did not form distinct clusters. This pattern suggests that these subtypes may be driven by common transcriptional programs or have other, much stronger biological drivers than their known distinct epigenetic drivers or that we lack power to detect transcriptional differences.

We performed GSVA for Hallmark cancer gene sets (Figure 5B) and quantified immune cell fractions using quanTIseq (Figures 5C and S6E), results from which recapitulated previously described tumor biology. For example, HGG, DMG, MB, and ATRT tumors are known to upregulate *MYC*,[55] which in turn activates *E2F* and S phase.[56] Indeed, we detected significant (Bonferroni-corrected p < 0.05) upregulation of *MYC* and *E2F* targets, as well as G2M (cell cycle phase following S phase) in MBs, ATRTs, and HGGs compared with several other cancer groups. In contrast, LGGs showed significant downregulation (Bonferroni-corrected p < 0.05, multiple cancer group comparisons) of these pathways. Schwannomas and neurofibromas, which have an inflammatory immune microenvironment of T and B lymphocytes and tumor-associated macrophages (TAMs), are driven by upregulation of cytokines such as interferon γ (IFNγ), interleukin-1 (IL-1), IL-6, and tumor necrosis factor α (TNF-α).[57] GSVA revealed significant upregulation of these cytokines in hallmark pathways (Bonferroni-corrected p < 0.05, multiple cancer group comparisons) (Figure 5B), and monocytes dominated these tumors' immune cell repertoire (Figure 5C). We also observed significant upregulation of pro-inflammatory cytokines IFNα and IFNγ in both LGGs and craniopharyngiomas when compared with either MBs or ependymomas (Bonferroni-corrected p < 0.05) (Figure 5B). Together, these results support previous proteogenomic findings that aggressive MBs and ependymomas have lower immune infiltration compared with *BRAF*-driven LGGs and craniopharyngiomas.[58]

Although CD8[+] T cell infiltration across all cancer groups was minimal (Figure 5C), we observed a signal in specific cancer molecular subtypes (group 3 and 4 MBs) as well as outlier tumors (*BRAF*-driven LGG, *BRAF*-driven and WT GNG, and CNS embryonal tumors, not otherwise specified (NOS); Figure S6E). Surprisingly, the classically immunologically cold HGGs and DMGs[59,60] contained higher overall fractions of immune cells, primarily monocytes, dendritic cells, and natural killer (NK) cells (Figure 5C). Thus, quanTIseq might have actually captured microglia within these immune cell fractions.

While we did not detect notable prognostic effects of immune cell infiltration on overall survival in HGGs or DMGs, we found that high levels of macrophage M1 and monocytes were

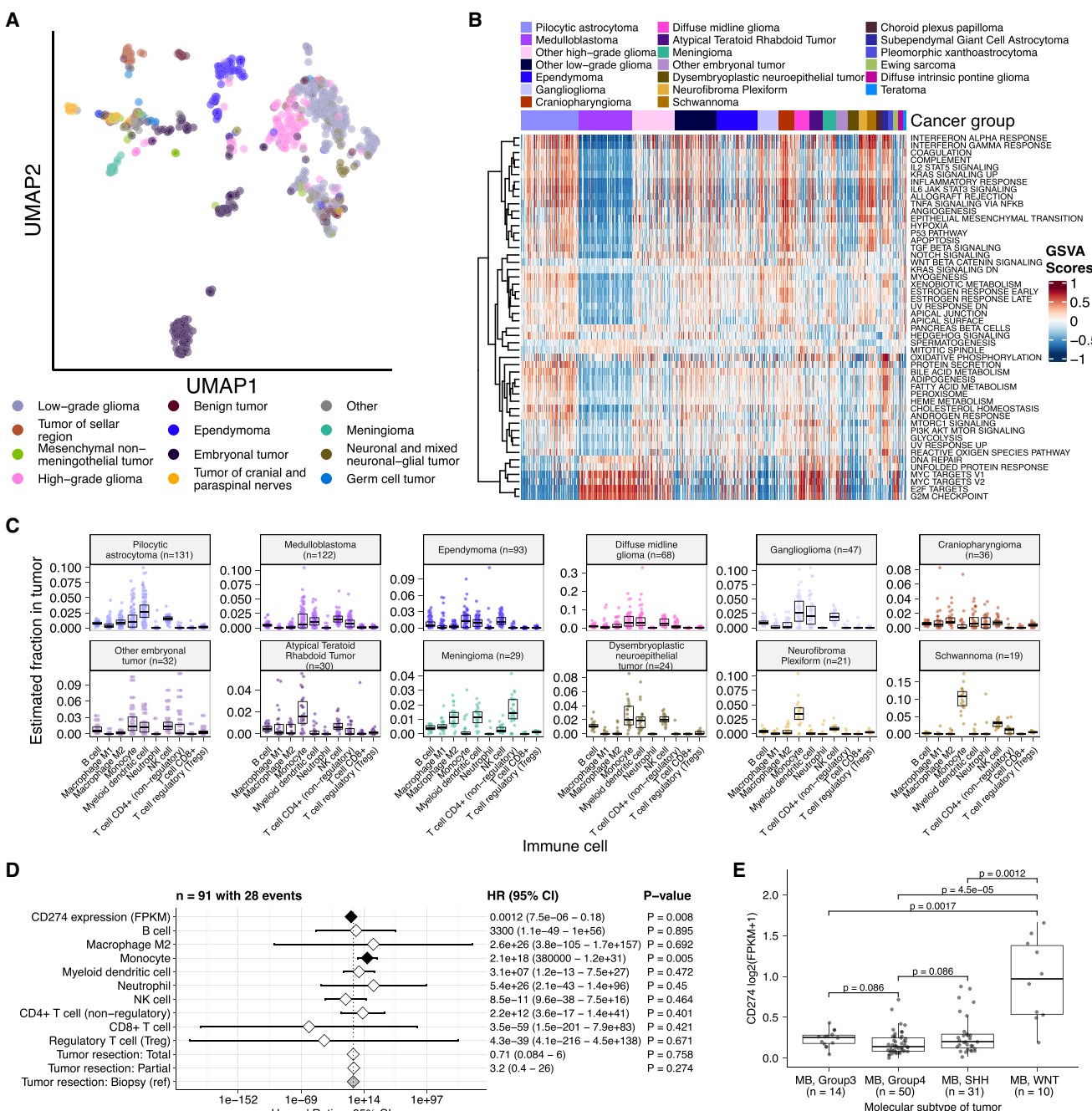

**Figure 5. Transcriptomic and immune landscape of pediatric brain tumors**

(A) First two dimensions of transcriptome data UMAP, with points colored by broad histology.

(B) Heatmap of GSVA scores for Hallmark gene sets with tumors ordered by cancer group.

(C) Boxplots of quanTIseq estimates of immune cell proportions in cancer groups with n >15 tumors. Note: other HGGs and other LGGs have immune cell proportions similar to DMG and pilocytic astrocytoma, respectively, and are not shown.

(D) Forest plot depicting additive effects of *CD274* expression, immune cell proportion, and extent of tumor resection on OS of medulloblastoma patients. HRs with 95% confidence intervals and p values (multivariate Cox) are listed. Black diamonds denote significant p values, and gray diamonds denote reference groups. Note: the macrophage M1 HR was 0 (coefficient = −9.90e4) with infinite upper and lower confidence intervals (CIs) and thus was not included in the figure.

(E) Boxplot of *CD274* expression (log2 FPKM) for medulloblastomas grouped by subtype. Bonferroni-corrected p values from Wilcoxon tests are shown. Boxplot represents 5% (lower whisker), 25% (lower box), 50% (median), 75% (upper box), and 95% (upper whisker) quantiles. Only stranded RNA-seq data are plotted.

associated with poorer overall survival (monocyte HR = 2.1e18, 95% CI = 3.80e5 to 1.2e31, p = 0.005, multivariate Cox) in MBs (Figure 5D). We further reproduced previous findings (Figure 5E) that MBs typically have low expression of *CD274* (PD-L1).[61] We also found that higher expression of *CD274* was significantly associated with improved overall prognosis for MB tumors, although marginal (HR = 0.0012, 95% CI = 7.5e−06 to 0.18, p = 0.008, multivariate Cox) (Figure 5D). This result may be explained by the higher expression of *CD274* observed in WNT subtype tumors by us and others,[62] as this diagnosis carries the best prognosis of all MB subgroups (Figure 5E).

We additionally explored the ratio of CD8[+] to CD4[+] T cells across tumor subtypes. This ratio has been associated with better immunotherapy response and prognosis following PD-L1 inhibition in non-small cell lung cancer or adoptive T cell therapy in multiple stage III or IV cancers.[63,64] While adamantinomatous craniopharyngiomas and group 3 and 4 MBs had the highest ratios (Figure S6F), very few tumors had ratios greater than 1, highlighting an urgent need to identify novel therapeutics for pediatric brain tumors with poor prognosis.

Finally, we explored the potential influence of tumor purity by repeating selected transcriptomic analyses restricted to only samples with high tumor purity (see STAR Methods). Results from these analyses were broadly consistent (Figures S7D–S7I) with results derived from all stranded RNA-seq samples.

## DISCUSSION

The CBTN released the PBTA raw genomic data in September 2018 without embargo, allowing researchers immediate access to begin making discoveries on behalf of children with CNS tumors everywhere. Since this publication, the CBTN has approved over 200 data research projects[4] from 69 different institutions, with 60% from non-CBTN sites. We created the OpenPBTA as an open, real-time, reproducible analysis framework to genomically characterize pediatric brain tumors, bringing together basic and translational researchers, clinicians, and data scientists. We provide reusable code and data resources, paired with cloud-based availability of source and derived data resources, to the pediatric oncology community, encouraging interdisciplinary collaboration. To our knowledge, this initiative represents the first large-scale, collaborative, open analysis of genomic data coupled with open manuscript writing, wherein we comprehensively analyzed the PBTA cohort. Using available WGS, WXS, and RNA-seq data, we generated high-confidence consensus SNV and CNV calls, prioritized putative oncogenic fusions, and established over 40 scalable and rigorously reviewed modules to perform common downstream cancer genomics analyses. We detected expected patterns of genomic lesions, mutational signatures, and aberrantly regulated signaling pathways across multiple pediatric brain tumor histologies.

Assembling large, pan-histology cohorts of fresh frozen samples and associated clinical phenotypes and outcomes requires a multiyear, multiinstitutional framework, like those provided by CBTN and PNOC. As such, uniform clinical molecular subtyping was largely not performed for this cohort at the time of sample collection. Since DNA methylation data for these samples were not yet available to classify molecular subtypes, we created

RNA- and DNA-based subtyping modules aligned with molecularly defined diagnoses. We worked closely with pathologists and clinicians to assign research-grade integrated diagnoses for 60% of tumors while discovering incorrectly diagnosed or misidentified samples in the OpenPBTA cohort. For example, we subtyped MB tumors, of which only 35% (43/122) had prior subtype information from pathology reports, using *MMS2* or *MedulloClassifier*[22,23] and subsequently applied the consensus of these methods to subtype all MBs.

We advanced the integrative analyses and cross-cohort comparison via a number of validated modules. We used an expression classifier to determine whether tumors have dysfunctional *TP53*[42] and the EXTEND algorithm to determine their degree of telomerase activity using a 13-gene signature.[44] Interestingly, we found that hypermutant HGGs universally displayed *TP53* dysregulation, unlike adult cancers like colorectal cancer and gastric adenocarcinoma, where *TP53* dysregulation in hypermutated tumors is less common.[65,66] Furthermore, high *TP53* scores were a significant prognostic marker for poor overall survival for patients with tumor types including H3 K28-mutant DMGs and ependymomas. We also show that EXTEND scores are a robust surrogate measure for telomerase activity in pediatric brain tumors. By assessing *TP53* and telomerase activity prospectively from expression data, information usually only attainable with DNA sequencing and/or qPCR, we incorporated oncogenic biomarker and prognostic knowledge, thereby expanding our biological understanding of these tumors.

We identified enrichment of hallmark cancer pathways and characterized the immune cell landscape across pediatric brain tumors, demonstrating that tumors in some histologies, such as schwannomas, craniopharyngiomas, and LGGs, may have an inflammatory tumor microenvironment. Notably, we observed upregulation of IFNγ, IL-1, IL-6, and TNFα in craniopharyngiomas, tumors difficult to resect due to their anatomical location and critical surrounding structures. Neurotoxic side effects have been reported in response to IFNα immunotherapy,[67,68] leading researchers to propose additional immune vulnerabilities, such as IL-6 inhibition and immune checkpoint blockade, as cystic adamantinomatous craniopharyngiomas therapies.[69–73] Our results support this endeavor. Finally, we reproduced the overall known poor infiltration of CD8[+] T cells and general low expression of *CD274* (PD-L1) in pediatric brain tumors, highlighting that we urgently need novel therapeutic strategies for tumors unlikely to respond to immune checkpoint blockade therapy.

While large-scale collaborative efforts may take a longer time to complete, adoption an open science framework substantially mitigated this concern. By maintaining all data, analytical code, and results in public repositories, we ensured that such logistics did not hinder progress in pediatric cancer research. Indeed, the OpenPBTA is already a foundational data analysis and processing layer for several discovery research and translational projects and will continue to add other genomic modalities and analyses, including germline, epigenomic, single-cell, splicing, imaging, and model drug response data. For example, the OpenPBTA RNA fusion filtering module led to the development of the R package *annoFuse*[74] and an R Shiny application *shinyFuse*. Leveraging OpenPBTA's MB subtyping and immune deconvolution analyses,

Dang and colleagues showed that SHH tumors are enriched with monocyte and microglia-derived macrophages, which may accumulate following radiation therapy.[9] Expression and CNV analyses demonstrated that *GPC2* is a highly expressed and copy-number-gained immunotherapeutic target in ETMRs, MBs, choroid plexus carcinomas, H3 WT HGGs, and DMGs. Foster and colleagues therefore developed a chimeric antigen receptor (CAR) directed against *GPC2*, which shows preclinical efficacy in mouse models.[11] Another study harnessed the OpenPBTA to integrate germline variants, discovering that pediatric patients with HGG with alternative telomere lengthening are enriched for pathogenic or likely pathogenic germline variants in the MMR pathway, possess oncogenic *ATRX* mutations, and have increased TMB.[12] Moreover, the OpenPBTA has enabled a framework to support real-time integration of clinical trial subjects as they enrolled in the PNOC008 HGG clinical trial[75] or the PNOC027 MB clinical trial,[76] allowing researchers and clinicians to link tumor biology to translational impact through clinical decision support during tumor board discussions. Finally, as part of the NCI's CCDI, the OpenPBTA was recently expanded into OpenPedCan, a pan-pediatric cancer effort (https://github.com/PediatricOpenTargets/OpenPedCan-analysis) that enabled the creation of the pediatric Molecular Targets Platform (https://moleculartargets.ccdi.cancer.gov/) in support of the RACE Act. An additional, large-scale cohort of >1,500 tumor samples and associated germline DNA is undergoing harmonization as part of CBTN CCDI-Kids First NCI and Common Fund project (https://commonfund.nih.gov/kidsfirst/2021X01projects#FY21_Resnick) and will be immediately integrated with OpenPBTA data through OpenPedCan. The OpenPBTA has paved the way for new modes of collaborative data-driven discovery using open, reproducible, and scalable analyses that will continue to grow over time. We anticipate that this foundational work will have an ongoing, long-term impact for pediatric oncology researchers, ultimately accelerating translation and leading to improved outcomes for children with cancer.

All code and processed data are openly available through GitHub, CAVATICA, Zenodo, and PedcBioPortal (see STAR Methods).

### Limitations of the study

Notably, PBTA brain tumor samples were collected over decades, and RNA samples were prepared using two distinct library preparations (stranded or poly-A; Figure S7A) by multiple sequencing centers. While we noted a strong library preparation batch effect (Figure S7B) and a possible sequencing center batch effect (Figure S7C), cancer groups are highly unbalanced across library preparations (Figure S7A). We did not perform batch correction because removing batch effects across unbalanced groups may induce false differences among groups.[77,78] Instead, we circumvent batch effects by grouping only stranded RNA-seq expression data, which comprise the vast majority of the PBTA cohort, for the transcriptomic analyses presented in Figures 4 and 5. As the batch correction strategy depends highly on research goals,[78] we provide library preparation-specific expression matrices in the OpenPBTA data release for others to adapt to their needs. A second potential limitation is that performing analyses with all samples, rather than samples with high

tumor purity, might result in loss of information, such as subclonal variants or low-level oncogenic pathway expression. To this end, we reperformed transcriptomic analyses using only samples with high tumor purity (see STAR Methods for details), and indeed, results were broadly consistent with those derived from the full cohort (Figures S7D–S7I). To enable more robust statistical analysis and presentation of results, we randomly selected one independent specimen from patients with duplicate sequenced samples per tumor event rather than combining the data. This practice did not induce notable differences if the selected specimen changed over time, e.g., with a new data release. Finally, because this initial PBTA cohort mostly contains samples collected at diagnosis from one tumor section/punch, we could not reliably perform systematic intratumoral and/or longitudinal analyses, though we expect nearly 100 paired longitudinal tumors from the NIH X01 CA267587-01 pediatric brain tumor cohort to be released through OpenPedCan for future exploration.

### CONSORTIA

The past and present members of the Children's Brain Tumor Network who contributed to the generation of specimens and data are Adam C. Resnick, Alexa Plisiewicz, Allison M. Morgan, Allison P. Heath, Alyssa Paul, Amanda Saratsis, Amy Smith, Ana Aguilar, Ana Guerreiro Stücklin, Anastasia Arynchyna, Andrea Franson, Angela J. Waanders, Angela N. Viaene, Anita Nirenberg, Anna Maria Buccoliero, Anna Yaffe, Anny Shai, Anthony Bet, Antoinette Price, Arlene Luther, Ashley Plant, Augustine Eze, Bailey K. Farrow, Baoli Hu, Beth Frenkel, Bo Zhang, Bobby Moulder, Bonnie Cole, Brian M. Ennis, Brian R. Rood, Brittany Lebert, Carina A. Leonard, Carl Koschmann, Caroline Caudill, Caroline Drinkwater, Cassie N. Kline, Catherine Sullivan, Chanel Keoni, Chiara Caporalini, Christine Bobick-Butcher, Christopher Mason, Chunde Li, Claire Carter, Claudia MaduroCoronado, Clayton Wiley, Cynthia Wong, David E. Kram, David Haussler, David Kram, David Pisapia, David Ziegler, Denise Morinigo, Derek Hanson, Donald W. Parsons, Elizabeth Appert, Emily Drake, Emily Golbeck, Ena Agbodza, Eric H. Raabe, Eric M. Jackson, Erin Alexander, Esteban Uceda, Eugene Hwang, Fausto Rodriguez, Gabrielle S. Stone, Gary Kohanbash, Gavriella Silverman, George Rafidi, Gerald Grant, Gerri Trooskin, Gilad Evrony, Graham Keyes, Hagop Boyajian, Holly B. Lindsay, Holly C. Beale, Ian F. Pollack, James Johnston, James Palmer, Jane Minturn, Jared Pisapia, Jason E. Cain, Jason R. Fangusaro, Javad Nazarian, Jeanette Haugh, Jeff Stevens, Jeffrey P. Greenfield, Jeffrey Rubens, Jena V. Lilly, Jennifer L. Mason, Jessica B. Foster, Jim Olson, Jo Lynne Rokita, Joanna J. Phillips, Jonathan Waller, Josh Rubin, Judy E. Palma, Justin McCroskey, Justine Rizzo, Kaitlin Lehmann, Kamnaa Arya, Karlene Hall, Katherine Pehlivan, Kenneth Seidl, Kimberly Diamond, Kristen Harnett, Kristina A. Cole, Krutika S. Gaonkar, Lamiya Tauhid, Laura Prolo, Leah Holloway, Leslie Brosig, Lina Lopez, Lionel Chow, Madhuri Kambhampati, Mahdi Sarmady, Margaret Nevins, Mari Groves, Mariarita Santi-Vicini, Marilyn M. Li, Marion Mateos, Mateusz Koptyra, Matija Snuderl, Matthew Miller, Matthew Sklar, Matthew Wood, Meghan Connors, Melissa Williams, Meredith Egan, Michael Fisher, Michael Koldobskiy, Michelle Monje,

Migdalia Martinez, Miguel A. Brown, Mike Prados, Miriam Bornhorst, Mirko Scagnet, Mohamed AbdelBaki, Monique Carrero-Tagle, Nadia Dahmane, Nalin Gupta, Nathan Young, Nicholas A. Vitanza, Nicholas Tassone, Nicholas Van Kuren, Nicolas Gerber, Nithin D. Adappa, Nitin Wadhwani, Noel Coleman, Obi Obayashi, Olena M. Vaske, Olivier Elemento, Oren Becher, Philbert Oliveros, Phillip B. Storm, Pichai Raman, Prajwal Rajappa, Rintaro Hashizume, Rishi R. Lulla, Robert Keating, Robert M. Lober, Ron Firestein, Sabine Mueller, Sameer Agnihotri, Samuel G. Winebrake, Samuel Rivero-Hinojosa, Sarah Diane Black, Sarah Leary, Schuyler Stoller, Shannon Robins, Sharon Gardner, Shelly Wang, Sherri Mayans, Sherry Tutson, Shida Zhu, Sofie R. Salama, Sonia Partap, Sonika Dahiya, Sriram Venneti, Stacie Stapleton, Stephani Campion, Stephanie Stefankiewicz, Stewart Goldman, Swetha Thambireddy, Tatiana S. Patton, Teresa Hidalgo, Theo Nicolaides, Thinh Q. Nguyen, Thomas W. McLean, Tiffany Walker, Toba Niazi, Tobey MacDonald, Valeria Lopez-Gil, Valerie Baubet, Whitney Rife, Xiao-Nan Li, Ximena Cuellar, Yiran Guo, Yuankun Zhu, and Zeinab Helil.

The past and present members of the Pacific Pediatric Neuro-Oncology Consortium who contributed to the generation of specimens and data are Adam C. Resnick, Alicia Lenzen, Alyssa Reddy, Amar Gajjar, Ana Guerreiro Stucklin, Anat Epstein, Andrea Franson, Angela Waanders, Anne Bendel, Anu Banerjee, Ashley Margol, Ashley Plant, Brian Rood, Carl Koschmann, Carol Bruggers, Caroline Hastings, Cassie N. Kline, Christina Coleman Abadi, Christopher Tinkle, Corey Raffel, Dan Runco, Daniel Landi, Daphne Adele Haas-Kogan, David Ashley, David Ziegler, Derek Hanson, Dong Anh Khuong Quang, Duane Mitchell, Elias Sayour, Eric Jackson, Eric Raabe, Eugene Hwang, Fatema Malbari, Geoffrey McCowage, Girish Dhall, Gregory Friedman, Hideho Okada, Ibrahim Qaddoumi, Iris Fried, Jae Cho, Jane Minturn, Jason Blatt, Javad Nazarian, Jeffrey Rubens, Jena V. Lilly, Jennifer Elster, Jennifer L. Mason, Jessica Schulte, Jonathan Schoenfeld, Josh Rubin, Karen Gauvain, Karen Wright, Katharine Offer, Katie Metrock, Kellie Haworth, Ken Cohen, Kristina A. Cole, Lance Governale, Linda Stork, Lindsay Kilburn, Lissa Baird, Maggie Skrypek, Marcia Leonard, Margaret Shatara, Margot Lazow, Mariella Filbin, Maryam Fouladi, Matthew Miller, Megan Paul, Michael Fisher, Michael Koldobskiy, Michael Prados, Michal Yalon Oren, Mimi Bandopadhayay, Miriam Bornhorst, Mohamed AbdelBaki, Nalin Gupta, Nathan Robison, Nicholas Whipple, Nick Gottardo, Nicholas A. Vitanza, Nicolas Gerber, Patricia Robertson, Payal Jain, Peter Sun, Priya Chan, Richard S Lemons, Robert Wechsler-Reya, Roger Packer, Russ Geyer, Ryan Velasco, Sabine Mueller, Sahaja Acharya, Sam Cheshier, Sarah Leary, Scott Coven, Sebastian M. Waszak, Sharon Gardner, Sri Gururangan, Stewart Goldman, Susan Chi, Tab Cooney, Tatiana S. Patton, Theodore Nicolaides, and Tom Belle Davidson.

## STAR★METHODS

Detailed methods are provided in the online version of this paper and include the following:

- KEY RESOURCES TABLE
- RESOURCE AVAILABILITY
  - ○ Lead contact
  - ○ Materials availability
  - ○ Data and code availability
- EXPERIMENTAL MODEL AND STUDY PARTICIPANT DETAILS
  - ○ Model generation
- METHOD DETAILS
  - ○ Nucleic acids extraction and library preparation
  - ○ Data generation
  - ○ DNA WGS alignment
  - ○ Quality control of sequencing data
  - ○ Germline variant calling
  - ○ Somatic mutation calling
  - ○ Somatic copy number variant calling (WGS samples only)
  - ○ Somatic structural variant calling (WGS samples only)
  - ○ Gene expression
- QUANTIFICATION AND STATISTICAL ANALYSIS
  - ○ Tumor purity (*tumor-purity-exploration* module)
  - ○ Breakpoint density (WGS samples only; *chromosomal-instability* analysis module)
  - ○ Chromothripsis analysis (WGS samples only; *chromothripsis* analysis module)
  - ○ Oncoprint figure generation (*oncoprint-landscape* analysis module)
  - ○ Mutational signatures (*mutational-signatures* analysis module)
  - ○ Tumor mutation burden (*snv-callers* analysis module)
  - ○ Clinical data harmonization
  - ○ TP53 alteration annotation (*tp53_nf1_score* analysis module)
  - ○ Prediction of participants' genetic sex
  - ○ Selection of independent samples (*independent-samples* analysis module)
  - ○ Quantification of telomerase activity using gene expression data (*telomerase-activity-prediction* analysis module)
  - ○ Survival models (*survival-analysis* analysis module)

### SUPPLEMENTAL INFORMATION

### ACKNOWLEDGMENTS

We graciously thank the patients and families who have donated tumors to the CBTN and/or the PNOC, without which this research would not be possible. Philanthropic support has ensured the CBTN's ability to collect, store, manage, and distribute specimens and data. The authors thank the following collaborators who contributed or supervised analyses present in the analysis repository that were not included in the manuscript: William Amadio, Holly C. Beale, Ellen T. Kephart, A. Geoffrey Lyle, and Olena M. Vaske. Finally, we thank Yuanchao Zhang for adding to the project codebase, Jessica B. Foster for helpful discussions while drafting the manuscript, and Gina D. Mawla for identifying and reporting OpenPBTA data issues. The following donors have provided leadership level support for CBTN: CBTN Executive Council members, Brain Tumor Board of Visitors, Children's Brain Tumor Foundation, Easie Family Foundation, Kortney Rose Foundation, Lilabean Foundation, Minnick Family Charitable Fund, Perricelli Family, Psalm 103 Foundation, and Swifty Foundation. This work was funded through the Alex's Lemonade Stand

Foundation (ALSF) Childhood Cancer Data Lab (C.S.G.); the ALSF Young Investigator Award (J.L.R.); ALSF Catalyst Awards (J.L.R., A.C.R., P.B.S., and S.J.S.); the ALSF CCDL Postdoctoral Training Grant (S.M.F.); the Children's Hospital of Philadelphia Division of Neurosurgery (P.B.S. and A.C.R.); the Australian government, Department of Education (A.P.H.); St. Anna Kinderkrebsforschung, Austria (A.R.P.); the Mildred Scheel Early Career Center Dresden P2, funded by the German Cancer Aid (A.R.P.); NIH grants 3P30 CA016520-44S5 (A.C.R.), U2C HL138346-03 (A.C.R., A.P.H.), U24 CA220457-03 (A.C.R.), K12GM081259 (S.M.F.), and R03-CA23036 (S.J.D.); NIH contract nos. HHSN261200800001E (S.J.D.) and 75N91019D00024, task order no. 75N91020F00003 (J.L.R., A.C.R., A.P.H.); and the Intramural Research Program of the Division of Cancer Epidemiology and Genetics of the National Cancer Institute. The content of this publication does not necessarily reflect the views or policies of the Department of Health and Human Services, nor does mention of trade names, commercial products, or organizations imply endorsement by the United States government.

## AUTHOR CONTRIBUTIONS

Conceptualization, P.B.S., P.R., B.R.R., R.R.L., J.N., S.M., M.P., A.J.W., C.J.K., A.C.R., C.S.G., and J.L.R.; methodology, J.A.S., K.S.G., C.L.S., C.J.B., K.S.R., Y.Z., N.D., H.M.X., P.R., A.A.K., S.K., M.A.B., C.S.G., J.L.R., and J.N.T.; software, J.A.S., C.L.S., K.S.G., S.J.S., B.K.F., Y.Y., T.K., N.V.K., E.W., C.S.G., J.L.R., and J.N.T.; validation, J.A.S., S.J.S., C.L.S., C.J.B., M.S., P.J., A.N.V., S.M.F., S.J.D., D.H., R.S.K., and J.N.T.; formal analysis, J.A.S., K.S.G., S.J.S., C.L.S., C.J.B., R.J., K.S.R., Y.Z., L.E.E., D.P.M., Y.Y., T.K., N.N., M.P.K., N.D., P.R., Z.V., X.H., B.Z., A.R.P., B.M.E., S.Z., Y.G., S.K., R.S.K., E.W., M.A.B., J.L.R., and J.N.T.; investigation, J.A.S., K.S.G., S.J.S., C.L.S., C.J.B., K.S.R., Y.Z., N.D., M.S., J.K., P.J., Z.V., C.K., A.N.V., G.P.W., S.J.D., R.S.K., J.L.R., and J.N.T.; writing – original draft, J.A.S., K.S.G., S.J.S., C.L.S., C.J.B., R.J., K.S.R., L.E.E., N.N., M.P.K., S.Z., S.K., J.W., and J.L.R.; writing – review & editing, J.A.S., K.S.G., S.J.S., C.L.S., R.J., M.S., J.K., S.C.M., J.V.L., C.K., G.P.W., A.R.P., S.J.D., S.Z., Y.G., A.S.M., K.A.C., P.J.M., D.R.S., C.S.G., J.L.R., and J.N.T.; visualization, J.A.S., S.J.S., C.L.S., C.J.B., R.J., N.N., S.Z., and J.N.T.; supervision, J.A.S., S.J.S., Y.Z., H.M.X., C.K., S.J.D., S.Z., A.J.W., J.L.M., D.R.S., A.C.R., C.S.G., J.L.R., and J.N.T.; data curation, K.S.G., R.J., Y.Z., B.K.F., S.R., P.J., L.S., B.Z., B.M.E., M.C.K., N.V.K., N.C., M.A.B., J.L.R., and J.N.T.; funding acquisition, S.J.S., P.B.S., J.V.L., A.P.H., S.M.F., A.R.P., S.J.D., A.C.R., C.S.G., and J.L.R.; resources, P.B.S., C.B., and A.C.R.; project administration, J.V.L., A.P.H., C.S.G., and J.N.T. Except for the first and last four authors, authorship order was determined as follows: authors who contributed to the OpenPBTA code base are listed based on the number of modules included in the manuscript to which that individual contributed, and in the case of ties, a random order is used. All remaining authors are then listed in a random order. Code for determining authorship order can be found in the *count-contributions* module of the OpenPBTA analysis repository.

## DECLARATION OF INTERESTS

C.S.G.'s spouse was an employee of Alex's Lemonade Stand Foundation, which was a sponsor of this research. J.A.S., C.L.S., C.J.B., S.J.S., and J.N.T. are or were employees of Alex's Lemonade Stand Foundation, a sponsor of this research. A.J.W. is a member of the Scientific Advisory boards for Alexion and DayOne Biopharmaceuticals.

## INCLUSION AND DIVERSITY

We worked to ensure ethnic or other types of diversity in the recruitment of human subjects. We worked to ensure sex balance in the selection of non-human subjects. We worked to ensure diversity in experimental samples through the selection of the cell lines. We worked to ensure diversity in experimental samples through the selection of the genomic datasets. One or more of the authors of this paper self-identifies as an underrepresented ethnic minority in their field of research or within their geographical location. One or more of the authors of this paper self-identifies as a gender minority in their field of research. One or more of the authors of this paper self-identifies as a member of the LGBTQIA+ community. One or more of the authors of this paper self-identifies as living with a disability. One or more of the authors of this paper received support from a program designed to increase minority representation in their field of research.

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

## Resource

CellPress

156. Crotty, T.B., Scheithauer, B.W., Young, W.F., Davis, D.H., Shaw, E.G., Miller, G.M., and Burger, P.C. (1995). Papillary craniopharyngioma: a clinicopathological study of 48 cases. J. Neurosurg. *83*, 206–214. https://doi.org/10.3171/jns.1995.83.2.0206.

157. Bunin, G.R., Surawicz, T.S., Witman, P.A., Preston-Martin, S., Davis, F., and Bruner, J.M. (1998). The descriptive epidemiology of craniopharyngioma. J. Neurosurg. *89*, 547–551. https://doi.org/10.3171/jns.1998.89.4.0547.

158. Chang, M.T., Bhattarai, T.S., Schram, A.M., Bielski, C.M., Donoghue, M.T.A., Jonsson, P., Chakravarty, D., Phillips, S., Kandoth, C., Penson, A., et al. (2018). Accelerating discovery of functional mutant alleles in cancer. Cancer Discov. *8*, 174–183. https://doi.org/10.1158/2159-8290.cd-17-0321.

159. Chang, M.T., Asthana, S., Gao, S.P., Lee, B.H., Chapman, J.S., Kandoth, C., Gao, J., Socci, N.D., Solit, D.B., Olshen, A.B., et al. (2016). Identifying recurrent mutations in cancer reveals widespread lineage diversity and mutational specificity. Nat. Biotechnol. *34*, 155–163. https://doi.org/10.1038/nbt.3391.

160. Harms, K.L., and Chen, X. (2006). The functional domains in p53 family proteins exhibit both common and distinct properties. Cell Death Differ. *13*, 890–897. https://doi.org/10.1038/sj.cdd.4401904.

161. Guha, T., and Malkin, D. (2017). Inherited TP53 mutations and the Li–Fraumeni syndrome. Cold Spring Harb. Perspect. Med. *7*, a026187. https://doi.org/10.1101/cshperspect.a026187.

162. Li, H., Handsaker, B., Wysoker, A., Fennell, T., Ruan, J., Homer, N., Marth, G., Abecasis, G., and Durbin, R.; 1000 Genome Project Data Processing Subgroup (2009). The sequence alignment/map format and SAMtools. Bioinformatics *25*, 2078–2079. https://doi.org/10.1093/bioinformatics/btp352.

163. Kaplan, E.L., and Meier, P. (1958). Nonparametric estimation from incomplete observations. J. Am. Stat. Assoc. *53*, 457–481. https://doi.org/10.2307/2281868.

164. Mantel, N. (1966). Evaluation of survival data and two new rank order statistics arising in its consideration. Cancer Chemother. Rep. *50*, 163–170.

165. Cox, D.R. (1972). Regression models and life-tables. J. Roy. Stat. Soc. B *34*, 187–202. https://doi.org/10.1111/j.2517-6161.1972.tb00899.x.

## STAR★METHODS

### KEY RESOURCES TABLE

| REAGENT or RESOURCE | SOURCE | IDENTIFIER |
|---|---|---|
| **Chemicals, peptides, and recombinant proteins** | | |
| Recover Cell Culture Freezing media | Gibco | Cat# 12648010 |
| Hank's Balanced Salt Solution (HBSS) | Gibco | Cat# 14175095 |
| Papain | SciQuest | Cat# LS003124 |
| Ovomucoid | SciQuest | Cat# 542000 |
| DNase | Roche | Cat# 10104159001 |
| RNase A | Qiagen | Cat# 19101 |
| 100μm cell strainer | Greiner Bio-One | Cat# 542000 |
| DMEM/F-12 medium | Sigma | Cat# D8062 |
| Fetal Bovine Serum (FBS) | Hyclone | Cat# SH30910.03 |
| GlutaMAX | Gibco | Cat# 35050061 |
| Penicillin/Streptomycin-Amphotericin B | Lonza | Cat# 17-745E |
| Normocin | Invivogen | Cat# ant-nr-2 |
| B-27 supplement minus vitamin A | Gibco | Cat# 12587-010 |
| N-2 supplement | Gibco | Cat# 17502001 |
| Epidermal growth factor | Gibco | Cat# PHG0311L |
| Basic fibroblast growth factor | PeproTech | Cat# 100-18B |
| Heparin | Sigma | Cat# H3149 |
| **Critical commercial assays** | | |
| GenePrint 24 STR profiling kit | Promega | Cat# B1870 |
| DNA/RNA AllPrep Kit | Qiagen | Cat# 80204 |
| TruSeq RNA Sample Prep Kit | Illumina | Cat# FC-122-1001 |
| KAPA Library Preparation Kit | Roche | Cat# KK8201 |
| AllPrep DNA/RNA/miRNA Universal kit | Qiagen | Cat# 80224 |
| QIAsymphony DSP DNA Midi Kit | Qiagen | Cat# 937255 |
| KAPA Hyper-Prep kit | Roche | Cat# 08098107702 |
| RiboErase kit | Roche | Cat# 07962304001 |
| **Deposited data** | | |
| Raw and harmonized WGS, WXS, Panel, RNA-Seq | KidsFirst Data Resource Center, This project | Open Pediatric Brain Tumor Atlas et al.[79] |
| Merged summary files | This project | https://cavatica.sbgenomics.com/u/cavatica/openpbta |
| Merged summary files and downstream analyses | This project | https://github.com/AlexsLemonade/OpenPBTA-analysis[80] |
| Processed data | This project | https://pedcbioportal.kidsfirstdrc.org/study/summary?id=openpbta |
| Data underlying figures and molecular alterations | This project | Shapiro et al.[81] |
| **Experimental models: Cell lines** | | |
| CBTN pediatric brain tumor-derived cell lines | Ijaz et al.[13] | See Table S1 for identifiers |
| **Software and algorithms** | | |
| Data processing and analysis software | Multiple | See Table S5 for identifiers |
| OpenPBTA workflows repository | This project | https://github.com/d3b-center/OpenPBTA-workflows[82] |
| OpenPBTA analysis repository | This project | https://github.com/AlexsLemonade/OpenPBTA-analysis[80] |
| OpenPBTA manuscript repository | This project | https://github.com/AlexsLemonade/OpenPBTA-manuscript |
| **Other** | | |
| TCGA WXS dataset | NIH The Cancer Genome Atlas (TCGA) | dbGAP: phs000178.v11.p8 |
| Cancer hotspots | MSKCC | https://www.cancerhotspots.org/#/download (v2) |

*(Continued on next page)*

**Continued**

| REAGENT or RESOURCE | SOURCE | IDENTIFIER |
|---|---|---|
| Reference genomes | Broad Institute | https://s3.console.aws.amazon.com/s3/buckets/broad-references/hg38/v0/ |
| Reference genome hg38, patch release 12 | UCSC | http://hgdownload.soe.ucsc.edu/goldenPath/hg38/bigZips/ |
| Human Cytoband file | UCSC | http://hgdownload.cse.ucsc.edu/goldenpath/hg38/database/cytoBand.txt.gz |
| CDS from GENCODE v27 annotation | GENCODE | https://www.gencodegenes.org/human/release_27.html |
| PFAM domains and locations | UCSC | http://hgdownload.soe.ucsc.edu/goldenPath/hg38/database/pfamDesc.txt.gz; https://pfam.xfam.org/family/PF07714 |
| BSgenome.Hsapiens.UCSC.hg38 annotations | Bioconductor | https://bioconductor.org/packages/release/data/annotation/html/BSgenome.Hsapiens.UCSC.hg38.html |
| gnomAD v2.1.1 (exome and genome) | Genome Aggregation Database | https://gnomad.broadinstitute.org/downloads#v2-liftover-variants |
| KEGG MMR gene set v7.5.1 | Broad Institute | https://www.gsea-msigdb.org/gsea/msigdb/download_geneset.jsp?geneSetName=KEGG_MISMATCH_REPAIR |
| ClinVar Database (2022-05-07) | NCBI | https://ftp.ncbi.nlm.nih.gov/pub/clinvar/vcf_GRCh38/archive_2.0/2022/clinvar_20220507.vcf.gz |

## RESOURCE AVAILABILITY

### Lead contact
Requests for access to OpenPBTA raw data and/or specimens may be directed to and will be fulfilled by Jo Lynne Rokita (rokita@chop.edu).

### Materials availability
This study did not create new, unique reagents.

### Data and code availability
- Raw and harmonized WGS, WXS, and RNA-Seq data derived from human samples are available within the KidsFirst Portal[79] upon access request to the CBTN (https://cbtn.org/) as of the date of the publication. In addition, merged summary files are openly accessible at https://cavatica.sbgenomics.com/u/cavatica/openpbta or via download script in the https://github.com/AlexsLemonade/OpenPBTA-analysis repository. Summary data are visible within PedcBioPortal at https://pedcbioportal.kidsfirstdrc.org/study/summary?id=openpbta. Associated DOIs are listed in the key resources table. Data underlying manuscript figures are available on Zenodo.[81]
- All original code was developed within the following repositories and is publicly available as follows. Primary data analyses can be found at https://github.com/d3b-center/OpenPBTA-workflows. Downstream data analyses can be found at https://github.com/AlexsLemonade/OpenPBTA-analysis. Manuscript code can be found at https://github.com/AlexsLemonade/OpenPBTA-manuscript. Associated DOIs are listed in the key resources table. Software versions are documented in Table S5 as an appendix to the key resources table.
- Any additional information required to reanalyze the data reported in this paper is available from the lead contact upon request.
- Data releases: We maintained a data release folder on Amazon S3, downloadable directly from S3 or our open-access CAVATICA project, with merged files for each analysis (See data and code availability section). As we produced new results (e.g., tumor mutation burden calculations) that we expected to be used across multiple analyses, or identified data issues, we created new data releases in a versioned manner. We reran all manuscript-specific analysis modules with the latest data release (v23) prior to submission and subsequently created a GitHub repository-tagged release to ensure reproducibility.

## EXPERIMENTAL MODEL AND STUDY PARTICIPANT DETAILS

The Pediatric Brain Tumor Atlas specimens are comprised of samples from Children's Brain Tumor Network (CBTN)[4] and the Pediatric Pacific Neuro-Oncology Consortium (PNOC). The CBTN is a collaborative, multi-institutional (32 institutions worldwide)

research program dedicated to the study of childhood brain tumors. PNOC is an international consortium dedicated to bringing new therapies to children and young adults with brain tumors. We also include blood and tumor biospecimens from newly-diagnosed diffuse intrinsic pontine glioma (DIPG) patients as part of the PNOC003 clinical trial PNOC003/NCT02274987.[14]

### Model generation

Previously, CBTN-generated cell lines were derived from either fresh tumor tissue directly obtained from surgery performed at Children's Hospital of Philadelphia (CHOP) or from prospectively collected tumor specimens stored in Recover Cell Culture Freezing medium (cat# 12648010, Gibco). Tumor tissue was dissociated using enzymatic method with papain as described.[13] Briefly, we washed tissue with HBSS (cat# 14175095, Gibco), and tissue was minced and incubated with activated papain solution (cat# LS003124, SciQuest) for up to 45 min. Ovomucoid solution (cat# 542000, SciQuest) was used to inactivate the papain, tissue was briefly treated tissue with DNase (cat# 10104159001, Roche) and passed through a 100μm cell strainer (cat# 542000, Greiner Bio-One). Two cell culture conditions were initiated based on the number of cells available. For cultures utilizing the fetal bovine serum (FBS), cells were plated a minimum density of 3 × 105 cells/mL in DMEM/F-12 medium (cat# D8062, Sigma) supplemented with 20% FBS (cat# SH30910.03, Hyclone), 1% GlutaMAX (cat# 35050061, Gibco), Penicillin/Streptomycin-Amphotericin B Mixture (cat# 17-745E, Lonza), and 0.2% Normocin (cat# ant-nr-2, Invivogen). For serum-free media conditions, cells were plated at minimum density of 1 × 106 cells/mL in DMEM/F12 medium supplemented with 1% GlutaMAX, 1X B-27 supplement minus vitamin A (cat# 12587-010, Gibco), 1x N-2 supplement (cat# 17502001, Gibco), 20 ng/mL epidermal growth factor (cat# PHG0311L, Gibco), 20 ng/mL basic fibroblast growth factor (cat# 100-18B, PeproTech), 2.5 μg/mL heparin (cat# H3149, Sigma), Penicillin/Streptomycin-Amphotericin B Mixture, and 0.2% Normocin. All cell lines used for nucleic acid extraction were confirmed to be mycoplasma-free. Guardian Forensic Sciences performed GenePrint 24 (cat# B1870, Promega), short tandem repeat (STR) analysis on cell line extracted DNA to both confirm identity and that they were free of cross-contamination. Additionally, we performed *NGSCheckMate*[83] on matched DNA and RNA cell line (tumor) and peripheral blood (normal) CRAM files to further confirm identity.

## METHOD DETAILS

### Nucleic acids extraction and library preparation
*PNOC samples*
The Translational Genomic Research Institute (TGEN; Phoenix, AZ) performed DNA and RNA extractions on tumor biopsies using a DNA/RNA AllPrep Kit (Qiagen, #80204). All RNA used for library prep had a minimum RIN of seven, but no QC thresholds were implemented for the DNA. For library preparation, 500 ng of nucleic acids were used as input for RNA-Seq, WXS, and targeted DNA panel (panel) sequencing. RNA library preparation was performed using the TruSeq RNA Sample Prep Kit (Illumina, #FC-122-1001) with poly-A selection, and the exome prep was performed using KAPA Library Preparation Kit (Roche, #KK8201) using Agilent's SureSelect Human All Exon V5 backbone with custom probes. The targeted DNA panel developed by Ashion Analytics (formerly known as the GEM Cancer panel) consisted of exonic probes against 541 cancer genes. Both panel and WXS assays contained 44,000 probes across evenly spaced genomic loci used for genome-wide copy number analysis. For the panel, additional probes tiled across intronic regions of 22 known tumor suppressor genes and 22 genes involved in common cancer translocations for structural analysis. All extractions and library preparations were performed according to manufacturer's instructions.
*CBTN samples*
Blood, tissue, and cell line DNA/RNA extractions were performed at the Biorepository Core at CHOP. Briefly, 10–20 mg frozen tissue, 0.4-1mL of blood, or 2e6 cells pellet was used for extractions. Tissues were lysed using a Qiagen TissueLyser II (Qiagen) with 2 × 30 s at 18Hz settings using 5 mm steel beads (cat# 69989, Qiagen). Both tissue and cell pellets processes included a CHCl3 extraction and were run on the QIACube automated platform (Qiagen) using the AllPrep DNA/RNA/miRNA Universal kit (cat# 80224, Qiagen). Blood was thawed and treated with RNase A (cat#, 19101, Qiagen); 0.4-1mL was processed using the Qiagen QIAsymphony automated platform (Qiagen) using the QIAsymphony DSP DNA Midi Kit (cat# 937255, Qiagen). DNA and RNA quantity and quality was assessed by PerkinElmer DropletQuant UV-VIS spectrophotometer (PerkinElmer) and an Agilent 4200 TapeStation (Agilent, USA) for RIN and DIN (RNA Integrity Number and DNA Integrity Number, respectively). The NantHealth Sequencing Center, BGI at CHOP, or the Genomic Clinical Core at Sidra Medical and Research Center performed library preparation and sequencing. BGI at CHOP and Sidra Medical and Research Center used in house, center-specific workflows for sample preparation. At NantHealth Sequencing Center, DNA sequencing libraries were prepared for tumor and matched-normal DNA using the KAPA Hyper-Prep kit (cat# 08098107702, Roche), and tumor RNA-Seq libraries were prepared using KAPA Stranded RNA-Seq with RiboErase kit (cat# 07962304001, Roche).

### Data generation

NantHealth and Sidra performed 2x150 bp WGS on paired tumor (∼60X) and constitutive DNA (∼30X) samples on an Illumina X/400. BGI at CHOP performed 2x100 bp WGS sequenced at 60X depth for both tumor and normal samples. NantHealth performed ribosomal-depleted whole transcriptome stranded RNA-Seq to an average depth of 200M. BGI at CHOP performed poly-A or ribosomal-depleted whole transcriptome stranded RNA-Seq to an average depth of 100M. The Translational Genomic Research Institute (TGEN; Phoenix, AZ) performed paired tumor (∼200X) and constitutive whole exome sequencing (WXS) or targeted DNA panel (panel)

and poly-A selected RNA-Seq (~200M reads) for PNOC tumor samples. The panel tumor sample was sequenced to 470X, and the normal panel sample was sequenced to 308X. PNOC 2x100 bp WXS and RNA-Seq libraries were sequenced on an Illumina HiSeq 2500.

### DNA WGS alignment

We used *BWA-MEM*[84] to align paired-end DNA-seq reads to the version 38 patch release 12 of the *Homo sapiens* genome reference, obtained as a FASTA file from UCSC (see key resources table). Next, we used the Broad Institute's Best Practices[85] to process Binary Alignment/Map files (BAMs) in preparation for variant discovery. We marked duplicates using *SAMBLASTER*,[86] and we merged and sorted BAMs using *Sambamba*[87] We used the *BaseRecalibrator* submodule of the Broad's Genome Analysis Tool Kit *GATK*[88] to process BAM files. Lastly, for normal/germline input, we used the *GATK HaplotypeCaller*[89] submodule on the recalibrated BAM to generate a genomic variant call format (GVCF) file. This file is used as the basis for germline calling, described in the SNP calling for B-allele frequency (BAF) generation section.

We obtained references from the Broad Genome References on AWS bucket with a general description of references at https://s3. amazonaws.com/broad-references/broad-references-readme.html.

### Quality control of sequencing data

To confirm sample matches and remove mis-matched samples from the dataset, we performed *NGSCheckMate*[83] on matched tumor/normal CRAM files. Briefly, we processed CRAMs using *BCFtools* to filter and call 20k common single nucleotide polymorphisms (SNPs) using default parameters. We used the resulting VCFs to run *NGSCheckMate*. Per *NGSCheckMate* author recommendations, we used ≤ 0.61 as a correlation coefficient cutoff at sequencing depths >10 to predict mis-matched samples. We determined RNA-Seq read strandedness by running the *infer_experiment.py* script from *RNA-SeQC*[90] on the first 200k mapped reads. We removed any samples whose calculated strandedness did not match strandedness information provided by the sequencing center. We required that at least 60% of RNA-Seq reads mapped to the human reference for samples to be included in analysis. During OpenPBTA analysis, we identified some samples which were mis-identified or potentially swapped. Through collaborative analyses and pathology review, these samples were removed from our data releases and from the Kids First portal. Sample removal and associated justifications were documented in the OpenPBTA data release notes.

### Germline variant calling

#### SNP calling for B-allele frequency (BAF) generation

We performed germline haplotype calls using the *GATK* Joint Genotyping Workflow on individual GVCFs from the normal sample alignment workflow. Using only SNPs, we applied the *GATK* generic hard filter suggestions to the VCF, with an additional requirement of 10 reads minimum depth per SNP. We used the filtered VCF as input to *Control-FREEC* and *CNVkit* (below) to generate B-allele frequency (BAF) files. This single-sample workflow is available in the D3b GitHub repository. References can be obtained from the Broad Genome References on AWS bucket, and a general description of references can be found at https://s3.amazonaws.com/broad-references/broad-references-readme.html.

#### Assessment of germline variant pathogenicity

For patients with hypermutant samples, we first added population frequency of germline variants using *ANNOVAR*[91] and pathogenicity scoring from ClinVar[92] using *SnpSift*.[93] We then filtered for variants with read depth ≥ 15, variant allele fraction ≥ 0.20, and which were observed at < 0.1% allele frequency across each population in the Genome Aggregation Database (see key resources table). Finally, we retained variants in genes included in the KEGG MMR gene set (see key resources table), *POLE*, and/or *TP53* which were ClinVar-annotated as pathogenic (P) or likely pathogenic (LP) with review status of ≥ 2 stars. All P/LP variants were manually reviewed by an interdisciplinary team of scientists, clinicians, and genetic counselors. This workflow is available in the D3b GitHub repository.

### Somatic mutation calling

#### SNV and indel calling

We used four variant callers to call SNVs and indels from paired tumor/normal samples with Targeted Panel, WXS, and/or WGS data: *Strelka2*,[94] *Mutect2*,[95] *Lancet*,[96] and *VarDictJava*.[97] *VarDictJava*-only calls were not retained since ~39M calls with low VAF were uniquely called and may be potential false positives. (~1.2M calls were called by *Mutect2*, *Strelka2*, and *Lancet* and included consensus CNV calling as described below.) We used only *Strelka2*, *Mutect2* and *Lancet* to analyze WXS samples from TCGA. TCGA samples were captured using various WXS target capture kits and we downloaded the BED files from the GDC portal. The manufacturers provided the input interval BED files for both panel and WXS data for PBTA samples. We padded all panel and WXS BED files were by 100 bp on each side for *Strelka2*, *Mutect2*, and *VarDictJava* runs and by 400 bp for the *Lancet* run. For WGS calling, we utilized the non-padded BROAD Institute interval calling list wgs_calling_regions.hg38.interval_list, comprised of the full genome minus N bases, unless otherwise noted below. We ran *Strelka2*[94] using default parameters for canonical chromosomes (chr1-22, X,Y,M), as recommended by the authors, and we filtered the final *Strelka2* VCF for PASS variants. We ran *Mutect2* from *GATK* according to Broad best practices outlined from their Workflow Description Language (WDL), and we filtered the final *Mutect2* VCF for PASS variants. To manage memory issues, we ran *VarDictJava*[97] using 20 kb interval chunks of the input BED,

padded by 100 bp on each side, such that if an indel occurred in between intervals, it would be captured. Parameters and filtering followed BCBIO standards except that variants with a variant allele frequency (VAF) $\geq$ 0.05 (instead of $\geq$ 0.10) were retained. The 0.05 VAF increased the true positive rate for indels and decreased the false positive rate for SNVs when using *VarDictJava* in consensus calling. We filtered the final *VarDictJava* VCF for PASS variants with *TYPE=StronglySomatic*. We ran *Lancet* using default parameters, except for those noted below. For input intervals to *Lancet* WGS, we created a reference BED from only the UTR, exome, and start/stop codon features of the GENCODE 31 reference, augmented as recommended with PASS variant calls from *Strelka2* and *Mutect2*. We then padded these intervals by 300 bp on each side during *Lancet* variant calling. Per recommendations for WGS samples, we augmented the Lancet input intervals described above with PASS variant calls from *Strelka2* and *Mutect2* as validation.[98]

### VCF annotation and MAF creation

We normalized INDELs with *bcftools norm* on all PASS VCFs using the *kfdrc_annot_vcf_sub_wf.cwl* subworkflow, release v3 (See Table S5). The Ensembl Variant Effect Predictor (*VEP*),[99] reference release 93, was used to annotate variants and bcftools was used to add population allele frequency (AF) from gnomAD.[100] We annotated SNV and INDEL hotspots from v2 of Memorial Sloan Kettering Cancer Center's (MSKCC) database (See key resources table) as well as the *TERT* promoter mutations C228T and C250T.[101] We annotated SNVs by matching amino acid position (*Protein_position* column in MAF file) with SNVs in the MSKCC database, we matched splice sites to *HGVSp_Short* values in the MSKCC database, and we matched INDELs based on amino acid present within the range of INDEL hotspots values in the MSKCC database. We removed non-hotspot annotated variants with a normal depth less than or equal to 7 and/or gnomAD allele frequency (AF) greater than 0.001 as potential germline variants. We matched *TERT* promoter mutations using hg38 coordinates as indicated in ref. [101]: C228T occurs at 5:1295113 is annotated as existing variant *s1242535815*, *COSM1716563*, or *COSM1716558*, and is 66 bp away from the TSS; C250T occurs at Chr5:1295135, is annotated as existing variant *COSM1716559*, and is 88 bp away from the TSS. We retained variants annotated as *PASS* or *HotSpotAllele=1* in the final set, and we created MAFs using MSKCC's *vcf2maf* tool.

### Gather SNV and INDEL hotspots

We retained all variant calls from *Strelka2*, *Mutect2*, or *Lancet* that overlapped with an SNV or INDEL hotspot in a hotspot-specific MAF file, which we then used for select analyses as described below.

### Consensus SNV calling

Our SNV calling process led to separate sets of predicted mutations for each caller. We considered mutations to describe the same change if they were identical for the following MAF fields: *Chromosome*, *Start_Position*, *Reference_Allele*, *Allele*, and *Tumor_Sample_Barcode*. *Strelka2* does not call multinucleotide variants (MNV), but instead calls each component SNV as a separate mutation, so we separated MNV calls from *Mutect2* and *Lancet* into consecutive SNVs before comparing them to *Strelka2* calls. We examined VAFs produced by each caller and compared their overlap with each other (Figure S2). *VarDictJava* calls included many variants that were not identified by other callers (Figure S2C), while the other callers produced results that were relatively consistent with one another. Many of these *VarDictJava*-specific calls were variants with low allele frequency (Figure S2B). We therefore derived consensus mutation calls as those shared among the other three callers (*Strelka2*, *Mutect2*, and *Lancet*), and we did not further consider *VarDictJava* calls due to concerns it called a large number of false positives. This decision had minimal impact on results because *VarDictJava* also identified nearly every mutation that the other three callers identified, in addition to many unique mutations.

### Somatic copy number variant calling (WGS samples only)

We used *Control-FREEC*[102,103] and *CNVkit*[104] for copy number variant calls. For both algorithms, the *germline_sex_estimate* (described below) was used as input for sample sex and germline variant calls (above) were used as input for BAF estimation. *Control-FREEC* was run on human genome reference hg38 using the optional parameters of a 0.05 coefficient of variation, ploidy choice of 2–4, and BAF adjustment for tumor-normal pairs. *Theta2*[105] used *VarDictJava* germline and somatic calls, filtered on PASS and strongly somatic, to infer tumor purity. *Theta2* purity was added as an optional parameter to *CNVkit* to adjust copy number calls. *CNVkit* was run on human genome reference hg38 using the optional parameters of Theta2 purity and BAF adjustment for tumor-normal pairs. We used *GISTIC*[106] on the *CNVkit* and the consensus CNV segmentation files to generate gene-level copy number abundance (Log R Ratio) as well as chromosomal arm copy number alterations using the parameters specified in the (*run-gistic* analysis module in the OpenPBTA Analysis repository).

### Consensus CNV calling

For each caller and sample, we called CNVs based on consensus among *Control-FREEC*,[102,103] *CNVkit*,[104] and *Manta*.[107] We specifically included CNVs called significant by *Control-FREEC* (p value <0.01) and *Manta* calls that passed all filters in consensus calling. We removed sample and consensus caller files with more than 2,500 CNVs because we expected these to be noisy and derive poor quality samples based on cutoffs used in *GISTIC*.[106] For each sample, we included the regions in the final consensus set: 1) regions with reciprocal overlap of 50% or more between at least two of the callers; 2) smaller CNV regions in which more than 90% of regions are covered by another caller. We did not include any copy number alteration called by a single algorithm in the consensus file. We defined copy number as *NA* for any regions that had a neutral call for the samples included in the consensus file. We merged CNV regions within 10,000 bp of each other with the same direction of gain or loss into single region. We filtered out any CNVs that overlapped 50% or more with immunoglobulin, telomeric, centromeric, segment duplicated regions, or that were shorter than 3000 bp.

### Somatic structural variant calling (WGS samples only)

We used *Manta*[107] for structural variant (SV) calls, and we limited to regions used in *Strelka2*. The hg38 reference for SV calling used was limited to canonical chromosome regions. We used *AnnotSV*[108] to annotate *Manta* output. All associated workflows are available in the workflows GitHub repository.

### Gene expression

#### Abundance estimation

We used *STAR*[109] to align paired-end RNA-seq reads, and we used the associated alignment for all subsequent RNA analysis. We used Ensembl GENCODE 27 "Comprehensive gene annotation" (see key resources table) as a reference. We used *RSEM*[110] for both FPKM and TPM transcript- and gene-level quantification.

#### Gene expression matrices with unique HUGO symbols

To enable downstream analyses, we next identified gene symbols that map to multiple Ensembl gene identifiers (in GENCODE v27, 212 gene symbols map to 1866 Ensembl gene identifiers), known as multi-mapped gene symbols, and ensured unique mappings (*collapse-rnaseq* analysis module in the OpenPBTA Analysis repository). To this end, we first removed genes with no expression from the *RSEM* abundance data by requiring an FPKM >0 in at least 1 sample across the PBTA cohort. We computed the mean FPKM across all samples per gene. For each multi-mapped gene symbol, we chose the Ensembl identifier corresponding to the maximum mean FPKM, using the assumption that the gene identifier with the highest expression best represented the expression of the gene. After collapsing gene identifiers, 46,400 uniquely-expressed genes remained in the poly-A dataset, and 53,011 uniquely-expressed genes remained in the stranded dataset.

#### Gene fusion detection

We set up *Arriba*[111] and *STAR-Fusion*[112] fusion detection tools using CWL on CAVATICA. For both of these tools, we used aligned BAM and chimeric SAM files from *STAR* as inputs and *GRCh38_gencode_v27* GTF for gene annotation. We ran *STAR-Fusion* with default parameters and annotated all fusion calls with the *GRCh38_v27_CTAT_lib_Feb092018.plug-n-play.tar.gz* file from the *STAR-Fusion* release. For *Arriba*, we used a blacklist file *blacklist_hg38_GRCh38_2018-11-04.tsv.gz* from the *Arriba* release to remove recurrent fusion artifacts and transcripts present in healthy tissue. We provided *Arriba* with strandedness information for stranded samples, or we set it to auto-detection for poly-A samples. We used *FusionAnnotator* on *Arriba* fusion calls to harmonize annotations with those of *STAR-Fusion*. The RNA expression and fusion workflows can be found in the D3b GitHub repository. The *FusionAnnotator* workflow we used for this analysis can be found in the D3b GitHub repository.

## QUANTIFICATION AND STATISTICAL ANALYSIS

All p values are two-sided unless otherwise stated. Z-scores were calculated using the formula $z = (x - \mu)/\sigma$ where $x$ is the value of interest, $\mu$ is the mean, and $\sigma$ is the standard deviation.

### Tumor purity (*tumor-purity-exploration* module)

Estimating tumor fraction from RNA directly is challenging because most assume tumor cells comprise all non-immune cells,[113] which is not a valid assumption for many diagnoses in the PBTA cohort. We therefore used Theta2 (as described in the "somatic copy number variant calling section" method details section) to infer tumor purity from WGS samples, further assuming that co-extracted RNA and DNA samples had the same tumor purity. We then created a set of stranded RNA-Seq data thresholded by median tumor purity of the cancer group to rerun selected transcriptomic analyses: *telomerase-activity-prediction*, *tp53_nf1_score*, *transcriptomic-dimension-reduction*, *immune-deconv*, and *gene set enrichment analysis*. Note that these thresholded analyses, which only considered stranded RNA samples that also had co-extracted DNA, were performed in their respective OpenPBTA analyses modules (not within *tumor-purity-exploration*).

### Recurrently mutated genes and co-occurrence of gene mutations (*interaction-plots* analysis module)

Using the consensus SNV calls, we identified genes that were recurrently mutated in the OpenPBTA cohort, including nonsynonymous mutations with a VAF >5% among the set of independent samples. We used *VEP*[99] annotations, including "High" and "Moderate" consequence types as defined in the R package *Maftools*,[114] to determine the set of nonsynonymous mutations. For each gene, we then tallied the number of samples that had at least one nonsynonymous mutation.

For genes that contained nonsynonymous mutations in multiple samples, we calculated pairwise mutation co-occurrence scores. This score was defined as $I(-\log_{10}(P))$ where $I$ is 1 when the odds ratio is >1 (indicating co-occurrence), and $-1$ when the odds ratio is <1 (indicating mutual exclusivity), with $P$ defined by Fisher's Exact Test.

### Focal copy number calling (*focal-cn-file-preparation* analysis module)

We added the ploidy inferred via *Control-FREEC* to the consensus CNV segmentation file and used the ploidy and copy number values to define gain and loss values broadly at the chromosome level. We used *bedtools coverage*[115] to add cytoband status using the UCSC cytoband file[116] (See key resources table). The output status call fractions, which are values of the loss, gain, and callable fractions of each cytoband region, were used to define dominant status at the cytoband-level. We calculated the weighted means of each status call fraction using band length. We used the weighted means to define the dominant status at the chromosome arm-level.

A status was considered dominant if more than half of the region was callable and the status call fraction was greater than 0.9 for that region. We adopted this 0.9 threshold to ensure that the dominant status fraction call was greater than the remaining status fraction calls in a region.

We aimed to define focal copy number units to avoid calling adjacent genes in the same cytoband or arm as copy number losses or gains where it would be more appropriate to call the broader region a loss or gain. To determine the most focal units, we first considered the dominant status calls at the chromosome arm-level. If the chromosome arm dominant status was callable but not clearly defined as a gain or loss, we instead included the cytoband-level status call. Similarly, if a cytoband dominant status call was callable but not clearly defined as a gain or loss, we instead included gene-level status call. To obtain the gene-level data, we used the *IRanges* package in R[117] to find overlaps between the segments in the consensus CNV file and the exons in the GENCODE v27 annotation file (See key resources table). If the copy number value was 0, we set the status to "deep deletion". For autosomes only, we set the status to "amplification" when the copy number value was greater than two times the ploidy value. We plotted genome-wide gains and losses in (Figure S3C) using the R package *ComplexHeatmap*.[118]

### Breakpoint density (WGS samples only; *chromosomal-instability* analysis module)

We defined breakpoint density as the number of breaks per genome or exome per sample. For Manta SV calls, we filtered to retain "PASS" variants and used breakpoints from the algorithm. For consensus CNV calls, if |log2 ratio| > log2(1), we annotated the segment as a break. We then calculated breakpoint density as:

$$\text{breakpoint density} = \frac{N \text{ breaks}}{\text{Size in Mb of effectively surveyed genome}}$$

### Chromothripsis analysis (WGS samples only; *chromothripsis* analysis module)

Considering only chromosomes 1–22 and X, we identified candidate chromothripsis regions in the set of independent tumor WGS samples with ShatterSeek,[119] using Manta SV calls that passed all filters and consensus CNV calls. We modified the consensus CNV data to fit *ShatterSeek* input requirements as follows: we set CNV-neutral or excluded regions as the respective sample's ploidy value from *Control-FREEC*, and we then merged consecutive segments with the same copy number value. We classified candidate chromothripsis regions as high- or low-confidence using the statistical criteria described by the *ShatterSeek* authors.

### Immune profiling and deconvolution (immune-deconv *analysis module*)

We used the R package *immunedeconv*[120] with the method *quanTIseq*[121] to deconvolute various immune cell types in tumors using collapsed FPKM RNA-seq, with samples batched by library type and then combined. The *quanTIseq* deconvolution method directly estimates absolute fractions of 10 immune cell types that represent inferred proportions of the cell types in the mixture. Therefore, we utilized *quanTIseq* for inter-sample, intra-sample, and inter-histology score comparisons.

### Gene Set Variation Analysis (gene set enrichment analysis *analysis module*)

We performed Gene Set Variation Analysis (GSVA) on collapsed, log2-transformed RSEM FPKM data for stranded RNA-Seq samples using the *GSVA* Bioconductor package.[122] We specified the parameter *mx.diff=TRUE* to obtain Gaussian-distributed scores for each of the MSigDB hallmark gene sets.[123] We compared GSVA scores among histology groups using ANOVA and subsequent Tukey tests; p values were Bonferroni-corrected for multiple hypothesis testing. We plotted scores by cancer group using the *ComplexHeatmap* R package (Figure 5B).[118]

### Transcriptomic dimension reduction (transcriptomic-dimension-reduction *analysis module*)

We applied Uniform Mani-fold Approximation and Projection (UMAP)[124] to log2-transformed FPKM data for stranded RNA-Seq samples using the *umap* R package (See key resources table). We considered all stranded RNA-Seq samples for this analysis, but we removed genes whose FPKM sum across samples was less than 100. We set the UMAP number of neighbors parameter to 15.

### Fusion prioritization (fusion_filtering *analysis module*)

We performed artifact filtering and additional annotation on fusion calls to prioritize putative oncogenic fusions. Briefly, we considered all in-frame and frameshift fusion calls with at least one junction read and at least one gene partner expressed (TPM >1) to be true calls. If a fusion call had a large number of spanning fragment reads compared to junction reads (spanning fragment minus junction read greater than ten), we removed these calls as potential false positives. We prioritized a union of fusion calls as true calls if the fused genes were detected by both callers, the same fusion was recurrent within a broad histology grouping (>2 samples), or the fusion was specific to the given broad histology. If either 5′ or 3′ genes fused to more than five different genes within a sample, we removed these calls as potential false positives. We annotated putative driver fusions and prioritized fusions based on partners containing known kinases, oncogenes, tumor suppressors, curated transcription factors,[125] COSMIC genes, and/or known TCGA fusions from curated references. Based on pediatric cancer literature review, we added *MYBL1*,[126] *SNCAIP*,[127] *FOXR2*,[128] *TTYH1*,[129] and *TERT*[130–133] to the oncogene list, and we added *BCOR*[128] and *QKI*[134] to the tumor suppressor gene list.

### Oncoprint figure generation (oncoprint-landscape *analysis module*)

We used *Maftools*[114] to generate oncoprints depicting the frequencies of canonical somatic gene mutations, CNVs, and fusions for the top 20 genes mutated across primary tumors within broad histologies of the OpenPBTA dataset. We collated canonical genes from the literature for low-grade gliomas (LGGs),[25] embryonic tumors,[26,28,29,135,136] high-grade gliomas (HGGs),[14,31,32,137] and other

tumors: ependymomas, craniopharyngiomas, neuronal-glial mixed tumors, histiocytic tumors, chordoma, meningioma, and choroid plexus tumors.[33,138–146]

### Mutational signatures (*mutational-signatures* analysis module)

We obtained weights (i.e., exposures) for signature sets using the *deconstructSigs* R package function *whichSignatures()*[147] from consensus SNVs with the BSgenome.Hsapiens.UCSC.hg38 annotations (see key resources table). Specifically, we estimated signature weights across samples for eight signatures previously identified in the Signal reference set of signatures ("RefSig") as associated with adult central nervous system (CNS) tumors.[40] These eight RefSig signatures are 1, 3, 8, 11, 18, 19, N6, and MMR2. Weights for signatures fall in the range zero to one inclusive. *deconstructSigs* estimates the weights for each signature across samples and allows for a proportion of unassigned weights referred to as "Other" in the text. These results do not include signatures with small contributions; *deconstructSigs* drops signature weights that are less than 6%.[147] We plotted mutational signatures for patients with hypermutant tumors (Figure 4E) using the R package *ComplexHeatmap*.[118]

### Tumor mutation burden (*snv-callers* analysis module)

We consider tumor mutation burden (TMB) to be the number of consensus SNVs per effectively surveyed base of the genome. We considered base pairs to be effectively surveyed if they were in the intersection of the genomic ranges considered by the callers used to generate the consensus and where appropriate, regions of interest, such as coding sequences. We calculated TMB as:

$$TMB = \frac{\text{\# of coding sequence SNVs}}{\text{Size in Mb of effectively surveyed genome}}$$

We used the total number coding sequence consensus SNVs for the numerator and the size of the intersection of the regions considered by *Strelka2* and *Mutect2* with coding regions (CDS from GENCODE v27 annotation, see key resources table) as the denominator.

### Clinical data harmonization

#### WHO classification of disease types

Table S1 contains a README, along with sample technical, clinical, and additional metadata used for this study.

#### Molecular subtyping

We performed molecular subtyping on tumors in the OpenPBTA to the extent possible. The *molecular_subtype* field in *pbta-histologies.tsv* contains molecular subtypes for tumor types selected from *pathology_diagnosis* and *pathology_free_text_diagnosis* fields as described below, following World Health Organization 2016 classification criteria.[20] We further categorized broad tumor histologies into smaller groupings we denote "cancer groups."

Medulloblastoma (MB) subtypes SHH, WNT, Group 3, and Group 4 were predicted using the consensus of two RNA expression classifiers: *MedulloClassifier*[22] and *MM2S*[23] on the RSEM FPKM data (*molecular-subtyping-MB* analysis module). The 43 "true positive" subtypes were manually curated from pathology reports by two independent reviewers.

High-grade glioma (HGG) subtypes were derived (*molecular-subtyping-HGG* analysis module) using the following criteria.

1. If any sample contained an *H3F3A* p.K28M, *HIST1H3B* p.K28M, *HIST1H3C* p.K28M, or *HIST2H3C* p.K28M mutation and no *BRAF* p.V600E mutation, it was subtyped as *DMG, H3 K28.*
2. If any sample contained an *HIST1H3B* p.K28M, *HIST1H3C* p.K28M, or *HIST2H3C* p.K28M mutation and a *BRAF* p.V600E mutation, it was subtyped as *DMG, H3 K28, BRAF V600E.*
3. If any sample contained an *H3F3A* p.G35V or p.G35R mutation, it was subtyped as *HGG, H3 G35.*
4. If any high-grade glioma sample contained an *IDH1* p.R132 mutation, it was subtyped as *HGG, IDH.*
5. If a sample was initially classified as HGG, had no defining histone mutations, and a BRAF p.V600E mutation, it was subtyped as *BRAF V600E.*
6. All other high-grade glioma samples that did not meet any of these criteria were subtyped as *HGG, H3 wildtype.*

Embryonal tumors were included in non-MB and non-ATRT embryonal tumor subtyping (*molecular-subtyping-embryonal* analysis module) if they met any of the following criteria.

1. A *TTYH1* (5′ partner) fusion was detected.
2. A *MN1* (5′ partner) fusion was detected, with the exception of *MN1::PATZ1* since it is an entity separate of CNS HGNET-MN1 tumors.[148].
3. Pathology diagnoses included "Supratentorial or Spinal Cord PNET" or "Embryonal Tumor with Multilayered Rosettes".
4. A pathology diagnosis of "Neuroblastoma", where the tumor was not indicated to be peripheral or metastatic and was located in the CNS.
5. Any sample with "embryonal tumor with multilayer rosettes, ros (who grade iv)", "embryonal tumor, nos, congenital type", "ependymoblastoma" or "medulloepithelioma" in pathology free text.

Non-MB and non-ATRT embryonal tumors identified with the above criteria were further subtyped (*molecular-subtyping-embryonal* analysis module) using the criteria below.[149–152]

1. Any RNA-seq biospecimen with *LIN28A* overexpression, plus a *TYH1* fusion (5′ partner) with a gene adjacent or within the C19MC miRNA cluster and/or copy number amplification of the C19MC region was subtyped as *ETMR, C19MC-altered* (Embryonal tumor with multilayer rosettes, chromosome 19 miRNA cluster altered).[129,153].
2. Any RNA-seq biospecimen with *LIN28A* overexpression, a *TTYH1* fusion (5′ partner) with a gene adjacent or within the C19MC miRNA cluster but no evidence of copy number amplification of the C19MC region was subtyped as *ETMR, NOS* (Embryonal tumor with multilayer rosettes, not otherwise specified).[129,153]
3. Any RNA-seq biospecimen with a fusion having a 5′ *MN1* and 3′ *BEND2* or *CXXC5* partner were subtyped as *CNS HGNET-MN1* [Central nervous system (CNS) high-grade neuroepithelial tumor with *MN1* alteration].
4. Non-MB and non-ATRT embryonal tumors with internal tandem duplication (as defined in[154]) of *BCOR* were subtyped as *CNS HGNET-BCOR* (CNS high-grade neuroepithelial tumor with *BCOR* alteration).
5. Non-MB and non-ATRT embryonal tumors with overexpression and/or gene fusions in *FOXR2* were subtyped as *CNS NB-FOXR2* (CNS neuroblastoma with *FOXR2* activation).
6. Non-MB and non-ATRT embryonal tumors with *CIC::NUTM1* or other *CIC* fusions, were subtyped as *CNS EFT-CIC* (CNS Ewing sarcoma family tumor with *CIC* alteration).[128]
7. Non-MB and non-ATRT embryonal tumors that did not fit any of the above categories were subtyped as *CNS Embryonal, NOS* (CNS Embryonal tumor, not otherwise specified).

Neurocytoma subtypes central neurocytoma (CNC) and extraventricular neurocytoma (EVN) were assigned (*molecular-subtyping-neurocytoma* analysis module) based on the primary site of the tumor.[155] If the tumor's primary site was "ventricles," we assigned the subtype as CNC; otherwise, we assigned the subtype as EVN.

Craniopharyngiomas (CRANIO) were subtyped (*molecular-subtyping-CRANIO* analysis module) into adamantinomatous (*CRANIO, ADAM*), papillary (*CRANIO, PAP*) or undetermined (*CRANIO, To be classified*) based on the following criteria.[156,157]

1. Craniopharyngiomas from patients over 40 years old with a *BRAF* p.V600E mutation were subtyped as *CRANIO, PAP*.
2. Craniopharyngiomas from patients younger than 40 years old with mutations in exon 3 of *CTNNB1* were subtyped as *CRANIO, ADAM*.
3. Craniopharyngiomas that did not fall into the above two categories were subtyped as *CRANIO, To be classified*.

A molecular subtype of *EWS* was assigned to any tumor with a *EWSR1* fusion or with a *pathology_diagnosis* of *Ewings Sarcoma* (*molecular-subtyping-EWS* analysis module).

LGG or glialneuronal tumors (GNT) were subtyped (*molecular-subtyping-LGAT* analysis module) based on SNV, fusion, and CNV status based on[21] and as described below.

1. If a sample contained a *NF1* somatic mutation, either nonsense or missense, it was subtyped as *LGG, NF1-somatic*.
2. If a sample contained *NF1* germline mutation, as indicated by a patient having the neurofibromatosis cancer predisposition, it was subtyped as *LGG, NF1-germline*.
3. If a sample contained the *IDH* p.R132 mutation, it was subtyped as *LGG, IDH*.
4. If a sample contained a histone p.K28M mutation in either *H3F3A, H3F3B, HIST1H3B, HIST1H3C*, or *HIST2H3C*, or if it contained a p.G35R or p.G35V mutation in *H3F3A*, it was subtyped as *LGG, H3*.
5. If a sample contained *BRAF* p.V600E or any other non-canonical *BRAF* mutations in the kinase (PK_Tyr_Ser-Thr) domain PF07714 (see key resources table), it was subtyped as *LGG, BRAF V600E*.
6. If a sample contained *KIAA1549::BRAF* fusion, it was subtyped as *LGG, KIAA1549::BRAF*.
7. If a sample contained SNV or indel in either *KRAS, NRAS, HRAS, MAP2K1, MAP2K2, MAP2K1, ARAF, RAF1*, or non-kinase domain of *BRAF*, or if it contained *RAF1* fusion, or *BRAF* fusion that was not *KIAA1549::BRAF*, it was subtyped as *LGG, other MAPK*.
8. If a sample contained SNV in either *MET, KIT* or *PDGFRA*, or if it contained fusion in *ALK, ROS1, NTRK1, NTRK2, NTRK3* or *PDGFRA*, it was subtyped as *LGG, RTK*.
9. If a sample contained *FGFR1* p.N546K, p.K656E, p.N577, or p. K687 hotspot mutations, or tyrosine kinase domain tandem duplication (See key resources table), or *FGFR1* or *FGFR2* fusions, it was subtyped as *LGG, FGFR*.
10. If a sample contained *MYB* or *MYBL1* fusion, it was subtyped as *LGG, MYB/MYBL1*.
11. If a sample contained focal CDKN2A and/or CDKN2B deletion, it was subtyped as *LGG, CDKN2A/B*.

For LGG tumors that did not have any of the above molecular alterations, if both RNA and DNA samples were available, it was subtyped as *LGG, wildtype*. Otherwise, if either RNA or DNA sample was unavailable, it was subtyped as *LGG, To be classified*.

If pathology diagnosis was *Subependymal Giant Cell Astrocytoma (SEGA)*, the *LGG* portion of molecular subtype was recoded to *SEGA*.

Lastly, for all LGG- and GNT-subtyped samples, if the tumors were glialneuronal in origin, based on *pathology_free_text_diagnosis* entries of *desmoplastic infantile*, *desmoplastic infantile ganglioglioma*, *desmoplastic infantile astrocytoma* or *glioneuronal*, each was recoded as follows: If pathology diagnosis is *Low-grade glioma/astrocytoma (WHO grade I/II)* or *Ganglioglioma*, the *LGG* portion of the molecular subtype was recoded to *GNT*.

Ependymomas (EPN) were subtyped (*molecular-subtyping-EPN* analysis module) into *EPN, ST RELA*, *EPN, ST YAP1*, *EPN, PF A* and *EPN, PF B* based on evidence for these molecular subgroups as described in Pajtler et al.[139] Briefly, fusion, CNV and gene expression data were used to subtype EPN as follows.

1. Any tumor with fusions containing *RELA* as fusion partner, e.g., *C11orf95::RELA*, *LTBP3::RELA*, was subtyped as *EPN, ST RELA.*
2. Any tumor with fusions containing *YAP1* as fusion partner, such as *C11orf95::YAP1*, *YAP1::MAMLD1* and *YAP1::FAM118B*, was subtyped as *EPN, ST YAP1.*
3. Any tumor with the following molecular characterization would be subtyped as *EPN, PF A*:
   ● *CXorf67* expression *Z* score of over 3
   ● *TKTL1* expression *Z* score of over 3 and 1q gain
4. Any tumor with the following molecular characterization would be subtyped as *EPN, PF B*:
   ● *GPBP17* expression *Z* score of over 3 and loss of 6q or 6p
   ● *IFT46* expression *Z* score of over 3 and loss of 6q or 6p

Any tumor with the above molecular characteristics would be exclusively subtyped to the designated group.

For all other remaining EPN tumors without above molecular characteristics, they would be subtyped to *EPN, ST RELA* and *EPN, ST YAP1* in a non-exclusive way (e.g., a tumor could have both *EPN, ST RELA* and *EPN, ST YAP1* subtypes) if any of the following alterations were present.

1. Any tumor with the following alterations was assigned *EPN, ST RELA*:
   ● *PTEN::TAS2R1* fusion
   ● chromosome 9 arm (9p or 9q) loss
   ● RELA expression *Z* score of over 3
   ● L1CAM expression *Z* score of over 3
2. Any tumor with the following alterations was assigned *EPN, ST YAP1*:
   ● *C11orf95::MAML2* fusion
   ● chromosome 11 short arm (11p) loss
   ● chromosome 11 long arm (11q) gain
   ● ARL4D expression *Z* score of over 3
   ● CLDN1 expression *Z* score of over 3

After all relevant tumor samples were subtyped by the above molecular subtyping modules, the results from these modules, along with other clinical information (such as pathology diagnosis free text), were compiled in the *molecular-subtyping-pathology* module and integrated into the OpenPBTA data in the *molecular-subtyping-integrate* module.

### TP53 alteration annotation (*tp53_nf1_score* analysis module)

We annotated *TP53* altered HGG samples as either *TP53 lost* or *TP53 activated* and integrated this within the molecular subtype. To this end, we applied a *TP53* inactivation classifier originally trained on TCGA pan-cancer data[42] to the matched RNA expression data, with samples batched by library type. Along with the *TP53* classifier scores, we collectively used consensus SNV and CNV, SV, and reference databases that list *TP53* hotspot mutations[158,159] and functional domains[160] to determine *TP53* alteration status for each sample. We adopted the following rules for calling either *TP53 lost* or *TP53 activated*.

1. If a sample had either of the two well-characterized *TP53* gain-of-function mutations, p.R273C or p.R248W,[43] we assigned *TP53 activated* status.
2. Samples were annotated as *TP53 lost* if they contained i) a *TP53* hotspot mutation as defined by IARC *TP53* database or the MSKCC cancer hotspots database[158,159] (see also, key resources table), ii) two *TP53* alterations, including SNV, CNV or SV, indicative of probable biallelic alterations; iii) one *TP53* somatic alteration, including SNV, CNV, or SV or a germline *TP53* mutation indicated by the diagnosis of Li-Fraumeni syndrome (LFS),[161] or iv) one germline *TP53* mutation indicated by LFS and the *TP53* classifier score for matched RNA-Seq was greater than 0.5.

### Prediction of participants' genetic sex

Participant metadata included a reported gender. We used WGS germline data, in concert with the reported gender, to predict participant genetic sex so that we could identify sexually dimorphic outcomes. This analysis may also indicate samples that may have been contaminated. We used the *idxstats* utility from *SAMtools*[162] to calculate read lengths, the number of mapped reads, and the

corresponding chromosomal location for reads to the X and Y chromosomes. We used the fraction of total normalized X and Y chromosome reads that were attributed to the Y chromosome as a summary statistic. We manually reviewed this statistic in the context of reported gender and determined that a threshold of less than 0.2 clearly delineated female samples. We marked fractions greater than 0.4 as predicted males, and we marked samples with values in the inclusive range 0.2–0.4 as unknown. We performed this analysis through CWL on CAVATICA. We added resulting calls to the histologies file under the column header *germline_sex_estimate*.

### Selection of independent samples (*independent-samples* analysis module)

Certain analyses required that we select only a single representative specimen for each individual. In these cases, we identified a single specimen by prioritizing primary tumors and those with whole-genome sequencing available. If this filtering still resulted in multiple specimens, we randomly selected a single specimen from the remaining set.

### Quantification of telomerase activity using gene expression data (*telomerase-activity-prediction* analysis module)

We predicted telomerase activity of tumor samples using the recently developed *EXTEND* method,[44] with samples batched by library type. Briefly, *EXTEND* estimates telomerase activity based on the expression of a 13-gene signature. We derived this signature by comparing telomerase-positive tumors and tumors with activated alternative lengthening of telomeres pathway, a group presumably negative of telomerase activity.

### Survival models (*survival-analysis* analysis module)

We calculated overall survival (OS) as days since initial diagnosis and performed several survival analyses on the OpenPBTA cohort using the survival R package. We performed survival analysis for patients by HGG subtype using the Kaplan-Meier estimator[163] and a log rank test (Mantel-Cox test)[164] on the different HGG subtypes. Next, we used multivariate Cox (proportional hazards) regression analysis[165] to model the following: a) *tp53 scores + telomerase scores + extent of tumor resection + LGG group + HGG group*, in which *tp53 scores* and *telomerase scores* are numeric, *extent of tumor resection* is categorical, and *LGG group* and *HGG group* are binary variables indicating whether the sample is in either broad histology grouping, b) *tp53 scores + telomerase scores + extent of tumor resection* for each *cancer_group* with an N>=3 deceased patients (DIPG, DMG, HGG, MB, and EPN), and c) *quantiseq cell type fractions + CD274 expression + extent of tumor resection* for each *cancer_group* with an N>=3 deceased patients (DIPG, DMG, HGG, MB, and EPN), in which *quantiseq cell type fractions* and *CD274 expression* are numeric.

