## [Document S2. Transparent peer review records for Shapiro et al. · Cell Genomics]

OpenPBTA: An Open Pediatric Brain Tumor Atlas

Author list

Joshua A. Shapiro, Krutika S. Gaonkar, Stephanie J. Spielman, Candace L. Savonen, Chante J. Bethell, Run Jin, Komal S. Rathi, Yuankun Zhu, Laura E. Egolf, Bailey K. Farrow, Daniel P. Miller, Yang Yang, Tejaswi Koganti, Nighat Noureen, Mateusz P. Koptyra, Nhat Duong, Mariarita Santi, Jung Kim, Shannon Robins, Phillip B. Storm, Stephen C. Mack, Jena V. Lilly, Hongbo M. Xie, Payal Jain, Pichai Raman, Brian R. Rood, Rishi R. Lulla, Javad Nazarian, Adam A. Kraya, Zalman Vaksman, Allison P. Heath, Cassie Kline, Laura Scolaro, Angela N. Viaene, Xiaoyan Huang, Gregory P. Way, Steven M. Foltz, Bo Zhang, Anna R. Poetsch, Sabine Mueller, Brian M. Ennis, Michael Prados, Sharon J. Diskin, Siyuan Zheng, Yiran Guo, Shrivats Kannan, Angela J. Waanders, Ashley S. Margol, Meen Chul Kim, Derek Hanson, Nicholas Van Kuren, Jessica Wong, Rebecca S. Kaufman, Noel Coleman, Christopher Blackden, Kristina A. Cole, Jennifer L. Mason, Peter J. Madsen, Carl J. Koschmann, Douglas R. Stewart, Eric Wafula, Miguel A. Brown, Adam C. Resnick, Casey S. Greene, Jo Lynne Rokita, Jaclyn N. Taroni, Children's Brain Tumor Network, Pacific Pediatric Neuro-Oncology Consortium

Summary

Initial submission: Received : September 12th 2022

Scientific editor: Judith Nicholson

First round of review: Number of reviewers: 2
Revision invited : October 5th 2022
Revision received : February 28th 2023

Second round of review: Number of reviewers: 2
Accepted : 4th May 2023

Data freely available: Yes

Code freely available: Yes

This transparent peer review record is not systematically proofread, type-set, or edited. Special characters, formatting, and equations may fail to render properly. Standard procedural text within the editor's letters has been deleted for the sake of brevity, but all official correspondence specific to the manuscript has been preserved.

Referees' reports, first round of review

Reviewer 1

In this Resource study, the authors present the multi-omic datasets (DNA, RNA) compiled by the Children's Brain Tumor Network (CBTN). Much of the CBTTTC and PNOOC data was generated a few years ago and have been made available to investigators (e.g., via Cavatica). CBTN as an organization has been most helpful to any outside investigators using CBTN data, which is commendable. This present study brings together the respective CBTTTC and PNOOC data into a combined resource. It is a good thing for CBTN to have already made these data available to scientific investigators, without making scientific progress wait for this resource paper to be published. The sample-level results compiled in this study (e.g., the DNA- and RNA-related variables, subtyping, and updated histology calls) should help to further annotate the CBTN samples, which other researchers could use as a guide in their own studies utilizing these data.

Specific comments:

1. It seems that the study reports little in terms of new findings or insights. That ought to be fine for a Resource article. The results write-up could perhaps do a better job in places to note which findings are new (or not previously observed based on the author's knowledge) and which are in line with expectations based on past observations. The somatic mutation and fusion events all appear to be within what would be expected.
2. For the RNA-seq dataset, the CBTTTC and PNOOC data were separately generated by the respective data centers. Are we certain that there are no batch effects in the data between CBTTTC and PNOOC samples? It may be that aligning the samples uniformly would have avoided this issue. Do the PNOOC samples represent different histologies from those of CBTTTC and would the UMAP analysis show tumors of the same histology from different centers clustering together?

3. For Data File S1, can we get an additional field indicating which samples are CBTTTC and which are PNOC.
4. There should be a sample-level table for what samples were profiled on which data platforms. For example, some samples may have be profiled by WGS but not WXS and vice versa.
5. Results paragraph 1 could perhaps indicate how many samples were from CBTTTC and how many were from PNOC.
6. For Results and Figures noting the n, it should be indicated somewhere whether the "n" refers to patients or to tumors. Some patients have multiple tumors profiled apparently. When listing the n for a number of categories of interest in a given paragraph, it may be fine to simply note on the first instance whether the n refers to tumors or patients (e.g., "n=12 tumors"). Along these lines, Table 1 should have two columns for n, one for tumors and one for patients.
7. The Supplementary Data File with sample-level DNA-related results needs to be considerably expanded to included all relevant data presented in figures 2 and 3, such as mutation and fusion events for each sample. Underscoring the resource value of this study, one should be able to readily look up which samples had a mutation of interest (just involving the top mutated genes, i.e. the genes represented in the figures).
8. Table 2 lists treatment information for the subset of hypermutated patients. Would treatment information be something to include in the supplementary data files (e.g., for Data File S1)?
9. Wherever a p-value is indicated in the results or a figure, the statistical test used to compute the test should be indicated next to the p-value (e.g., "p=0.02, univariate Cox"). If the same test was used to generated multiple p-values reported in a given paragraph, it should be fine to simply note the test used on the first instance. It would arguably be asking too much of the reader to have to dig through the Methods to understand what test was applied to generate which p-value. If the method used to generate the p-value was very complex, then one can note "see Methods" but otherwise it is more helpful to simply note the test

next to the p-value. For Cox survival analyses, the test should note whether a univariate or multivariate Cox analysis was used.

10. Page 22, multivariate survival analyses include whether a tumor was low-grade (LGG group) or high-grade (HGG group). Does that mean there is an LGG variable to denote the LGG cases and another HGG variable to denote the HGG cases? These respective cases would be mutually exclusive, and then there would be all the other non-glioma histopathologic subtypes, which may also be associated with survival. Wouldn't it be more straightforward to simply have one histology variable, which could be incorporated as a factor in the Cox or regression models?

11. Do the results in Figure 5D represent a multivariate analysis?

12. For all box plots and violin plots, the corresponding figure legends should define the boundaries. For example, "Box plot represents 5% (lower whisker), 25% (lower box), 50% (median), 75% (upper box), and 95% (upper whisker)."

13. Can the following be stated in the Methods section: "All p-values are two-sided unless otherwise specified"?

14. In Discussion, it seems that there could be an acknowledgement of all the ongoing scientific projects that are already utilizing CBTN data. The CBTN.org website lists numerous registered scientific projects and publications utilizing CBTN data. This speaks as to the value of these data as a resource. It is a good thing that CBTN released these data to the scientific community early on, rather than to make everyone wait until this initial resource paper was published. This would be in line with the stated Open Science goals.

15. One drawback with the "open science" model for assembling this manuscript is that things take a lot longer. It seems that this manuscript would have been a few years in the making. This would be in line with similar types of resource studies led by big scientific consortium efforts. The timeline for publication is not an issue here, as the data have already been made available to investigators without publication restrictions. The open science model may work in some instances, though for the individual scientific projects currently registered at

CBTN.org, we can expect that these by and large would be laboratory-driven.

Reviewer 2

The manuscript titled "OpenPBTA: An Open Pediatric Brain Tumor Atlas" by Shapiro et al. describes the first of its kind initiative where through online open collaboration of methodology, data and analysis, scientists, clinicians and data scientists have come together to analyse a large paediatric brain tumour cohort to provide an invaluable resource to better understand these paediatric cancers to unravel molecular biology underpinning these cancers. This manuscript is well written and the initiative of OpenPBTA is a welcome advance to increase global collaboration of data sharing and methodology especially with the increase of personalised medicine platforms worldwide generating an increasing amount of data. However, there are a few major concerns that the authors should address prior to publication. This work could be improved and enhanced by providing additional analysis of the datasets to give a more complete picture by looking at tumour purity associations, microsatellite instability, and improved methodologies that harmonise the datasets and are correlated with the conclusions drawn throughout the manuscript.

Major concerns:

1. A key statement made in the summary about the use of OpenPBTA in molecular tumour boards to aid in clinical decision making. However, this statement is not supported throughout the manuscript as the authors provide no evidence of this claim. Supporting evidence to show the use of OpenPBTA in global molecular tumour boards such as known paediatric precision medicine programs such as INFORM2, ZERO, PROFYLE, PCGP would support this claim and the use of this platform increasing clinical decision making would really strengthen this argument and the overall value of this platform.
2. Overall clarity throughout entire manuscript of which analysis is done on which datasets. The authors have rich and valuable datasets of tumours from primary sample, progressive disease, post-treatment, relapse and second malignancy. Would add value to the analysis to clarify throughout which datasets were used throughout. In addition, clarity in the relationship of these samples in Fig 1B would be highly valuable to the reader and those seeking to understand and use this data in the future. This will also strengthen the authors findings and associations made throughout. Additional results on the molecular changes underpinning the samples that have multiple biospecimens would be highly

valuable and provide a unique insight into these tumours.

3. Additions to the methodologies deployed in the paper would provide enhancements and aid the authors in drawing conclusions more clearly. In the methodology there is no mention of batch correction applied to the transcriptomics data which comes from multiple centres, various preparation and sequencing technologies even though all were ran through the same pipelines. In addition, tumour purity is not assessed from the WGS data and reported on. Improvements to understanding the transcriptomic landscape of paediatric brain tumours including histological clustering would improve if batch correction was applied and tumour purity was taken into consideration.

4. Co-occurrence analysis should be revisited as the conclusions the authors are drawing are in relation to the entire cohort but clear co-occurrences identified are already known in the literature and cancer subtypes mentioned. For example the authors most significant finding is the co-occurrence of TP53 with H3F3A but this comes as no surprise with the high number of DMG samples in the cohort and this co-occurrence is exclusive to this subtype only. This extends to the co-occurrence of ATRX and EGFR which is also specific to DMG only. The statistical confidence intervals are also rather large and the stats are perhaps more influenced by large cohort numbers and the molecular drivers of this specific tumour type.

Minor concerns:

1. Authors address the versioning of data analysis as changes are made but addressing whether all data presented in the manuscript is re-analysed with additional code improvements would be valuable and the time/cost of each of the analysis modules would be beneficial as analysis occurs in a cloud platform.

2. Open collaboration is an excellent resource for code and analysis share to produce data for manuscripts more readily however more information surrounding the process post submission would be beneficial to understand the integrity of the data. Does the contributor get involved in deeper discussions of the data if required, is any data rejected? Beyond rigorous review of the data is there quality assurance checks where two separate centres analyse the same data with the modules to ensure reproducibility?

3. Authors mention WHO classification 2021 but report on 2016 classification, updating to the latest WHO classifications would improve clarity around specific subtypes. For example in the DMG oncoplot the new subtype of DMG H3K-altered is apparent in these samples.

4. Clarity in methodology around somatic mutation analysis in samples lacking

matching germline needs to be specified and clarified.

5. Telomerase scores are they related to TERT rearrangement events or TERT promoter mutations. Authors mention in the methodology that they explored TERT promoter mutations but no mention if there is a correlation in the results.

6. Conclusions drawn about immunologically cold and hot samples is poorly defined and assessed with limited supporting evidence to state such claims.

7. Discussion mentions using packages MM2S and medulloPackage to subtype medulloblastoma tumours but not shown in results.

8. No mention of z-score calculation approach in methodology

9. In consistency between ependymal and ependymoma tumour this needs to be harmonised throughout

Authors' response to the first round of review

Reviewer #1:

In this Resource study, the authors present the multi-omic datasets (DNA, RNA) compiled by the Children's Brain Tumor Network (CBTN). Much of the CBTTTC and PNOCC data was generated a few years ago and have been made available to investigators (e.g., via Cavatica). CBTN as an organization has been most helpful to any outside investigators using CBTN data, which is commendable. This present study brings together the respective CBTTTC and PNOCC data into a combined resource. It is a good thing for CBTN to have already made these data available to scientific investigators, without making scientific progress wait for this resource paper to be published. The sample-level results compiled in this study (e.g., the DNA- and RNA-related variables, subtyping, and updated histology calls) should help to further annotate the CBTN samples, which other researchers could use as a guide in their own studies utilizing these data. Specific comments: 1. It seems that the study reports little in terms of new findings or insights. That ought to be fine for a Resource article. The results write-up could perhaps do a better job in places to note which findings are new (or not previously observed based on the author's knowledge) and which are in line with expectations based on past observations. The somatic mutation and fusion events all appear to be within what would be expected.

Thank you for acknowledging the importance of real-time data access and the value of this work. We have now updated the results to better clarify what new versus recapitulated findings. The major innovation of this manuscript is the

first-in-kind coupled collaborative open analysis and manuscript writing framework in GitHub. We characterize, to our knowledge, the largest brain tumor cohort to date and provide rich clinical annotations, including creation of over 40 scalable downstream cancer analysis modules, including a set of modules to systematically molecularly subtype tumors with paired WGS/RNA-Seq. We implemented multiple transcriptomic classifiers or packages to inform biological insights: two medulloblastoma classifiers to identify subgroups, a TP53 classifier which enabled further prognostic stratification of H3 K28-mutant DMGs and ependymomas as well as enabled identification of TP53 dysregulation within hypermutant HGGs. Explicitly, Figures 2, 3A-B, S3A-B, 5A-B recapitulate known findings.

2. For the RNA-seq dataset, the CBTTTC and PNOG data were separately generated by the respective data centers. Are we certain that there are no batch effects in the data between CBTTTC and PNOG samples? It may be that aligning the samples uniformly would have avoided this issue. Do the PNOG samples represent different histologies from those of CBTTTC and would the UMAP analysis show tumors of the same histology from different centers clustering together? Thank you for pointing out this possibility.

We have indeed aligned and harmonized all samples using the same workflow. At this time, our PNOG cohort only contains data from one histology, DIPG. Of note is that the PNOG RNA-Seq samples were prepared using polyA libraries and as the majority of the CBTN data are stranded, we sought to re-sequence some of these samples at the onset of the OpenPBTA. Since stranded and polyA datasets have a large batch effect due to library type, we purposely separated the gene expression matrices by library type and subsequently performed all RNA-Seq specific analyses on those Response datasets separately. Thus, the UMAP clustering we show in Figures 5 and S6 were all performed on stranded data. Overall, the PBTA currently contains samples sequenced at four different centers (BGI, Nantomics, BGI@CHOP, and TGEN). To answer the broader question of whether tumors of the same histology sequenced at different centers cluster together, we performed UMAP on all stranded samples (excluding PNOG003 polyA samples sequenced at TGEN) and indeed show that for the nine cancer groups which satisfy these criteria, samples cluster by histology rather than sequencing center. We therefore do not observe a batch effect by sequencing center. Overall, UMAP visualization shows that sequencing center is likely not a severe batch effect, and these visualizations emphasize that sequencing center is highly confounded with diagnosis making formal batch

correction challenging. That said, there does appear to be a group of samples with different diagnoses from BGI@CHOP which cluster together, which may be a batch-induced effect. In addition, there are a couple DMGs that cluster with MBs, though there are multiple sequencing centers in this “cluster.” Arrows are shown in the second plot below pointing to these two caveated regions. This analysis can be found at: <https://github.com/AlexsLemonade/OpenPBTA-analysis/blob/master/analyses/transcriptomic-dimension-reduction/04-explore-sequencing-center-effects.Rmd>.

3. For Data File S1, can we get an additional field indicating which samples are CBTTTC and which are PNOC. Column “S” (“cohort”) in the currently included Data File S1 denotes the cohort as CBTN or PNOC. Future releases of OpenPBTA and OpenPedCan (<https://github.com/PediatricOpenTargets/OpenPedCan-analysis>) will no longer discriminate between CBTN and PNOC (and they are currently in the process of being compiled into one dbGAP study).

This will be done to ensure patient de-identifiability and inability for users to back-map to PNOC patient IDs based on prior studies.

4. There should be a sample-level table for what samples were profiled on which data platforms. For example, some samples may have been profiled by WGS but not WXS and vice versa.

Data File S1 is indeed a sample-level table. Column “E” in the currently included Data File S1 denotes the experimental strategy of each sample. All WXS samples have also had WGS performed, and all main figures show results derived from WGS and/or RNA-Seq samples. Common WXS and WGS samples were used in Supplemental Figure S2G to compare the variant allele frequency (VAF) from Lancet across experimental strategies.

5. Results paragraph 1 could perhaps indicate how many samples were from

CBTTC and how many were from PNOc.

Thank you for the suggestion. We have updated Results paragraph 1 to read: “We previously performed whole genome sequencing (WGS), whole exome sequencing (WXS), and RNA sequencing (RNA-Seq) on matched tumor and normal tissues as well as selected cell lines from 943 patients from the Pediatric Brain Tumor Atlas (PBTA), consisting of 911 patients from the [Children’s Brain Tumor Network (CBTN)](<https://CBTN.org>) and 32 patients from the [Pacific Pediatric Neuro-Oncology Consortium (PNOc)](<https://pnoc.us/>) (Figure {@fig:Fig1}A**).”**

6. For Results and Figures noting the n, it should be indicated somewhere whether the "n" refers to patients or to tumors. Some patients have multiple tumors profiled apparently. When listing the n for a number of categories of interest in a given paragraph, it may be fine to simply note on the first instance whether the n refers to tumors or patients (e.g., "n=12 tumors"). Along these lines, Table 1 should have two columns for n, one for tumors and one for patients.

Thank you for the suggestion. Indeed, there are patients with multiple tumors profiled, and for this reason, we had created a module called “independent-samples” (<https://github.com/AlexsLemonade/OpenPBTA-analysis/tree/master/analyses/independent-samples>). In this module, we randomly select one tumor per patient, prioritizing WGS samples, to obtain a list of one independent biospecimen per patient. These sample lists are generated with each new data release (we have now had 23). We have generated these lists for “primary tumors only” and “primary plus” (cases in which there is no primary but we randomly select another available tumor from: secondary, progression, or post-mortem). This ensures that we do not double count tumors for any analysis and that we do not have a selection bias. Throughout this manuscript, we present the findings for primary tumors (one per patient), unless otherwise noted (for example, Table 2, Figure 4E), and we have now updated the results section to reflect this. We have changed text throughout to read “tumors” instead of “samples” to clarify that we are referring to a tumor-specific analysis, rather than a possibility of multiple samples per tumor. In addition, we have added patient-level Ns to Table 1. Overall, 577 patients had 644 tumors molecularly subtyped.

7. The Supplementary Data File with sample-level DNA-related results needs to be considerably expanded to include all relevant data presented in figures 2 and 3, such as mutation and fusion events for each sample. Underscoring the resource value of this study, one should be able to readily look up which samples had a

mutation of interest (just involving the top mutated genes, i.e. the genes represented in the figures).

We agree that having sample-level data readily available is essential and for that reason, we include this within our GitHub and CAVATICA (<https://cavatica.sbgenomics.com/u/cavatica/openpbta>) data releases and relevant analysis modules. We prefer releasing the data in this manner because we only include primary tumor analyses within the manuscript, but release the entire dataset through GitHub. The repository and CAVATICA project are both open and the files can be easily downloaded by any user. As an example, one can retrieve the consensus MAF file from this link: <https://cavatica.sbgenomics.com/u/cavatica/openpbta/files/63c6b17bfacdd82011aa6299/>, consensus CNV seg file from this link: <https://cavatica.sbgenomics.com/u/cavatica/openpbta/files/63c6b17afacdd82011aa61ef/>, and the putative oncogenic fusion calls from this link: <https://cavatica.sbgenomics.com/u/cavatica/openpbta/files/63c6b17bfacdd82011aa6250/>. In addition, a user can browse the OpenPBTA PedcBioportal project (<https://pedcbioportal.kidsfirstdrc.org/study/summary?id=openpbta>) to look at individual-level data. All of these links are included in the STAR Methods in the “Data and code availability” section. In addition, we now include a high-level table in Supplemental Table 2 showing number of samples per histology with type of alteration per gene.

8. Table 2 lists treatment information for the subset of hypermutated patients. Would treatment information be something to include in the supplementary data files (e.g., for Data File S1)?

Thank you for the suggestion. The CBTN data recently underwent a massive transition to RedCap, but the treatment data is still very much under QC and harmonization by the CBTN Clinical Data Working Group. This includes assessing which patients are currently on open clinical trials, gathering all drug treatments from trials, and determining which information can be shared without risk of de-identification. Table 2 was created through extensive manual curation and QC. We agree this is a great idea and we will include this information when we are able within OpenPedCan.

9. Wherever a p-value is indicated in the results or a figure, the statistical test used to compute the test should be indicated next to the p-value (e.g., "p=0.02, univariate Cox"). If the same test was used to generate multiple p-values reported in a given paragraph, it should be fine to simply note the test used on the first instance. It would arguably be asking too much of the reader to have to

dig through the Methods to understand what test was applied to generate which p-value. If the method used to generate the p-value was very complex, then one can note "see Methods" but otherwise it is more helpful to simply note the test next to the p-value. For Cox survival analyses, the test should note whether a univariate or multivariate Cox analysis was used.

Thank you for this suggestion. All reported Cox statistics used a multivariate model as stated in the STAR Methods. We additionally add within the Figures 4 and 5 legends that this was a multivariate Cox.

10. Page 22, multivariate survival analyses include whether a tumor was low-grade (LGG group) or high-grade (HGG group). Does that mean there is an LGG variable to denote the LGG cases and another HGG variable to denote the HGG cases? These respective cases would be mutually exclusive, and then there would be all the other non-glioma histopathologic subtypes, which may also be associated with survival. Wouldn't it be more straightforward to simply have one histology variable, which could be incorporated as a factor in the Cox or regression models?

Apologies for the confusion. The variables were set up to categorize all tumors in the following way: 1. LGG group: "LGG" or "non-LGG" 2. HGG group: "HGG" or "non-HGG" We then added these two group variables into the additive model: "tp53 scores + telomerase scores + extent of tumor resection + LGG group + HGG group" These groups were created as covariates since it is known that prognosis improves with a better resection and worsens with a higher grade tumor. We used HGG only here, rather than "all high grade tumors" because the clinical data is not fully annotated for WHO grade. Therefore, we chose two variables with high confidence as covariates. It is not more straightforward to add the histology variable. With the low N per many individual histologies, the models fail to converge. We clarified the text to denote the LGG and HGG groups were glioma-specific.

11. Do the results in Figure 5D represent a multivariate analysis?

Yes and we have clarified the text as requested in comment 9.

12. For all box plots and violin plots, the corresponding figure legends should define the boundaries. For example, "Box plot represents 5% (lower whisker), 25% (lower box), 50% (median), 75% (upper box), and 95% (upper whisker)."

Thank you for the suggestion - we have added this to Figure 4, 5, and S6 legends.

13. Can the following be stated in the Methods section: "All p-values are two-

sided unless otherwise specified"?

We added the above statement to the QUANTIFICATION AND STATISTICAL ANALYSIS section in STAR Methods.

14. In Discussion, it seems that there could be an acknowledgement of all the ongoing scientific projects that are already utilizing CBTN data. The CBTN.org website lists numerous registered scientific projects and publications utilizing CBTN data. This speaks as to the value of these data as a resource. It is a good thing that CBTN released these data to the scientific community early on, rather than to make everyone wait until this initial resource paper was published. This would be in line with the stated Open Science goals.

Thank you for pointing this out. Indeed, the CBTN has 199 (and counting) approved data projects! We agree that these data are a valuable resource and think that we will find cures faster with earlier data sharing and collaboration. We have added the following to the beginning of our discussion: "The CBTN released the raw genomic data for the PBTA in September 2018 without embargo to allow researchers immediate access to begin making discoveries on behalf of children with CNS tumors everywhere. Since the release of the raw data, the CBTN has approved nearly 200 data research projects 4 from 69 different institutions, with 60% from non-CBTN sites."

15. One drawback with the "open science" model for assembling this manuscript is that things take a lot longer. It seems that this manuscript would have been a few years in the making. This would be in line with similar types of resource studies led by big scientific consortium efforts.

The timeline for publication is not an issue here, as the data have already been made available to investigators without publication restrictions. The open science model may work in some instances, though for the individual scientific projects currently registered at CBTN.org, we can expect that these by and large would be laboratory-driven. The OpenPBTA, its analyses and manuscript writing, began in September 2019 and our time from project inception to manuscript submission was three years in spite of a > 3 year pandemic which has (and still does) globally set back research. We have repeatedly learned and optimized many aspects of this project and indeed utilize the same type of GitHub data release, pull-request, code review, and/or continuous integration model in various ongoing research lab projects now. This collaborative model, while it may take additional time to organize and review code, saves time in the long run by ensuring reproducibility and has the added benefit of a second pair of eyes on the data and analyses. We at D3b have found it an extremely

efficient model for organization of code for manuscript data analyses and figure generation for both projects with internal or external code contributors. We have updated the results and discussion sections to reflect these benefits of an open science analysis framework.

Reviewer #2:

The manuscript titled "OpenPBTA: An Open Pediatric Brain Tumor Atlas" by Shapiro et al. describes the first of its kind initiative where through online open collaboration of methodology, data and analysis, scientists, clinicians and data scientists have come together to analyse a large paediatric brain tumour cohort to provide an invaluable resource to better understand these paediatric cancers to unravel molecular biology underpinning these cancers. This manuscript is well written and the initiative of OpenPBTA is a welcome advance to increase global collaboration of data sharing and methodology especially with the increase of personalised medicine platforms worldwide generating an increasing amount of data. However, there are a few major concerns that the authors should address prior to publication. This work could be improved and enhanced by providing additional analysis of the datasets to give a more complete picture by looking at tumour purity associations, microsatellite instability, and improved methodologies that harmonise the datasets and are correlated with the conclusions drawn throughout the manuscript.

Major concerns:

1. A key statement made in the summary about the use of OpenPBTA in molecular tumour boards to aid in clinical decision making. However, this statement is not supported throughout the manuscript as the authors provide no evidence of this claim. Supporting evidence to show the use of OpenPBTA in global molecular tumour boards such as known paediatric precision medicine programs such as INFORM2, ZERO, PROFYLE, PCGP would support this claim and the use of this platform increasing clinical decision making would really strengthen this argument and the overall value of this platform.

Thank you for this opportunity to clarify. We include in our discussion a reference to this since the trials which utilize the OpenPBTA data are active. This currently includes PNOC008, PNOC027, PNOC030, and will include Children's Hospital of Philadelphia (CHOP) tumor boards at the Division of Genome Diagnostics through our recently funded CHOP Frontier Program (<https://www.research.chop.edu/center-for-precision-medicine-for-high-risk-pediatric-cancer>). These aren't citable in manuscript form as they are ongoing,

but we do cite the PNOC008 and PNOC027 trials underway. This highlights the rapid and real-time utility of the work we are doing through OpenPBTA and will continue through OpenPedCan. Once trials are completed, forthcoming manuscripts will highlight key clinical decisions informed by the research molecular tumor board reports.

2. Overall clarity throughout entire manuscript of which analysis is done on which datasets. The authors have rich and valuable datasets of tumours from primary sample, progressive disease, post-treatment, relapse and second malignancy. Would add value to the analysis to clarify throughout which datasets were used throughout. In addition, clarity in the relationship of these samples in Fig 1B would be highly valuable to the reader and those seeking to understand and use this data in the future. This will also strengthen the authors findings and associations made throughout. Additional results on the molecular changes underpinning the samples that have multiple biospecimens would be highly valuable and provide a unique insight into these tumours.

Apologies for the confusion. We have updated the Somatic Mutational Landscape of Pediatric Brain Tumors section to read: “Unless otherwise noted, each analysis was performed for primary tumors using one tumor per patient.” Table S1 contains all patient sample-level data, including “tumor_descriptor” which annotates the primary, progressive, and post-mortem status of the samples. The treatment information is still being heavily QC-d (see response to Reviewer 1, point 8), and as such, we do not have this information yet available for distribution. All initial tumor biopsies were obtained pre-treatment. We fully agree that changes underpinning samples with multiple biospecimens would be informative and provide new insights into the evolution of these tumors, however, this initial PBTA cohort is skewed toward initial CNS tumors (<https://alexslemonade.github.io/OpenPBTA-analysis/analyses/sample-distribution-analysis/03-tumor-descriptor-an-d-assay-count.nb.html>), with only 42 samples containing paired T/N WGS and RNA-Seq for an initial tumor + a progressive or recurrent tumor (see table below). For this reason, we attempted to sequence as many longitudinal samples as possible for the soon-to-be-released Resnick Kids First X01 FY 21 cohort, which has on the order of ~120 paired samples. This cohort will provide rich insights into tumor evolution for multiple brain tumor types in future studies.

descriptors	n
Initial CNS Tumor	599
Progressive	68
Recurrence	34
Initial CNS Tumor, Progressive	25
Initial CNS Tumor, Recurrence	15
Progressive, Recurrence	5
Second Malignancy	5
Initial CNS Tumor, Second Malignancy	3
Initial CNS Tumor, Progressive, Recurrence	2
Recurrence, Second Malignancy	2
Initial CNS Tumor, Unavailable	1

Unavailable 1

3. Additions to the methodologies deployed in the paper would provide enhancements and aid the authors in drawing conclusions more clearly. In the methodology there is no mention of batch correction applied to the transcriptomics data which comes from multiple centres, various preparation and sequencing technologies even though all were ran through the same pipelines. In addition, tumour purity is not assessed from the WGS data and reported on. Improvements to understanding the transcriptomic landscape of paediatric brain tumours including histological clustering would improve if batch correction was applied and tumour purity was taken into consideration.

The biggest batch within this dataset is library type (stranded vs. polyA), and for this reason, we had separated all derived RNA-Seq release files by batch (eg:

TPM for stranded samples, TPM for polyA samples). We have additionally clarified within the methods and figure legends that these batches were analyzed separately (batched by library) for the transcriptomic modules (eg: UMAP, GSVA, EXTEND, TP53 classification, immune deconvolution). Unless otherwise noted in the manuscript, all transcriptomic figures display stranded RNA-Seq samples. Additionally, in response to Reviewer 1, point 2, we performed UMAP analysis by sequencing center and largely did not find a batch effect (new Supplemental Figure X), thus we are confident in our histological clustering results. Tumor purity is reported within Table S1, column “tumor_fraction”. This was estimated from WGS using TheTA2 (STAR Methods, <http://compbio.cs.brown.edu/projects/theta/>). In addition, in the latest data release created for this revision (v23-20230115), we have gathered biospecimen information regarding extraction and explore the reproducibility of the transcriptomic modules UMAP and GSVA using the TheTA2 DNA tumor purity estimate for RNA samples which were co-extracted with the DNA, positing that co-extracted RNA and DNA would have the same tumor purity. Overall, we observed a median tumor purity of 0.76 across WGS samples and stranded RNA-Seq samples. We do observe potential cancer-specific correlations in tumor purity (eg: lower purity in SEGA, PXA, teratoma below). We now include this as an additional panel in Figure S3A.

We created a new module to create a set of stranded RNA-Seq data thresholded by median tumor purity of the cancer group and rerun selected transcriptomic analyses as suggested above. The module can be found here:

[https://github.com/AlexsLemonade/OpenPBTA-](https://github.com/AlexsLemonade/OpenPBTA-analysis/tree/master/analyses/tumor-purity-exploration)

[analysis/tree/master/analyses/tumor-purity-exploration](https://github.com/AlexsLemonade/OpenPBTA-analysis/tree/master/analyses/tumor-purity-exploration). Overall, the results were broadly consistent with our results reported in the manuscript which utilized the full cohort of samples. Below is a figure we recreated using samples

thresholded for median tumor purity:

Panels A-C: TP53 classifier rerun (compare to manuscript Figure 4A-C)

Panel D: TP53 classifier scores and EXTEND telomerase activity scores (compare to manuscript Figure 4D) Panel E: UMAP of broad histologies (compare to manuscript Figure 5A) Panel F: Immune deconvolution (compare to manuscript

Figure 5C)

4. Co-occurrence analysis should be revisited as the conclusions the authors are drawing are in relation to the entire cohort but clear co-occurrences identified are already known in the literature and cancer subtypes mentioned. For example the authors most significant finding is the co-occurrence of TP53 with H3F3A but this comes as no surprise with the high number of DMG samples in the cohort and this co-occurrence is exclusive to this subtype only. This extends to the co-occurrence of ATRX and EGFR which is also specific to DMG only. The statistical confidence intervals are also rather large and the stats are perhaps more influenced by large cohort numbers and the molecular drivers of this specific tumour type.

Thank you for this comment. We agree that these conclusions agree with what is already known in the literature and in accordance with Reviewer 1, point 1, we have now clarified the results section to highlight what is recapitulating known literature versus what is novel in this manuscript. Please see the full response to Reviewer 1 above related to this. Related to the co-occurrence analysis in general, the confidence intervals are large because the number of mutations per gene is small. Additionally, we had few large enough broad histologies on which to perform this analysis and Figure 3 highlights that our methods for consensus SNV and CNV calls enable us to recapitulate known oncogenic drivers in the literature. We have updated the wording of this section to emphasize that we are observing expectations previously reported and that the pan-brain tumor analysis is driven by HGGs. “We analyzed mutational co-occurrence across the OpenPBTA, using a single tumor from each patient with available WGS (N = 668 patients). The top 50 mutated genes (see STAR Methods for details) in primary tumors are shown in Figure 3 by tumor type (A, bar plots), with co-occurrence scores illustrated in the heatmap (B). As expected, TP53 was the most frequently mutated gene across the OpenPBTA (8.7%, 58/668), significantly co-occurring with H3F3A (OR = 30.05, 95% CI: 14.5 - 62.3, $q = 2.34e-16$), ATRX (OR = 23.3, 95% CI: 9.6 - 56.3, $q = 8.72e-9$), NF1 (OR = 8.26, 95% CI: 3.5 - 19.4, $q = 7.40e-5$), and EGFR (OR = 17.5, 95% CI: 4.8 - 63.9, $q = 2e-4$), with all of these driven by HGGs and consistent with previous reports 32,35,36 . In embryonal tumors, mutations in CTNNB1 significantly co-occurred with mutations in TP53 (OR = 43.6 95% CI: 7.1 - 265.8, $q = 1.52e-3$) as well as with mutations in DDX3X (OR = 21.4, 95% CI: 4.7 - 97.9, $q = 4.15e-3$). These events were driven by medulloblastomas and have been previously reported as significantly mutated in this tumor type 37,38 . Mutations in FGFR1 and PIK3CA

significantly co-occurred in LGGs (OR = 77.25, 95% CI: 10.0 - 596.8, $q = 3.12e-3$), consistent with previous findings 38,39 . Of HGG tumors with mutations in TP53 or PPM1D, 53/55 (96.3%) had mutations in only one of these genes (OR = 0.17, 95% CI: 0.04 - 0.89, $q = 0.056$). This trend recapitulates previous observations that TP53 and PPM1D mutations tend to be mutually exclusive in HGGs 40 .”

Minor concerns:

1. Authors address the versioning of data analysis as changes are made but addressing whether all data presented in the manuscript is re-analysed with additional code improvements would be valuable and the time/cost of each of the analysis modules would be beneficial as analysis occurs in a cloud platform.

Thank you for this comment. While all primary analyses have occurred in AWS EC2 using CAVATICA (via Kids First pipelines) and release data is stored on AWS S3, the majority of development within the OpenPBTA-analysis repository is local, with very few modules requiring an AWS EC2 cloud environment. If EC2 is required, we have noted this, along with the amount of RAM required, in the respective analysis module README on GitHub. Since development has already been done, the time and cost to run a heavier module on EC2 is minimal. Running all analyses underlying the manuscript and generating figures takes 9 hours on an instance with 64GB of RAM and 4 CPUs, costing less than \$10 (current AWS US East pricing). We have additionally included clarifying points within the “Crowd-sourced Somatic Analyses to Create an Open Pediatric Brain Tumor Atlas” section: “We reran all manuscript-specific analysis modules with the latest data release (v23) prior to submission and subsequently created a GitHub repository-tagged release to ensure reproducibility.” “With very few exceptions noted in their respective analysis module READMEs, most modules can be run locally, and since they are run in the project Docker® 20 container, they can easily be scaled and/or used on an Amazon EC2 instance. Importantly, all analyses and manuscript writing were conducted openly throughout the research project, allowing any researcher in the world the opportunity to contribute.”

2. Open collaboration is an excellent resource for code and analysis share to produce data for manuscripts more readily however more information surrounding the process post submission would be beneficial to understand the integrity of the data.

Thank you for this comment. Within the STAR methods section, we discuss the QCs used to ensure high data integrity, including assessment at the nucleic acid

level. Biospecimens are consented and collected either through CBTN or PNO (Please see our recent CBTN publication for more information: <https://doi.org/10.1016/j.neo.2022.100846>). Concerning data QC, we included the following section with initial submission: “Quality Control of Sequencing Data To confirm sample matches and remove mis-matched samples from the dataset, we performed NGSCheckMate 82 on matched tumor/normal CRAM files. Briefly, we processed CRAMs using BCFtools to filter and call 20k common single nucleotide polymorphisms (SNPs) using default parameters. We used the resulting VCFs to run NGSCheckMate. Per NGSCheckMate author recommendations, we used ≤ 0.61 as a correlation coefficient cutoff at sequencing depths > 10 to predict mis-matched samples. We determined RNA-Seq read strandedness by running the `infer_experiment.py` script from RNA-SeQC83 on the first 200k mapped reads. We removed any samples whose calculated strandedness did not match strandedness information provided by the sequencing center. We required that at least 60% of RNA-Seq reads mapped to the human reference for samples to be included in analysis.”

Does the contributor get involved in deeper discussions of the data if required, is any data rejected?

Throughout the life of the OpenPBTA project, we work very closely with code contributors through a back-and-forth pull request, code review, code comment, code update, modified analysis process. If a scientific analysis is complex or requires domain expertise, we often have multiple reviewers look at the pull request such that all bases are covered, from code to statistics to biological inferences. We work collaboratively to produce the data in a rigorous manner. We have removed a sample’s data if we identified, through collaborative analyses and pathology review, that it was mis-identified or potentially swapped. Beyond rigorous review of the data is there quality assurance checks where two separate centres analyse the same data with the modules to ensure reproducibility? Through code review, two or more analysts run the same analysis within the project docker container, ensuring reproducibility by everyone who uses the modules. We added clarifying text to the “Crowd-sourced Somatic Analyses to Create an Open Pediatric Brain Tumor Atlas” section: “Importantly, this peer review process entailed two or more analysts running the same code within the same Docker® 20 container to ensure reproducibility of results that were derived from a specific data release.” 3. Authors mention WHO classification 2021 but report on 2016 classification, updating to the latest WHO classifications would improve clarity around specific

subtypes. For example in the DMG oncoplot the new subtype of DMG H3K-altered is apparent in these samples. We apologize for the confusion. When the project started in 2019, the WHO 2021 subtypes were not available, but they were published just before our initial submission, so we had mentioned them in the manuscript. Since updating 2016 to 2021 classifications would take a significant amount of time, we have removed reference to the 2021 classifications. We also acknowledge that some DMG H3 K28-altered samples will exist in this cohort, however we aimed to be definitive with our molecular subtyping in OpenPBTA. That is, we would only be comfortable calling these if, in addition to the pathogenic EZHIP or EGFR alterations, we had orthogonal data showing either loss of H3 K28me3 via immunohistochemistry (IHC) or methylation subtype H3 K28. At this time, we do not have these data within OpenPBTA (and methylation is forthcoming in OpenPedCan v12), so we will update through OpenPedCan. In addition, as many WHO 2021 classifications as possible will be updated within the OpenPedCan project (<https://github.com/PediatricOpenTargets/OpenPedCan-analysis>).

3. Clarity in methodology around somatic mutation analysis in samples lacking matching germline needs to be specified and clarified.

Apologies for the lack of clarity. All tumor samples analyzed herein do indeed have matched normal samples and variants were called from paired analysis. We note this in the section, “Crowd-sourced Somatic Analyses to Create an Open Pediatric Brain Tumor Atlas” “We previously performed whole genome sequencing (WGS), whole exome sequencing (WXS), and RNA sequencing (RNA-Seq) on matched tumor and normal tissues as well as selected cell lines 16 from 943 patients from the Pediatric Brain Tumor Atlas (PBTA), consisting of 911 patients from the Children’s Brain Tumor Network (CBTN) 4 and 32 patients from the Pacific Pediatric Neuro-Oncology Consortium (PNOC) 12,17 (Figure 1A). We then harnessed, and built upon, the benchmarking efforts of the Gabriella Miller Kids First Data Resource Center to develop robust and reproducible data analysis workflows within the CAVATICA platform to perform primary somatic analyses including calling of single nucleotide variants (SNVs), copy number variants (CNVs), structural variants (SVs), and gene fusions, often implementing multiple complementary methods (Figure S1) and STAR Methods).”

4. Telomerase scores are they related to TERT rearrangement events or TERT promoter mutations. Authors mention in the methodology that they explored TERT promoter mutations but no mention if there is a correlation in the results. **Apologies for the confusion. Our mention of TERT promoter**

(TERTp) mutations in the STAR methods refers to mutation hotspot scavenging. Since we took a conservative SNV consensus of 3/3 variant callers, we missed hotspot mutations called in 1/3 or 2/3 variant callers in a small number of cases. We utilized MSKCC's hotspot database to retrieve these calls, however, canonical TERT promoter mutations (-124C>T and -146C>T) were not included in their database, and we added these to our list. The EXTEND algorithm (<https://www.ncbi.nlm.nih.gov/pmc/articles/PMC7794223/>) generates telomerase activity scores from a 13-gene signature, positing the higher the score, the increased telomerase activity. The authors indeed show that samples with TERT rearrangements have higher telomerase scores, but did not fully assess the effect of TERTp. The authors suggest that one cannot rely solely on TERTp or TERT expression for telomerase activity prediction. They caution that "while -124 and -146 promoter mutations enhance the transcriptional output of TERT, the TERT gene can transcribe >20 splicing isoforms 30, but only the full-length transcript bearing all 16 exons can produce the catalytic subunit 31,32,33. Second, single-cell imaging studies showed that most TERT mRNAs localize in the nucleus, but not in the cytoplasm, and thus are not translated 34. Third, endogenous TERT protein and TERC are far more abundant than the assembled telomerase complex in cancer cell lines 23. Finally, TERC and accessory proteins can also impact telomerase activity. For example, telomerase activity in human T cells has been reported to relate to TERC levels rather than TERT 35. In addition, mutations in DKC1 and TERC both cause dyskeratosis congenita, a rare genetic syndrome related to impaired telomerase. 10,36" In the OpenPBTA, we found 3 tumors with TERT SV and eight tumors with TERTp mutations. TERT SVs were not annotated as pathogenic or likely pathogenic by dbVar. We now annotate TERTp samples which had stranded RNA-Seq performed in Supplemental Figure 5B-C. There was no statistically significant difference between EXTEND scores in TERTp and non-mutated samples ($W = 1840$, $p\text{-value} = 0.1196$), and this may very well be expected, as indicated above, TERT requires a specific transcript and subunits to generate active telomerase.

SampleID	tertp	tertSV	NormEXTENDScores	RNA_library
BS_02NZT8CE	TERTp mutation not observed	TERT SV present	0.37	stranded
BS_BMMY0H73	TERTp mutation not observed	TERT SV present	0.62	polya
BS_BA6AZWB3	TERTp mutation present	TERT SV not observed	0.57	stranded
BS_J1AB8PMA	TERTp mutation present	TERT SV not observed	0.72	stranded
BS_M15PW2N0	TERTp mutation present	TERT SV not observed	0.67	stranded
BS_MB7WN0ZB	TERTp mutation present	TERT SV not observed	0.47	stranded
BS_VMQMGSJY	TERTp mutation present	TERT SV not observed	0.37	stranded
BS_X4X1SFSE	TERTp mutation present	TERT SV not observed	0.47	stranded
BS_0VXZCRJS	TERTp mutation present	TERT SV not observed	0.67	polya
BS_NPA6CZQF	TERTp mutation present	TERT SV not observed	0.75	polya

6. Conclusions drawn about immunologically cold and hot samples is poorly defined and assessed with limited supporting evidence to state such claims.

Thank you for this feedback. We have removed references to immunologically hot and cold tumors and instead refer back to the manuscripts using the CD8+/CD4+ T cell ratios to infer response to either adoptive T cell therapy or PD-L1 inhibition, backing off of the strength of our claims.

7. Discussion mentions using packages MM2S and medulloPackage to subtype medulloblastoma tumours but not shown in results.

Thank you for bringing this to our attention. We have now added a sentence to the results explaining how these tumors were subtyped: “Uniquely, we used transcriptomic classification to subtype 122 medulloblastomas into SHH, WNT, Group 3, or Group 4 with MedulloClassifier 24 and MM2S 25 , achieving accuracies of 95% (41/43) and 91% (39/43), respectively.”

8. No mention of z-score calculation approach in methodology

Z-scores were calculated using $z = (x - \mu) / \sigma$ and we now have added this to the QUANTIFICATION AND STATISTICAL ANALYSIS section in STAR Methods.

9. In consistency between ependymal and ependymoma tumour this needs to be harmonised throughout

Apologies for the confusion. We had used “ependymal tumor” as the “broad histology”, aligning with the WHO 2016 broad categorization of all ependymal tumors and “ependymoma” when referring to a specific tumor or subset of tumors (<https://pubmed.ncbi.nlm.nih.gov/27157931/>). For clarity, we removed manuscript references to “Ependymal tumor” and replaced these with “Ependymoma(s)”.

Referees' report, second round of review

Reviewer 1

There is progress with the current revision, but a number of issues remain. I have three main concerns: 1) the batch effects issues involving RNA-seq are not adequately described in the manuscript; 2) the practice of selecting one tumor profile per patient, in the instance of multiple tumors per patient, is not well described in the manuscript; 3) The supplementary Excel files need to better capture the sample annotation and source data represented in the figures.

I remain unimpressed with the "open science" model and framework used to generate this manuscript. However, I don't necessarily need to be convinced otherwise. What's important here is that other analysts can work with the underlying CBTN data without being tied to working within the OpenPBTA framework and code. This involves making the underlying Source data available in the Supplementary files of the paper itself.

This OpenPBTA framework represents a very different way of working from what most bioinformatics/genomics analysts would be used to. Big science genomics consortiums such as TCGA, CPTAC, and ICGC have historically carried out their analyses very differently from OpenPBTA. I understand we went through a pandemic, but that hasn't stopped CPTAC from putting out several consortium papers during that time period, for example. The CPTAC/CBTN pediatric brain paper has been out for 2 ½ years now, and it seems that there is a second CBTN proteogenomics paper in preparation. It's fine if the authors see OpenPBTA as a possibly better way for doing integrative analyses in some respects, but the different perspectives represented by the readership should be appreciated in order to maximize the potential of the combined CBTN/PNOC datasets as a resource. Specific ways to improve the resource potential are noted below.

1) In response to my initial query, we find out that there are in fact serious batch effects in the RNA-seq data. As the authors note in their rebuttal: "Since stranded and polyA datasets have a large batch effect due to library type, we purposely separated the gene expression matrices by library type and subsequently performed all RNA-Seq specific analyses on those datasets separately." So, some analyses were not done on all tumors, but only in the majority that were of the stranded group, where most CBTN samples are stranded and the PNOC samples are PolyA. This is a major issue that anyone else who may wish to analyze these

data would need to take into account. But the issue is not highlighted particularly well in the manuscript itself; e.g., the term "batch effect" is not mentioned anywhere. The authors responded to my question on batch effects with a lot of information in the rebuttal letter itself, but this type of "inside baseball" needs to be conveyed in the paper itself. The authors need to make clear in the paper that anyone using the RNA-seq data needs to find a way to deal with the batch effects issue, e.g., this needs to be highlighted in the Results section (which should also explicitly note the distinction between stranded vs polyA; currently the term "stranded" is introduced in Results without any explanation). What types of analyses can be carried out on the full CBTN/PNOC dataset and what types should be carried out separately by batch? This issue seems to cut against the notion that OpenPBTA represents a harmonization between the CBTN and PNOC datasets. There might have been an opportunity here to effectively harmonize the two datasets by removing the RNA-seq batch effects issues, and thereby greatly increase the resource value of the study. However, this will have to be left to outside users of the data to deal with.

2) I don't really understand the author's stated practice of selecting one tumor profile per patient, in the instance of multiple tumors per patient. I suppose it's acceptable, so long as this practice is better communicated in the manuscript Results section. We know from previous papers analyzing multiple CBTN tumors from the same patient that these multiple tumors are not homogenous at the molecular level. For example, the CBTN/CPTAC paper, which analyzed 218 tumor samples from 199 patients, clearly demonstrated within patient differences (e.g., Figure 7). As noted in the present manuscript: "we present a comprehensive, collaborative, open genomic analysis of 1,074 tumors and 22 cell lines, comprised of 58 distinct brain tumor histologies from 943 patients." But not all 1,074 tumors were analyzed apparently, so which specific ones were? The authors state in their rebuttal: "Throughout this manuscript, we present the findings for primary tumors (one per patient), unless otherwise noted (for example, Table 2, Figure 4E), and we have now updated the results section to reflect this." But the Results section does not make this aspect of the study clear. For example, Figure 2 legend now states "N denotes the number of patients and tumors." But this does not clarify that only one tumor from each patient was analyzed. It needs to be clearly stated at the beginning of Results that one tumor per patient was selected for most analyses, and then for each analysis the authors need to state in the Results section and corresponding figure legend the number of tumors and patients analyzed. Also the Supplementary Data needs a tumor-level table that indicates

what analyses were done for which tumor.

3) The supplementary Data in the paper itself needs to be more complete and better organized. Nature journals require for their papers a Source Data Excel file, which as a minimum, contains the raw data underlying any graphs and charts (<https://www.nature.com/ncomms/submit/how-to-submit>). This should be required for the present manuscript. For example, for Figure 2 (mutation events), I should be able to get all the underlying information in an Excel table in the Supplementary data of the paper, from which I could re-generate Figure 2 if I wanted to. That means annotating which of the 1,074 tumors have BRAF fusions, which have RELA fusions, which have other fusions or mutations represented in Figure 2. We need a sample/tumor level Excel table, with 1,074 entries. This table would include the following information: tumor id, patient id, data platform availability (WGS yes/no, WXS yes/no; RNA-seq yes/no, etc), whether the tumor was not used in specific analyses (in the instance of multiple tumors per patient), disease status (recurrent, primary, etc.), histology from tissue source site, molecular subtype, final histology call made by OpenPBTA, PolyA/stranded, CBTN/PNOC, key mutation events and fusion events (e.g., those represented in Figure 2), and other information current represented in Tables S1, S2, and S3. It could all go into one Excel Table. If you want to break it down into separate tables, then keep the same tumor ordering for each table. There are plenty of good examples in the literature of providing a sample-level table in supplemental. I will point out one example from TCGA's Lung Adenocarcinoma paper (<https://www.ncbi.nlm.nih.gov/pmc/articles/PMC4231481/>) Supplementary Information 2 of that paper includes a "S_Table 7-Clinical&Molec_Summar" which represents exactly the type of format I want to see (also please see "S_Table 1_Samples" which has additional relevant information including available data platforms). Why can't we have something similar for this paper? I find the author's responses to my initial comments #3, 4, and 7 of the first version to be unacceptable. The comment #7 response refers me to the entire datasets, e.g. as stored in Cavatica or PedCBioportal. If I have to do that to get the data represented in the figures, then what is the Resource value represented by this manuscript? Gene fusion calls by RNA-seq can be tricky to parse out, for example. This present manuscript represents a snapshot of the combined CBTN/PNOC datasets, and this snapshot needs to be properly documented as part of the journal article itself. Links to outside datasets and the public datasets themselves are subject to change. In addition, Data File S1 is not a sample-level table; it's an analyte level table; for a given tumor, there's an entry for RNA-seq and a separate

entry for WGS; that's not what's being requested here.

Reviewer 2

The authors have addressed all of the concerns that were addressed previously. However, there are two minor comments that I feel need to be addressed further in the manuscript:

1. The authors did an excellent job comparing the effects of tumour purity on the effects of the transcriptomics data. Adding a comment in the manuscript related to the effects of tumour purity on the transcriptomic output is needed. I agree with the authors review that results are consistent but a tighter and less disperse data analysis comes from controlling for purity, especially in the UMAP clustering of the broad histologies. A comment on this observation would be highly valuable.
2. A statement should be added to the methods specifying the rationale for removal of data related to the process and criteria warranting the removal.

Otherwise, all other concerns raised have been addressed adequately and would very much like to see this highly important resource published in Cell Genomics which is extremely beneficial to the global paediatric oncology research community.

Authors' response to the second round of review

We thank both reviewers for their time reading and critically reviewing our manuscript. We believe their suggestions have strengthened this work and hope our dataset and code base become resources used by many in the pediatric oncology community. In response to the last set of reviewer comments, we made text updates (see point to point responses), added a new Supplemental Figure S7 to transparently display additional information about RNA library batches and tumor purity assessment, include a Limitations of the Study Section following the Discussion, and created a Zenodo dataset containing data underlying each Figure panel where relevant.

Reviewer #1

There is progress with the current revision, but a number of issues remain. I have three main concerns: 1) the batch effects issues involving RNA-seq are not

adequately described in the manuscript; 2) the practice of selecting one tumor profile per patient, in the instance of multiple tumors per patient, is not well described in the manuscript; 3) The supplementary Excel files need to better capture the sample annotation and source data represented in the figures. I remain unimpressed with the "open science" model and framework used to generate this manuscript. However, I don't necessarily need to be convinced otherwise. What's important here is that other analysts can work with the underlying CBTN data without being tied to working within the OpenPBTA framework and code. This involves making the underlying Source data available in the Supplementary files of the paper itself. This OpenPBTA framework represents a very different way of working from what most bioinformatics/genomics analysts would be used to. Big science genomics consortiums such as TCGA, CPTAC, and ICGC have historically carried out their analyses very differently from OpenPBTA. I understand we went through a pandemic, but that hasn't stopped CPTAC from putting out several consortium papers during that time period, for example. The CPTAC/CBTN pediatric brain paper has been out for 2 ½ years now, and it seems that there is a second CBTN proteogenomics paper in preparation. It's fine if the authors see OpenPBTA as a possibly better way for doing integrative analyses in some respects, but the different perspectives represented by the readership should be appreciated in order to maximize the potential of the combined CBTN/PNOC datasets as a resource. Specific ways to improve the resource potential are noted below.

1) In response to my initial query, we find out that there are in fact serious batch effects in the RNA-seq data. As the authors note in their rebuttal: "Since stranded and polyA datasets have a large batch effect due to library type, we purposely separated the gene expression matrices by library type and subsequently performed all RNA-Seq specific analyses on those datasets separately." So, some analyses were not done on all tumors, but only in the majority that were of the stranded group, where most CBTN samples are stranded and the PNOC samples are Response to Reviewers PolyA. This is a major issue that anyone else who may wish to analyze these data would need to take into account. But the issue is not highlighted particularly well in the manuscript itself; e.g., the term "batch effect" is not mentioned anywhere. The authors responded to my question on batch effects with a lot of information in the rebuttal letter itself, but this type of "inside

baseball" needs to be conveyed in the paper itself. The authors need to make clear in the paper that anyone using the RNA-seq data needs to find a way to deal with the batch effects issue, e.g., this needs to be highlighted in the Results section (which should also explicitly note the distinction between stranded vs polyA; currently the term "stranded" is introduced in Results without any explanation). What types of analyses can be carried out on the full CBTN/PNOC dataset and what types should be carried out separately by batch? This issue seems to cut against the notion that OpenPBTA represents a harmonization between the CBTN and PNOC datasets. There might have been an opportunity here to effectively harmonize the two datasets by removing the RNA-seq batch effects issues, and thereby greatly increase the resource value of the study. However, this will have to be left to outside users of the data to deal with.

We agree with the reviewer that the manuscript can be clarified to explicitly state that RNA-Seq samples within the OpenPBTA have undergone multiple types of library preparation. Supplemental Table 1 contains a column called "RNA_library" to denote stranded or poly-A preparation. We further expand upon this in the STAR Methods, Transcriptomic Analysis Results, and Limitations sections and include panels in Figure S7 to transparently display the library makeup of the dataset. Batch effects that arise from different library preparation strategies are well-known in the pediatric cancer research community, and there is no consensus about what approach should be used to correct for library preparation. For example, our collaborators at UCSC Treehouse have been working on this issue for several years and continue to release transcriptome data separated by library type. (See downloads here: <https://treehousegenomics.soe.ucsc.edu/public-data/#datasets>). To follow this convention, we have also released our data batched by library type (see below for example filenames). We did this to make it easier for users of this data. We anticipate the user(s) will read the description of the data files (<https://github.com/AlexsLemonade/OpenPBTA-analysis/blob/master/doc/data-files-description.md>) and/or note that the suffix of the file contains the library, thereby enabling the user(s) to make their own conscious choices to put the data together or keep it separate, depending on the analyses being performed. ppta-gene-counts-rsem-expected_count.polya.rds ppta-gene-counts-rsem-expected_count.stranded.rds ppta-gene-expression-kallisto.polya.rds ppta-gene-

expression-kallisto.stranded.rds pbta-gene-expression-rsem-fpkm-collapsed.polya.rds pbta-gene-expression-rsem-fpkm-collapsed.stranded.rds pbta-gene-expression-rsem-fpkm.polya.rds pbta-gene-expression-rsem-fpkm.stranded.rds pbta-gene-expression-rsem-tpm.polya.rds pbta-gene-expression-rsem-tpm.stranded.rds pbta-isoform-counts-rsem-expected_count.polya.rds pbta-isoform-counts-rsem-expected_count.stranded.rds pbta-isoform-expression-rsem-tpm.polya.rds pbta-isoform-expression-rsem-tpm.stranded.rds All samples were harmonized using the same upstream workflows. As noted in our initial response to reviewers' concerns about batch effect, the disease types are not balanced between library preparation strategies. We did not perform batch correction because removal of batch effects across such unbalanced groups may induce false differences between groups in some cases (1,2). We acknowledge batch effect correction methods that allow for specifying classes (i.e., disease type) are available (e.g., Surrogate Variable Analysis). However, we maintain that batch correction strategy should be highly dependent on an analyst's goals (2), and we cannot offer researchers a batch-corrected transcriptomic dataset that is appropriate for all use cases or experimental questions. This is now highlighted in the Limitations section.

2) I don't really understand the author's stated practice of selecting one tumor profile per patient, in the instance of multiple tumors per patient. I suppose it's acceptable, so long as this practice is better communicated in the manuscript Results section. We know from previous papers analyzing multiple CBTN tumors from the same patient that these multiple tumors are not homogenous at the molecular level. For example, the CBTN/CPTAC paper, which analyzed 218 tumor samples from 199 patients, clearly demonstrated within patient differences (e.g., Figure 7). As noted in the present manuscript: "we present a comprehensive, collaborative, open genomic analysis of 1,074 tumors and 22 cell lines, comprised of 58 distinct brain tumor histologies from 943 patients." But not all 1,074 tumors were analyzed apparently, so which specific ones were? The authors state in their rebuttal: "Throughout this manuscript, we present the findings for primary tumors (one per patient), unless otherwise noted (for example, Table 2, Figure 4E), and we have now updated the results section to reflect this." But the Results section does not make this aspect of the study clear. For example, Figure 2 legend

now states "N denotes the number of patients and tumors." But this does not clarify that only one tumor from each patient was analyzed. It needs to be clearly stated at the beginning of Results that one tumor per patient was selected for most analyses, and then for each analysis the authors need to state in the Results section and corresponding figure legend the number of tumors and patients analyzed. Also the Supplementary Data needs a tumor-level table that indicates what analyses were done for which tumor. In order to determine accurate frequencies of genetic alterations, recurrent events, and/or mutational co-occurrence or mutual exclusivity within cancer types, we ensured that we did not double count tumors through creation of an independent specimens module, which selects a single specimen per patient by prioritizing primary tumors and those with WGS available and randomly selecting a single specimen otherwise. We agree that no two samples are likely to be identical transcriptomically or at the DNA level, and acknowledge there is still much work to be done to investigate intratumoral heterogeneity of brain tumor samples. Our intention here was to convince the readers that the PBTA represents a high-quality clinically-annotated data resource faithfully characterized through the OpenPBTA, an innovative means of working collaboratively, reproducibly, and openly with researchers globally. For reference, 30/813 patients with primary tumor DNA available had replicate DNA sequencing data for their primary tumors (17 had duplicate WGS and the remaining 13 had duplicate WGS and/or WXS performed), and conclusions were unchanged during the revision with a new independent sample selection (release v22 to release v23). Figure 7 of the Petralia et. al. does indeed show differences within patient tumors, but it is important to note that with two exceptions (LGG: 7316-485, 7316-2193 and Ganglioglioma: 7316-2901, 7316-156), the tumors were derived from different phases of therapy, in which one expects tumor evolution and divergence – that is, one from the initial tumor and one from a progression or recurrence, or one from a progression and one at recurrence or autopsy. We previously noted in Point 2 to Reviewer 2 that we did not have enough samples to robustly investigate tumor evolution in this initial manuscript, which is why we largely focus on primary tumors. These points are now included in the Limitations Section. >As noted in the present manuscript: "we present a comprehensive, collaborative, open genomic analysis of 1,074 tumors and 22 cell lines, comprised of 58 distinct brain tumor histologies from 943 patients." But not all 1,074 tumors were analyzed apparently, so which specific ones were? We

apologize for the confusion. 1,074 is the total number of unique tumors and 22 is the total number of unique cell lines within the PBTA cohort (CBTN + PNOC). These are the total number of tumor collection events (denoted as “sample_id” in Table S1). Indeed, these were all processed through the OpenPBTA modules, and we have resulting data for all of them within the GitHub repository and on PedCbioPortal. Within the figures, we had to choose which data to present and in the manuscript, we largely chose to present the primary tumor data unless otherwise specified.

The authors state in their rebuttal: "Throughout this manuscript, we present the findings for primary tumors (one per patient), unless otherwise noted (for example, Table 2, Figure 4E), and we have now updated the results section to reflect this." But the Results section does not make this aspect of the study clear. The Results section was previously updated in the following places to denote when we depict only primary tumors using one tumor per patient: Under “Somatic Mutational Landscape”, we state: Paragraph 1: Unless otherwise noted, each analysis was performed for diagnostic tumors using one tumor per patient. Paragraph 2: Figure 2 and Figure S3B depict oncoprints recapitulating known histology-specific driver genes in primary tumors across OpenPBTA histologies, and Table S2 summarizes all detected alterations across cancer groups. Figure 2 Legend: Frequencies of canonical somatic gene mutations, CNVs, fusions, and TMB (top bar plot) for the top mutated genes across primary tumors within the OpenPBTA dataset...N denotes the number of unique tumors (one tumor per patient). Figure S3 Legend: N denotes the number of unique tumors with one tumor per patient used. Under “Mutational co-occurrence..” Section: Paragraph 1: We analyzed mutational co-occurrence across the OpenPBTA, using a single tumor from each patient (N = 668) with WGS. The top 50 mutated genes (see STAR Methods for details) in primary tumors are shown in Figure 3 by tumor type (A, bar plots), with co-occurrence scores illustrated in the heatmap (B). In “Oncoprint figure generation” Methods Section: Paragraph 1: We used Maftools 109 to generate oncoprints depicting the frequencies of canonical somatic gene mutations, CNVs, and fusions for the top 20 genes mutated across primary tumors within broad histologies of the OpenPBTA dataset. In “Independent specimens” Methods Section: Paragraph 1: Certain analyses required that we select only a single representative specimen for each individual. In these cases, we identified a

single specimen by prioritizing primary tumors and those with whole-genome sequencing available. If this filtering still resulted in multiple specimens, we randomly selected a single specimen from the remaining set. >For example, Figure 2 legend now states "N denotes the number of patients and tumors." But this does not clarify that only one tumor from each patient was analyzed. Figure 2 and S3 legends now state: "N denotes the number of unique tumors (one tumor per patient)."

3) The supplementary Data in the paper itself needs to be more complete and better organized. Nature journals require for their papers a Source Data Excel file, which as a minimum, contains the raw data underlying any graphs and charts (<https://www.nature.com/ncomms/submit/how-to-submit>). This should be required for the present manuscript. For example, for Figure 2 (mutation events), I should be able to get all the underlying information in an Excel table in the Supplementary data of the paper, from which I could re-generate Figure 2 if I wanted to. That means annotating which of the 1,074 tumors have BRAF fusions, which have RELA fusions, which have other fusions or mutations represented in Figure 2. We need a sample/tumor level Excel table, with 1,074 entries. This table would include the following information: tumor id, patient id, data platform availability (WGS yes/no, WXS yes/no; RNA-seq yes/no, etc), whether the tumor was not used in specific analyses (in the instance of multiple tumors per patient), disease status (recurrent, primary, etc.), histology from tissue source site, molecular subtype, final histology call made by OpenPBTA, PolyA/stranded, CBTN/PNOC, key mutation events and fusion events (e.g., those represented in Figure 2), and other information current represented in Tables S1, S2, and S3. It could all go into one Excel Table. If you want to break it down into separate tables, then keep the same tumor ordering for each table. There are plenty of good examples in the literature of providing a sample-level table in supplemental. I will point out one example from TCGA's Lung Adenocarcinoma paper (<https://www.ncbi.nlm.nih.gov/pmc/articles/PMC4231481/>) Supplementary Information 2 of that paper includes a "S_Table 7-Clinical&Molec_Summar" which represents exactly the type of format I want to see (also please see "S_Table 1_Samples" which has additional relevant information including available data platforms). Why can't we have something similar for this paper? I find the author's responses to my initial comments #3, 4, and 7 of the first version to be

unacceptable. The comment #7 response refers me to the entire datasets, e.g. as stored in Cavatica or PedCBioportal. If I have to do that to get the data represented in the figures, then what is the Resource value represented by this manuscript? Gene fusion calls by RNA-seq can be tricky to parse out, for example. This present manuscript represents a snapshot of the combined CBTN/PNOC datasets, and this snapshot needs to be properly documented as part of the journal article itself. Links to outside datasets and the public datasets themselves are subject to change. In addition, Data File S1 is not a sample-level table; it's an analyte level table; for a given tumor, there's an entry for RNA-seq and a separate entry for WGS; that's not what's being requested here.

In accordance with Cell Press authorship guidelines, we openly share all primary and secondary somatic data along with all code used to generate that data and the manuscript figures within AWS and a GitHub repository (<https://github.com/AlexsLemonade/OpenPBTA-analysis>), meeting criteria for digital longevity (creating releases and DOIs) and FAIR sharing standards. Creating a GitHub repository release and DOI ensures that the data and analyses are tied to a final manuscript version (here, the revision was 0.2.0: <https://github.com/AlexsLemonade/OpenPBTA-analysis/releases/tag/v0.2.0> and are not subject to change if a user downloaded them using the DOI (provided in the STAR Key Resources table). Within this GitHub repository, we provide one script which runs every manuscript analysis (<https://github.com/AlexsLemonade/OpenPBTA-analysis/blob/master/scripts/run-manuscript-analyses.sh>), one script which generates every manuscript figure (<https://github.com/AlexsLemonade/OpenPBTA-analysis/blob/master/figures/generate-figures.sh>), and one script which generates every manuscript table (<https://github.com/AlexsLemonade/OpenPBTA-analysis/blob/master/tables/run-manuscript-tables.sh>). Any user can go to v0.2.0, download the release data, pull the Docker image, rerun all analyses and identically reproduce the entire manuscript's figures and tables. This is the essence of OpenPBTA. The resource value of OpenPBTA is the harmonized and consensus calls for SNVs, CNVs, SVs, fusions, and expression, as well as the >40 downstream analysis modules whose code and data are available within those GitHub modules. For example, if a

researcher is interested in fusions, we provide raw STAR-fusion calls (pbta-fusion-starfusion.tsv.gz), raw Arriba fusion calls (pbta-fusion-arriba.tsv.gz), and an artifact-filtered, expression-filtered, oncogene+tumor suppressor gene+kinase+transcription factor annotated, and kinase domain retention annotated fusion file (pbta-fusion-putative-oncogenic.tsv) all readily available via AWS S3 or CAVATICA download. The putative oncogenic fusions are also visible in the OpenPBTA PedcBioportal study

(<https://pedcbioportal.org/study/summary?id=openpbta>), which will be frozen at the final version of the manuscript. We posit that researchers may find the full dataset useful, even though we present fusions from primary tumors in Figure 2. That being said, we recognize the value of making the data underlying individual figures more readily available so a broad audience of researchers may reuse it. We created a Zenodo dataset containing one CSV (comma-separated values) file per figure panel, accompanied by a README describing each table, and ordered by sample identifier when applicable. This will enable researchers to access data underlying each figure in a machine-readable format and directly examine which samples are included in a figure. Additionally, we now include a table of molecular alterations shown in Figure 2 for all tumor and cell line samples in the PBTA dataset in the Zenodo upload as a CSV file. Zenodo assigns a versioned DOI such that once this dataset is published, we cannot change the files that were included per the Zenodo documentation. Therefore, there will be a snapshot of the data underlying the figures, including Figure 2, tied to the final version of the manuscript with this approach. The Zenodo DOI for this OpenPBTA manuscript is: 10.5281/zenodo.7805408. References 1. Nygaard V, Rødland EA, Hovig E.

Methods that remove batch effects while retaining group differences may lead to exaggerated confidence in downstream analyses. *Biostatistics*. 2016 Jan;17(1):29–39. 2. Goh WWB, Wang W, Wong L. Why Batch Effects Matter in Omics Data, and How to Avoid Them. *Trends Biotechnol*. 2017 Jun;35(6):498–507.

Reviewer #2

The authors have addressed all of the concerns that were addressed previously. However, there are two minor comments that I feel need to be addressed further in the manuscript:

1. The authors did an excellent job comparing the effects of tumour purity on the effects of the transcriptomics data. Adding a comment in the manuscript related to the effects of tumour purity on the transcriptomic output is needed. I agree with the authors review that results are consistent but a tighter and less disperse data analysis comes from controlling for purity, especially in the UMAP clustering of the broad histologies. A comment on this observation would be highly valuable. We agree that this information is useful to the readers. We have now included a new Supplemental Figure S7 to transparently convey information about RNA libraries, sequencing centers, and tumor purity: (A) a bar plot showing distributions of disease type, broken up by library prep strategy (illustrating unbalanced disease types by library prep) (B) a UMAP plotting samples from both selection strategies on the same coordinates (C) the UMAP plot highlighting sequencing center previously included in response to Reviewer 1 point 2 (D-I) The thresholded tumor purity figure we previously included in the response to reviewers (Reviewer 2 point 3) In addition, we provide a comment at the end of the results section discussing the additional analyses performed controlling for high purity samples, tighter UMAP clustering, and overall result concordance with the total stranded sample analysis.

2. A statement should be added to the methods specifying the rationale for removal of data related to the process and criteria warranting the removal.

We have now added more information about the sample removal process in the STAR Methods section.

Otherwise, all other concerns raised have been addressed adequately and would very much like to see this highly important resource published in Cell Genomics which is extremely beneficial to the global paediatric oncology research community